# Linking aberrant chromatin features in chronic lymphocytic leukemia to transcription factor networks

Jan-Philipp Mallm[1,a], Murat Iskar[2,a], Naveed Ishaque[3,†,a], Lara C Klett[1,4], Sabrina J Kugler[5,6], Jose M Muino[7,‡], Vladimir B Teif[8], Alexandra M Poos[1,4,9,10], Sebastian Großmann[1,§], Fabian Erdel[1,11], Daniele Tavernari[1,¶], Sandra D Koser[12], Sabrina Schumacher[1], Benedikt Brors[12], Rainer König[9,10], Daniel Remondini[13], Martin Vingron[7], Stephan Stilgenbauer[6,††], Peter Lichter[2,14], Marc Zapatka[2], Daniel Mertens[5,6,*,b] & Karsten Rippe[1,**,b]

## Abstract

In chronic lymphocytic leukemia (CLL), a diverse set of genetic mutations is embedded in a deregulated epigenetic landscape that drives cancerogenesis. To elucidate the role of aberrant chromatin features, we mapped DNA methylation, seven histone modifications, nucleosome positions, chromatin accessibility, binding of EBF1 and CTCF, as well as the transcriptome of B cells from CLL patients and healthy donors. A globally increased histone deacetylase activity was detected and half of the genome comprised transcriptionally downregulated partially DNA methylated domains demarcated by CTCF. CLL samples displayed a H3K4me3 redistribution and nucleosome gain at promoters as well as changes of enhancer activity and enhancer linkage to target genes. A DNA binding motif analysis identified transcription factors that gained or lost binding in CLL at sites with aberrant chromatin features. These findings were integrated into a gene regulatory enhancer containing network enriched for B-cell receptor signaling pathway components. Our study predicts novel molecular links to targets of CLL therapies and provides a valuable resource for further studies on the epigenetic contribution to the disease.

**Keywords** bivalent promoter; DNA methylation; enhancer; gene regulatory networks; histone modifications

**Subject Categories** Cancer; Chromatin, Epigenetics, Genomics & Functional Genomics; Genome-Scale & Integrative Biology

**Mol Syst Biol. (2019) 15: e8339**

## Introduction

Genomic sequence analysis has identified a comprehensive set of leukemogenic candidate genes in chronic lymphocytic leukemia (CLL; Martin-Subero *et al*, 2013; Landau *et al*, 2015; Puente *et al*, 2015).

1 Division of Chromatin Networks, German Cancer Research Center (DKFZ) and Bioquant, Heidelberg, Germany
2 Division of Molecular Genetics, German Cancer Research Center (DKFZ), Heidelberg, Germany
3 Division of Theoretical Bioinformatics and Heidelberg Center for Personalized Oncology, German Cancer Research Center (DKFZ), Heidelberg, Germany
4 Faculty of Biosciences, Heidelberg University, Heidelberg, Germany
5 Mechanisms of Leukemogenesis, German Cancer Research Center (DKFZ), Heidelberg, Germany
6 Department of Internal Medicine III, University Hospital Ulm, Ulm, Germany
7 Department of Computational Molecular Biology, Max Planck Institute for Molecular Genetics, Berlin, Germany
8 School of Biological Sciences, University of Essex, Colchester, UK
9 Integrated Research and Treatment Center, Center for Sepsis Control and Care (CSCC), Jena University Hospital, Jena, Germany
10 Network Modeling, Leibniz Institute for Natural Product Research and Infection Biology—Hans Knöll Institute Jena, Jena, Germany
11 Centre de Biologie Intégrative (CBI), CNRS, UPS, Toulouse, France
12 Division of Applied Bioinformatics, German Cancer Research Center (DKFZ), Heidelberg, Germany
13 Department of Physics and Astronomy, Bologna University, Bologna, Italy
14 German Cancer Consortium (DKTK), Heidelberg, Germany
  *Corresponding author. Tel: +49-731-50045870; E-mail: daniel.mertens@uniklinik-ulm.de
  **Corresponding author. Tel: +49-6221-5451450; E-mail: karsten.rippe@dkfz.de
  aThese authors contributed equally to this work
  bThese authors contributed equally to this work as senior authors
  †Present address: Center for Digital Health and Charité—Universitätsmedizin Berlin, Berlin, Germany
  ‡Present address: Institute for Biology, Systems Biology of Gene Regulation, Humboldt-Universität zu Berlin, Berlin, Germany
  §Present address: Wellcome Trust Sanger Institute, Cambridge, UK
  ¶Present address: Department of Computational Biology, University of Lausanne (UNIL), Lausanne, Switzerland
  ††Present address: Klinik für Innere Medizin I, Universitätsklinikum des Saarlandes, Homburg, Germany

However, how these genetic changes drive the cellular and clinical pathophenotype of the disease is currently an open question (Zenz et al, 2010; Kipps et al, 2017). The complex molecular pathogenesis of CLL involves microenvironmental stimulation via aberrant signaling including the B-cell receptor (BCR), NF-κB, IL-4, and TLR pathways, among others (Abrisqueta et al, 2009; Zenz et al, 2010; Hallek, 2015; Stilgenbauer, 2015; Kipps et al, 2017). The relevance of BCR signaling in CLL is underlined by the clinical success of BCR signaling inhibitors like ibrutinib (Byrd et al, 2013; Burger et al, 2015) and idelalisib (Furman et al, 2014), and by the prognostic impact of somatic hypermutations and the gene usage of the immunoglobulin itself (Zenz et al, 2010; Duhren-von Minden et al, 2012). Remarkably, apart from a biased usage of the immunoglobulin genes and mutations in the BCR complex in a specific small subset of CLL patients, there are no recurrent genetic mutations within the components of the BCR signaling cascade. Rather, CLL cells display a massive global transcriptional deregulation that is affecting intracellular pathways and microenvironmental signaling toward cellular survival (Burger & Chiorazzi, 2013). Thus, it appears that a diverse set of genetic lesions conspires with epigenetic aberrations to drive cancerogenesis in a manner that is only partially understood. The relevance of deregulated epigenetic signaling for CLL is apparent from a number of findings. Epigenetic aberrations in a mouse model of CLL are among the earliest detectable modifications (Chen et al, 2009), and the loss of tumor suppression in 13q14.3 involves transcriptional deregulation by an epimutation (Mertens et al, 2006). Genome-wide DNA hypomethylation was already early recognized in CLL cells (Wahlfors et al, 1992; Lyko et al, 2004), and more recently, a strong correlation with transcriptional activity was observed (Kulis et al, 2015). The DNA methylation status is a surrogate marker for CLL patient subgroups that overexpress the ZAP70 kinase and the mutational status of the BCR-immunoglobulin genes that allow prognostic dichotomization of CLL into more or less aggressive cases (Cahill et al, 2013; Claus et al, 2014). The epigenetic subtypes of CLL defined by the DNA methylome may become important for patient stratification as they are of prognostic relevance (Queiros et al, 2015). These epigenetic subtypes are correlated with the two genetically defined subgroups of CLL that express a non-mutated or mutated immunoglobulin heavy-chain variable region gene (IGHV) and reflect the tumor cell of origin in an epigenetic continuum of B-cell development (Kulis et al, 2015; Oakes et al, 2016).

Here, we conducted a comprehensive characterization of the chromatin landscape in primary CLL cells. Our analysis revealed that the massive changes in the CLL-specific transcriptome can be linked to deregulated chromatin features and activity changes of a transcription factor (TF) network downstream of microenvironmental signaling cascades. Our comprehensive data set represents a rich resource for studying gene regulation and epigenomics in CLL. We exploited it to integrate chromatin features and TF binding with gene expression programs in CLL B cells and suggest molecular mechanisms for the aberrant survival of malignant CLL cells.

## Results

### Aberrant chromatin features identified in CLL

In order to characterize CLL chromatin modifications in correlation with transcriptional activity, we analyzed the chromatin landscape and the transcriptome of CD19[+] B cells from peripheral blood from 23 CLL patients and from 17 pools of non-malignant B cells (NBCs) of healthy donors (Figs 1 and EV1, Appendix Fig S1 and Table S1, Datasets EV1 and EV2). While a number of pathophysiological processes such as microenvironmental signaling occur in secondary lymphoid organs of CLL patients (Burger & Gribben, 2014), the comprehensive analysis of different epigenetic layers required the acquisition of sufficient numbers of CD19[+] B cells and was therefore conducted from peripheral blood. CLL patients were selected to assess the fundamental changes in the original, untreated, and non-evolved disease including both disease subtypes of IGHV mutated and non-mutated samples. NBC pools were from age-matched healthy donors. Based on the genome-wide DNA methylation profiles, CLL samples could be assigned to B-cell maturation stages as shown previously (Kulis et al, 2015; Oakes et al, 2016; Fig EV1A). These developmental changes of epigenetic signals were excluded here for the identification of differentially methylated regions (DMRs) between CLL cells and NBCs.

The different chromatin features we mapped are depicted at the transcription factor 4 (TCF4) locus as an example for a gene upregulated in CLL (Fig 1A). The readouts include DNA methylation by whole-genome bisulfite sequencing (WGBS), histone chromatin immunoprecipitation (ChIP-seq) of H3K4me1, H3K4me3, H3K9me3, H3K9ac, H3K27me3, H3K27ac, and H3K36me3, nucleosome occupancy from high-coverage MNase digestion followed by H3 ChIP-seq, and open chromatin sites identified by the assay for transposase-accessible chromatin (ATAC-seq). For selected samples, also ChIP-seq of EBF1 and CTCF was performed. In addition, RNA transcription was analyzed by strand-specific RNA-seq of long and short RNAs (Appendix Fig S1A). The added value of this comprehensive analysis is apparent from inspection of the TCF4 gene. The histone modifications predict downstream enhancers and intronic enhancers that become activated in CLL cells as judged from the enrichment of H3K4me1 and H3K27ac (Fig 1A). The predicted enhancer loci in this region were particularly extended (> 10 kb) and are therefore an example for so-called "super-enhancers" (SEs, see below; Whyte et al, 2013). In order to systematically evaluate histone modification changes, we annotated chromatin with a 12-state ChromHMM Hidden Markov model (Fig 1B). Chromatin states differed substantially between CLL samples and NBCs and showed transitions for repressive chromatin states 4, 5, and 6 (H3K9me3, H3K27me3) and potential enhancer states 1, 8, 9, and 11 (carrying H3K27ac and/or H3K4me1 while lacking H3K4me3; Fig 1B and C, Datasets EV4, EV6–EV7).

To link changes of chromatin features with TF binding, we identified accessible chromatin with ATAC-seq. The method detects TF binding by mapping open and bona fide active chromatin regions that are depleted of nucleosomes. The differentially accessible regions in CLL patients and NBCs comprised 38,072 loci of which ~ 24,400 loci were located at the transcription start site (TSS), regions of transcription, and active or repressed regions (Figs 1D and EV1E). Loss of ATAC signal in repressed regions points to a more heterochromatic conformation in CLL, while at active chromatin regions, it might indicate a reduced promoter/enhancer activity. The IGHV mutated vs. non-mutated CLL can be distinguished according to the ATAC-seq profiles (Rendeiro et al, 2016). However, only ~ 1% of the differential ATAC-seq peaks identified here between CLL and NBCs were related to the heterogeneity of IGHV

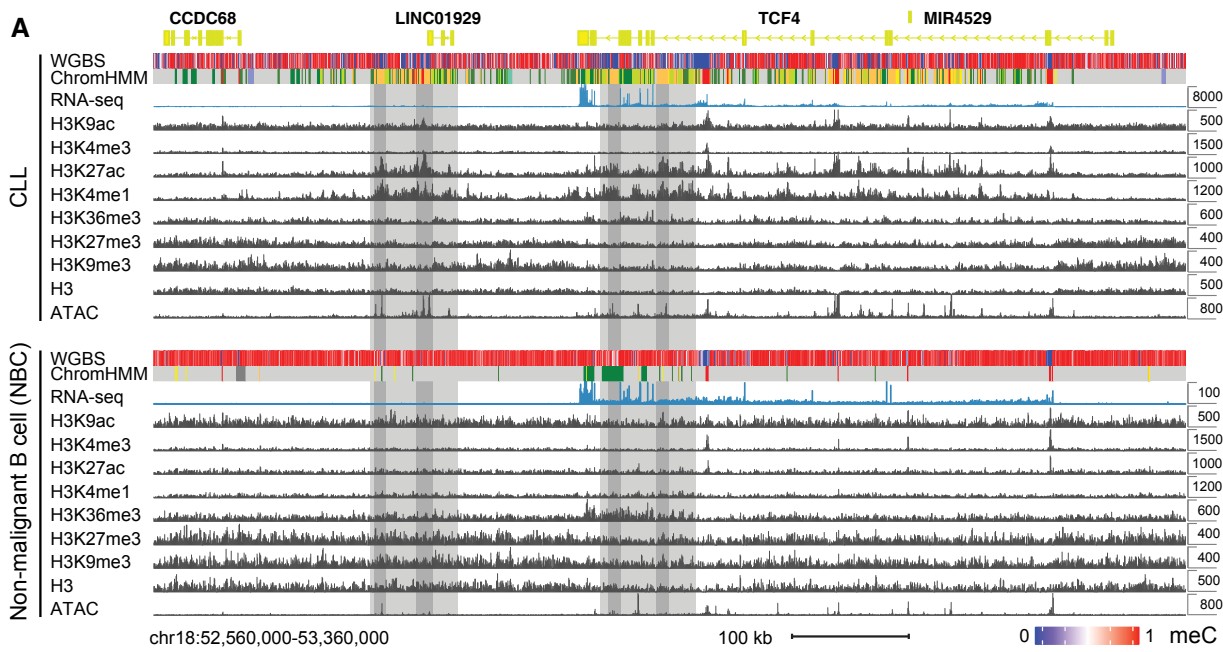

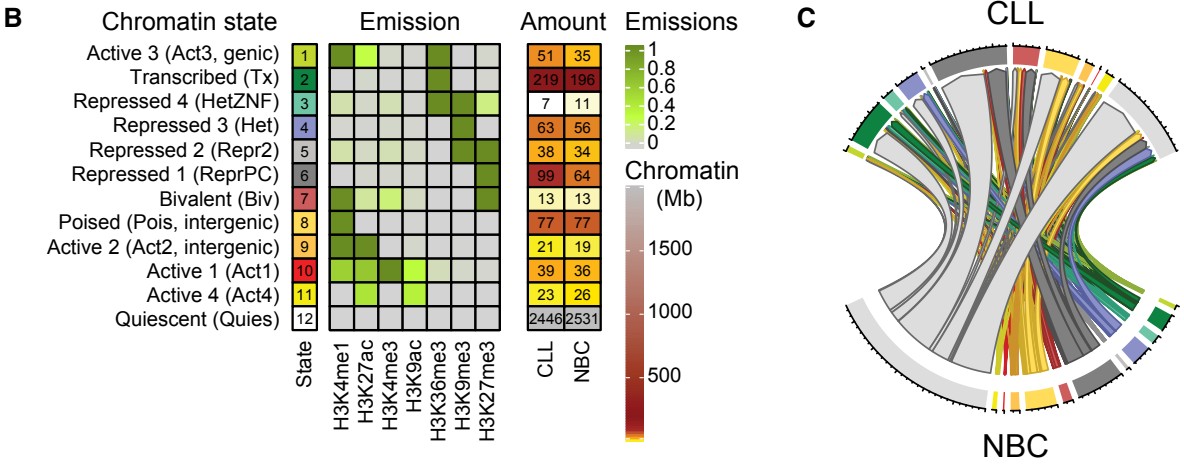

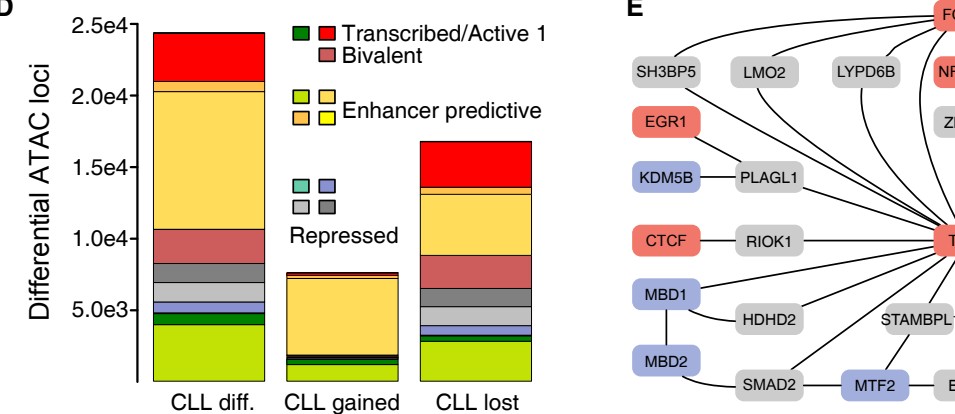

**Figure 1.**

**Figure 1. Chromatin feature annotation, open regions, and gene regulation.**

A  Chromatin features mapped here displayed differences between CLL patients and NBCs from healthy donors. As an example, the *TCF4* locus is shown for CLL1 and NBC donor H7 samples. The *TCF4* gene encodes for a transcription factor from the E protein family. Based on the increased H3K4me1, H3K27ac, and ATAC signal, two predicted enhancer loci were marked that became active in CLL. Note that the *y*-axis for RNA-seq is scaled differently for CLL (8,000) and NBCs (100) to visualize that the *TCF4* gene was not completely silenced but lowly expressed also in NBCs as evident also from the H3K36me3 mark. Light gray depicts active chromatin region and dark gray the confined enhancer locus coinciding with an open chromatin region. The chromatin state annotation is described in panel (B).

B  Chromatin segmentation of co-occurring histone modifications by ChromHMM yielding a model with 12 chromatin states. The indicated emission parameters for the contributions of individual histone marks and the average amount of each state (Mb) for CLL and NBC samples are given.

C  Chord diagram representation of genome-wide chromatin state changes between CLL and NBC. The amount of chromatin change is proportional to the size of the segments with each tick representing 4 Mb of chromatin. Color coding of chromatin states as in panel (B).

D  Distribution of ~ 24,400 annotated differentially accessible regions (ATAC-seq) in CLL compared to NBC samples ("CLL diff.") according to the chromatin state annotation. In total, 7,605 regions gained an ATAC-seq signal in CLL, while it was lost at 16,790 loci.

E  Part of the computed B-cell gene regulatory network showing TCF4 and its deregulated target genes as well as some of the adjacent nodes. The GRN was used to calculate the activity of regulators like TCF4 based on their target gene expression. Color code: TFs, red; target genes, gray; chromatin modifier, blue.

mutated and non-mutated IGHV CLL samples. This finding is illustrated for the *EBF1* TF locus in Fig EV1F.

The changes of the chromatin landscape were linked to the deregulated activity of TFs and chromatin modifiers in CLL according to the workflow depicted in Appendix Fig S1B. A B-cell-specific gene regulatory network (GRN) was constructed with the ARACNE framework (Alvarez *et al*, 2016). The GRN served as the backbone to integrate TFs and deregulated epigenetic signaling and comprised 2,804 regulators with a median value of 45 target genes. It was also used to compute the activity of TFs and chromatin modifiers from their target gene expression with our RNA-seq data. In total, 1,378 regulators displayed a differential activity between the CLL and NBC states ($P < 0.05$). As an example, TCF4 and selected deregulated target genes are shown in Fig 1E.

**Large repressive partially DNA methylated domains**

When comparing DNA methylation in CLL with NBC controls, we found a global hypomethylation in CLL as previously reported (Wahlfors *et al*, 1992; Lyko *et al*, 2004; Kulis *et al*, 2012). It was predominantly due to the formation of large partially methylated domains (PMDs; Figs 2A and EV2A, Appendix Fig S2A–D, Dataset EV3). Remarkably, the CLL DNA methylome contained a strikingly large genome fraction of ~ 50% PMDs in comparison with NBCs (< 1%; Fig 2A) with a significant overlap to PMDs previously identified for other tissues and cancer entities (Fig EV2B). The inter- and intra-sample variability of DNA methylation in CLL cells compared to NBC controls was high ($P = 0.005$, Wilcoxon rank-sum test, Appendix Fig S2C) and CLL cells harbored an increased fraction of intermediate DNA methylation within PMDs ($P = 1.6E-4$, Wilcoxon rank-sum test, Appendix Fig S2C). PMDs were enriched for lowly expressed and downregulated genes (Fig 2B, $P = 2E-48$, Fisher's exact test, Fig 2C), which can be rationalized by increased levels of repressive H3K9me3 and H3K27me3 histone marks (Fig 2D and Appendix Fig S2E). Regions with reduced transcriptional activity like the "B compartment" determined by Hi-C chromosome conformation capture (Fortin & Hansen, 2015) as well as lamina-associated domains (Guelen *et al*, 2008) were overrepresented in PMDs (Fig EV2C). In addition, active states (Appendix Fig S2F) and the H3K36me3 active transcription mark (Fig 2D) were depleted in PMDs, which were flanked by open chromatin (Fig EV2D). Our CTCF ChIP-seq data revealed an enrichment of CTCF binding at PMD boundaries, pointing to a potential function of CTCF to demarcate these regions and possibly limiting their further expansion

(Fig 2D). Of note, the majority (75%) of somatic mutations in CLLs were located in the PMDs identified here (Fig 2E), consistent with the increased mutation rates in heterochromatin regions (Schuster-Bockler & Lehner, 2012). On the level of local meC changes, we identified 8,671 differentially methylated regions (DMRs) of which 8,669 were hypomethylated in CLL (Fig EV2E). In total, 7,932 DMRs (91%) overlapped in CLL with predicted enhancer chromatin states (1, 8, 9, and 11; Appendix Fig S2G). Open chromatin regions within these DMRs as detected by ATAC-seq were enriched in binding motifs for NFATC1, EGR, and E2A (Fig EV2F).

**Global changes in the promoter-associated H3K4me3 histone mark**

Next, we investigated changes at promoters. When calculating correlation functions using raw mapped reads for H3K4me3, a small but significant extension of this signal by two additional modified nucleosomes was detected (~ 400 base pairs, $P < 0.002$, Figs 3A and B, and EV3A). We then extracted all extended H3K4me3 regions from peak calls with a $P$-value < 0.05 and a median change of 400 bp. This yielded 2,785 regions for CLL and only two regions for NBCs (Dataset EV5). The 2,785 CLL-specific extended regions spanned over annotated TSSs (Fig EV3A). We evaluated nucleosome occupancy using histone H3 maps obtained by MNase-ChIP-seq (Fig EV3B) and detected a significant gain of nucleosomes at 2,639 out of the 2,785 promoters (Figs 3C and EV3C, Dataset EV5). This change was accompanied by a loss of ATAC-seq signal in these regions (Fig 3D). A TF binding motif analysis yielded an enrichment for the bifunctional NFY regulator (Ceribelli *et al*, 2008), SP1 and ETS and KLF family motifs when using all unchanged H3K4me3 regions as background (Fig 3E). We thus conclude that extension of the H3K4me3 signal was linked to a gain of nucleosomes, which might also result in the masking of TF binding sites. Genes with an increased nucleosome occupancy around their TSS were found to be enriched for the BCR signaling pathway, such as *NFKB1* (Fig EV3C and D). From those 23 genes in the BCR signaling pathway, 18 (80%) were downregulated in CLL.

Using the active TSS state from our ChromHMM model, alternative TSSs were extracted that were specific for CLL. In total, 49 sites with alternative TSSs could be found that were independent of the extended H3K4me3 signal described above, e.g., from the *PITPNM2* gene (Appendix Fig S3A). In order to explore potential mechanisms of the CLL-specific activation of alternative promoters, we searched for enrichment of TF motifs compared to the promoter that was also

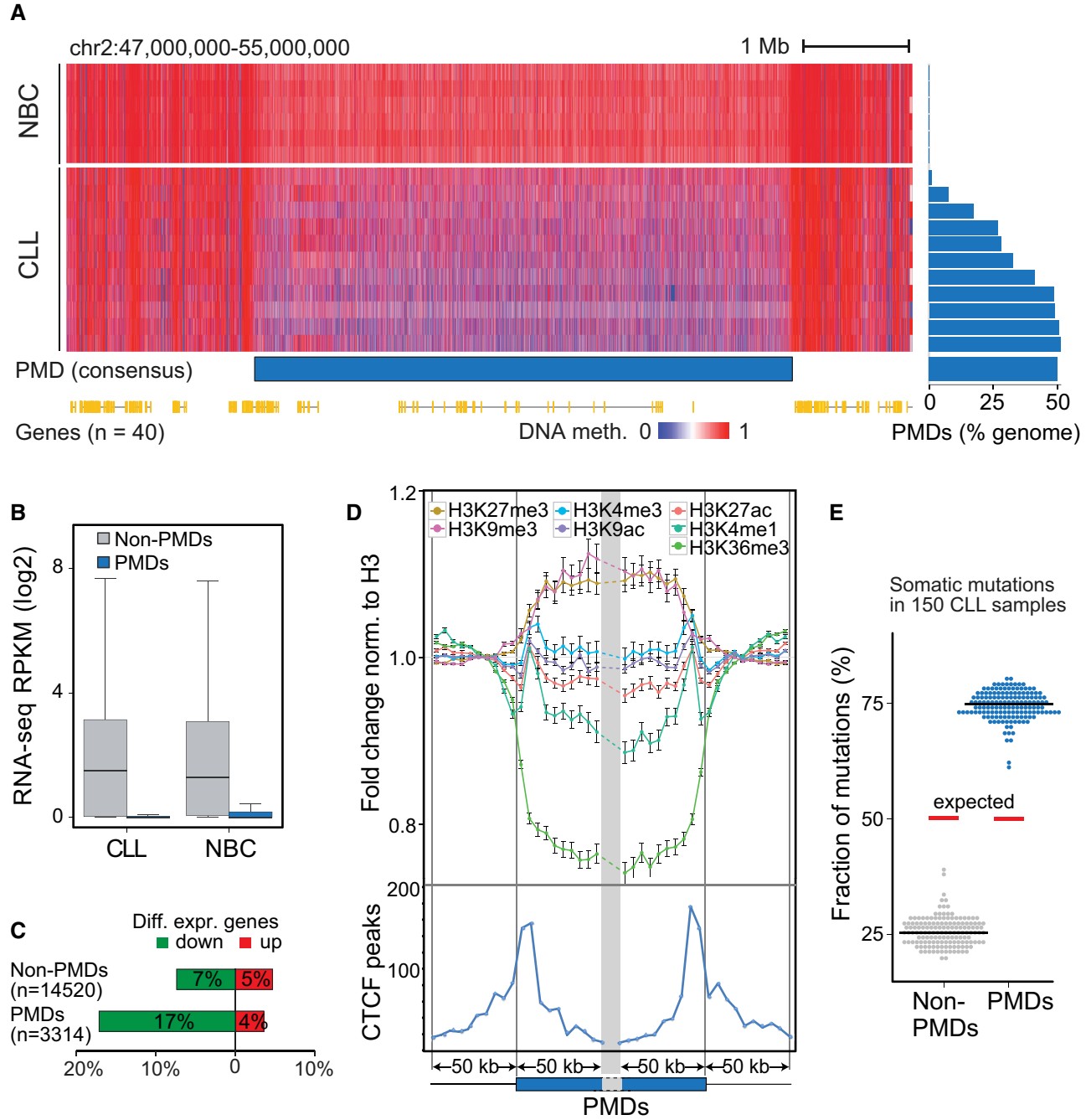

**Figure 2. Large partially methylated domains identified in CLL.**

A  Left, example of a large PMD on chromosome 2 derived from a consensus of CLL samples (*n* = 11). Right, genome-wide quantification of PMDs across CLL samples (*n* = 11) and NBCs (*n* = 6). The PMDs mapped with this set of 11 CLL samples were used for further analysis in figure panels (A–E) in combination with the RNA-seq and ChIP-seq analysis of the samples listed in Appendix Figure S1A. Red, methylated DNA; blue, unmethylated DNA.

B  Expression of genes located inside (blue) and outside (gray) the PMD regions. In the boxplot, maximum, third quartile, median, first quartile and minimum are indicated.

C  Fraction of differentially expressed genes inside or outside PMD regions. Up- and downregulated genes are shown in red and green, respectively.

D  Upper panel: Average signal of histone modification marks normalized to H3 and standard deviation in 5-kb windows around the ± 50 kb flanking regions of PMD boundaries. Normalized fold changes were calculated by dividing to the average signal flanking outside the PMD boundaries. Blue box, within PMDs; thin line, outside PMDs, norm.—normalized. Lower panel: Distribution of bound CTCF sites in CLL cells as determined by ChIP-seq (blue line) around the ± 50 kb flanking regions of PMD boundaries in 5-kb windows. The height of the curves gives the sum of the next nearest CTCF peak at the given distance to the PMD boundary.

E  Percentage of somatic mutations located inside (blue) or outside (gray) the PMD regions. Red line represents the expected ratio based on the genomic length of PMD and non-PMD regions. Mutation data were from Puente *et al* (2015).

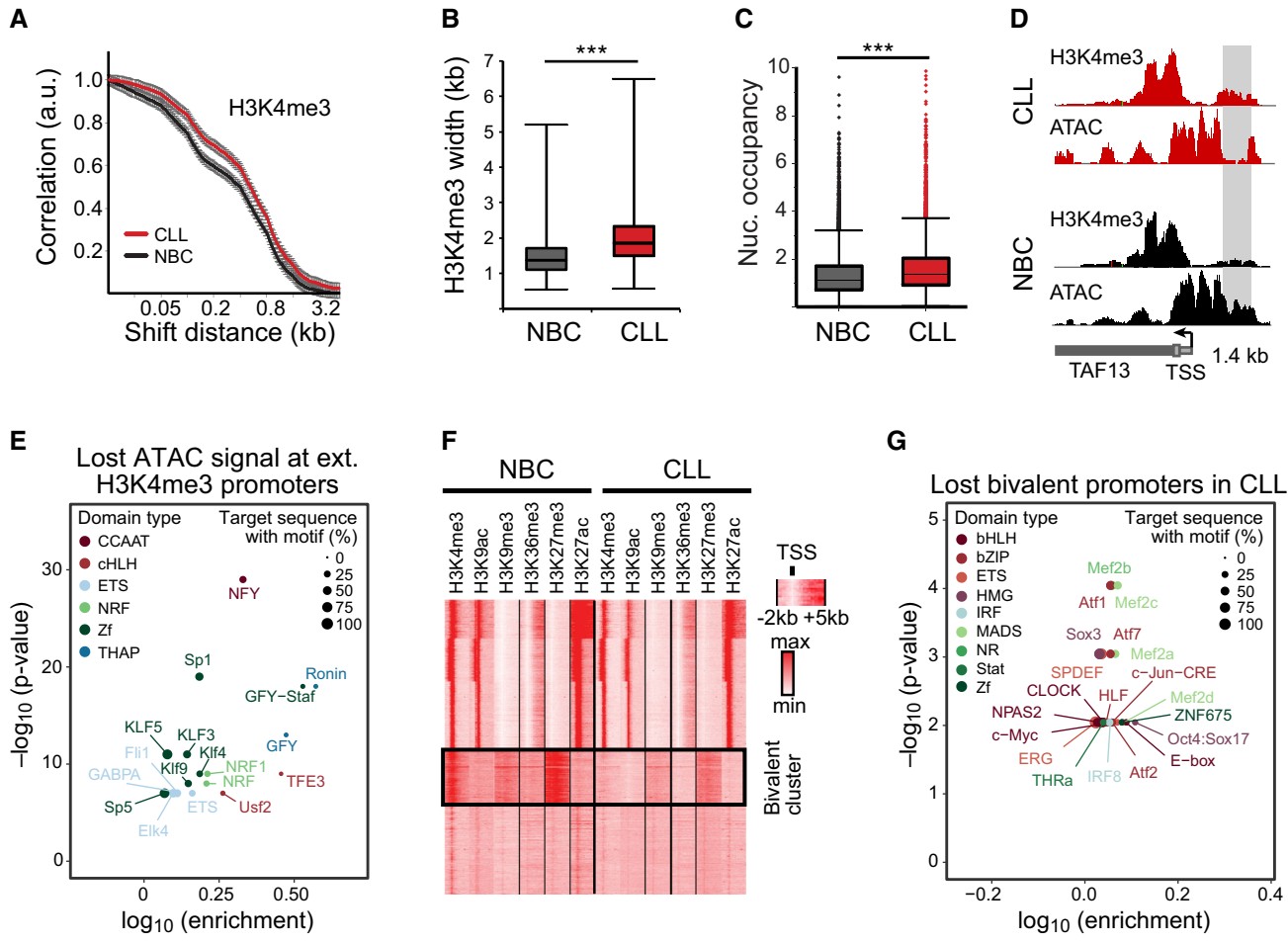

**Figure 3. H3K4me3 and nucleosome positioning changes at promoters.**

A   Correlation function of H3K4me3 ChIP-seq reads. A broadening of H3K4me3 domains in CLL by 1–2 nucleosomes was detected. The number of replicates analyzed was 11 (CLL) and 4 (NBC), respectively. Error bars represent the SEM.

B   H3K4me3 peak width distribution at common promoters in CLL and NBC controls. In the boxplot, maximum, third quartile, median, first quartile and minimum are indicated. The number of replicates analyzed was 11 (CLL) and 4 (NBC), respectively.

C   Distribution of nucleosome occupancy calculated from histone H3 ChIP-seq averaged over a 1,000-bp window within promoters. H3K4me3 regions displayed higher nucleosome density for CLL as compared to NBC samples. The boxplot representation and number of samples was the same as in panel (B).

D   Exemplary region at the TAF13 promoter showing higher H3K4me3 levels upstream of the TSS with lost ATAC signal (gray bar) as compared to the NBC control.

E   TF motif analysis of ATAC signal lost at CLL promoters with broadened H3K4me3 regions.

F   Heatmap of genome-wide histone modification patterns at promoters within −2 to 5 kb around the TSS (x-axis) for an NBC (H3) and a patient (CLL1) sample. Each line on the y-axis corresponds to one promoter. The clustering revealed one cluster with loss of H3K4me3 at bivalent promoters in CLL.

G   TF motif analysis at bivalent promoters that lost H3K4me3 in CLL.

active in NBC samples. The top four TFs identified at 67% of the CLL-specific alternative promoters were the lineage-associated TFs FOXA1, LEF1, POU3F1, and REPIN1 (Appendix Fig S3B).

An additional deregulated feature of the active H3K4me3 mark was linked to the repressive H3K27me3 modification. About 4,000 promoters that were bivalent in NBCs (simultaneous presence of H3K4me3 and H3K27me3) lost H3K4me3 in CLL (Fig 3F, ChromHMM state 7). Many of these genes were members of the HOX, FOX, SOX, and POU TF families that are functionally connected with developmental processes. No differential loss of H3K4me3 at bivalent promoters between IGVH mutated and unmutated CLL was detected (Fig EV3E). Only ~ 400 promoters were bivalent in CLL but not in NBCs (P < 0.001). Differences in apparent

promoter bivalency could also result from a heterogeneity of the individual marks. However, a truly bivalent state with H3K4me3 and H3K27me3 has only very low levels of transcription (Voigt et al, 2013). In contrast, a mixture of cells with fully active promoters (H3K4me3) and cells with silenced (H3K27me3) promoters should display transcriptional activity from the H3K4me3 fraction. For our set of promoters, we found that 67% of bivalent promoters were silent in NBCs (RPKM < 0.1) and thus represent *bona fide* bivalent promoters. We therefore propose that a large fraction of bivalent promoters that were poised for activation changed into a repressed state in CLL cells and retained only the H3K27me3 mark. In ~ 1,700 CLL promoters that lost H3K4me3, binding motifs of the MEF2 family of transcriptional activators were enriched (Fig 3G).

The MEF2 family TFs were suggested to regulate H3K4me3 (Pon & Marra, 2016; Di Giorgio *et al*, 2017) also in the context of H3K27me3 (Aziz *et al*, 2010). Furthermore, the H3K4-specific methylases KMT2B (MLL2) and KMT2D (MLL4), and to a lesser degree also KMT2A/C/E, were downregulated in CLL (Fig EV3F). Accordingly, we speculate that both the reduced activity of KMT2 enzymes and their impaired targeting by MEF2 TFs contribute to the loss of H3K4me3 at bivalent promoters in CLL.

**Annotation of potential enhancers**

The active ChromHMM states 1, 8, 9, and 11 comprised 238,820 regions (368 Mb) and represent loci that potentially contain enhancers. This set overlapped with the corresponding B- and T-cell chromatin states of the Roadmap consortium to 89.8 and 72.2%, respectively (Fig EV4A, Dataset EV9). The majority of previously reported enhancer loci from ENCODE and FANTOM projects were found in our state "Active 2" (state 9, H3K4me1, H3K27ac) and with a lower enrichment in states 1, 8,10, and 11 with the bivalent chromatin state 7 being the most enriched repressive state (Fig EV4B and C). Furthermore, binding sites of MEF2, IRF4, and FOXM1 identified by ENCODE were enriched in these states (Fig EV4B). The most prominent chromatin state transitions from NBCs to CLL cells were from "Quiescent" (state 12) to "Poised" (state 8, H3K4me1), from "Transcribed" (state 2, H3K36me3 in the body of transcribed genes) to "Active 3" (state 1, H3K4me1, H3K27ac, and H3K36me3, predictive for active intragenic enhancers) and from "Poised" to "Active 2" (state 9, H3K4me1 and H3K27ac, predictive of intergenic enhancers; Fig 4A and B). Genome-wide, the highest overlap with bidirectional expression of RNA was observed with "Active 1" (state 10, predictive for TSS) and "Active 2" (state 9, predictive for active intergenic enhancer; Fig 4A and C). Bidirectional RNA expression at enhancers could also discriminate between CLL and NBCs, identifying a set of disease-specific active enhancers (Fig 4D).

Next, super-enhancers were annotated by clusters of open chromatin regions co-occurring with large regions of H3K27ac domains in the range of 10 kb and above by using the ROSE software tool (Whyte *et al*, 2013; Dataset EV9). In total, 310 out of a union set of 1,289 SEs showed differential activity (false discovery rate (FDR) < 0.01). There were 219 up- and 91 downregulated SEs in CLL as compared to NBCs (Fig 4E). Exemplary predicted SEs that became active and upregulated the associated genes in CLL are shown for *TCF4, CREB3L2,* and *FMOD* (Figs 1A, and EV4D and E). The CLL-relevant genes that were up-/downregulated by proximal SEs are depicted in Fig 4E and F. The list includes the T-cell receptor alpha locus on chromosome 14 (*TRA locus*), the immune checkpoint receptor *CTLA4, FMOD* as a CLL tumor-associated antigen allowing for expansion of specific CD8[+] autologous T lymphocytes (Mayr *et al*, 2005), and *BCL2*, a highly successful therapeutic target (Kipps *et al*, 2017). For the latter, a SE-mediated regulation has been detected previously in non-Hodgkin lymphoma and CLL tissue (Meng *et al*, 2014). Interestingly, SE activity nearby the cyclin-dependent kinase inhibitor 1 gene *CDKN1A* (encoding p21) was reduced in CLL, which has been reported to be a p53-independent tumor suppressor (Abbas & Dutta, 2009). In addition, we found a number of TF genes being activated by proximal SEs in CLL that include *LEF1, ETV6,* and *NFATC1* in addition to *TCF4* and *CREB3L2* mentioned above.

**Impact of histone deacetylase inhibition on enhancer and promoter activities**

On a genome-wide scale, a large number of transitions from active chromatin states in CLL involved a loss of the H3K27ac modification (Fig 1D). Accordingly, we examined whether histone deacetylase (HDAC) activity differed between CLL and NBCs. A significant upregulation of several members from all HDAC classes (*HDAC1, HDAC3, HDAC8, HDAC9, SIRT5*) on the RNA level was apparent in CLL (Fig 4G). Direct measurements of the global activity of class I/II HDACs revealed an about fourfold higher activity in primary CLL cells compared to NBCs (Fig 4H). This activity could be efficiently inhibited with panobinostat. Upon treatment of primary B cells from

▶

---

**Figure 4. Differential enhancer activity in CLL and NBCs.**

A Overlap of active regions identified in CLL and NBCs by ChromHMM, ATAC peaks, or bidirectionally expressed RNA loci labeled as "Bidi". Venn diagram showing the total number of overlapping regions (not area-proportional).

B Chromatin transitions within differential active states between NBC and CLL. Heatmap representation of the amount of chromatin (log2 Mb + 1) transitioning from a particular state in NBC (rows) to CLL (columns). Transitions were considered for all recurrent active chromatin state regions (states 1, 8, 9, and 11) present in a minimum of three samples even if the consensus state was not an active state. Accordingly, the matrix includes transitions between non-active states at low frequencies.

C Chromatin states at bidirectionally transcribed predicted enhancers loci. All Bidi loci identified in NBC samples (*n* = 961) and CLL samples (*n* = 8,530) are shown. The Bidi loci show an enrichment of the states "Active 2 (predicted active enhancer)" and "Active 1 (predicted transcription start sites)".

D Clustering of samples via expression of bidirectionally regions that are differential between NBCs and CLL and quantified using DESeq2.

E Volcano plot of differential super-enhancers targeting known leukemia and cancer genes. Examples include SE loss at *CDKN1A, PI3KC2B,* and *KMT2B* (*MLL2*) and SE gain at *FMOD, CREB3L2, CTLA4, TCF4, LEF1,* and *BCL2*. Points represent non-differential SEs (gray) and differential SEs (FDR < 0.01) with fold change > 1 (orange).

F RNA expression changes of selected genes associated with differential SEs. Top: genes significantly (FDR < 0.05) upregulated by SEs in CLL. Bottom: genes significantly downregulated by SEs in CLL. In the boxplot, maximum, third quartile, median, first quartile and minimum are indicated. The number of replicates analyzed was 19 (CLL) and 7 (NBC), respectively.

G Comparison of normalized gene expression of histone deacetylases between CLL and NBCs. Histone deacetylases significantly upregulated in CLL are shown in bold. The boxplot representation and number of samples was the same as in panel (F).

H HDAC activity and its inhibition by panobinostat in B cells from CLL patients (red) in comparison with healthy donors (gray). Error bars indicate standard deviation measured in four biological replicates.

I Genome browser view of H3K27ac tracks (in gray) at exemplary genes for NBCs and CLL cells 24 h after mock and after panobinostat treatment. At genes such as *CDKN1A* (cell cycle control) and *KLF13*, reduced H3K27ac signal in CLL was increased upon HDAC inhibition to the level found in NBCs. *WNT11* is shown as an example of a *de novo* gain of an active enhancer due to treatment with panobinostat.

J Heatmap displaying changes in H3K27ac read occupancy in CLL upon panobinostat treatment for 24 h. A general gain of H3K27ac in enhancers upon panobinostat treatment was observed.

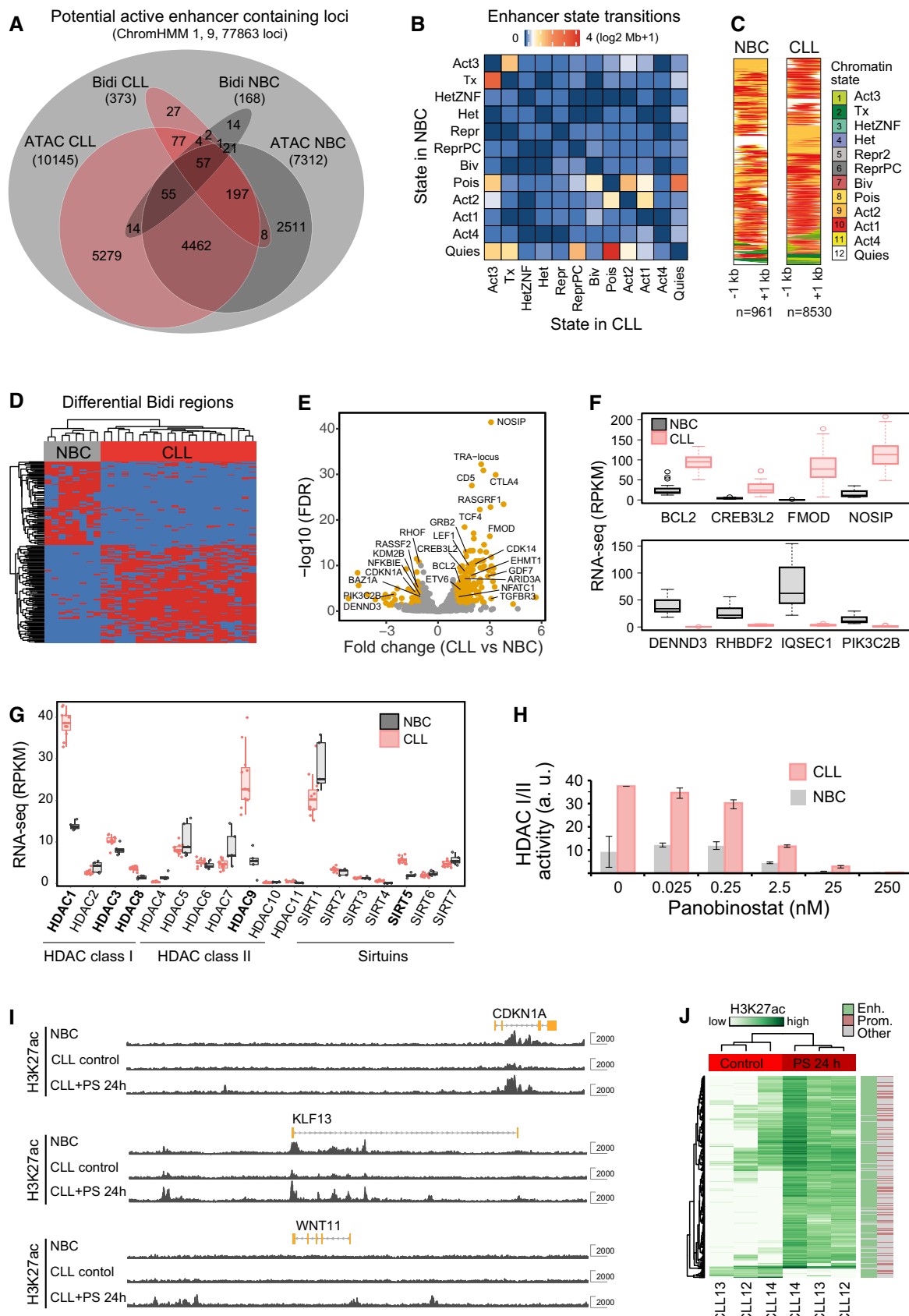

Figure 4.

CLL patients and healthy donors *in vitro* for 2 and 24 h according to the scheme depicted in Appendix Fig S4A, H3K9ac and H3K27ac acetylation levels increased as expected (Appendix Fig S4B). The panobinostat treatment was substantially more toxic to CLL cells than to NBCs (Appendix Fig S4C and D) and induced changes of gene expression and histone modifications (Fig 4I and J, Appendix Fig S4E). Deregulated H3K27ac patterns could be reverted in part by treatment with the HDAC inhibitor panobinostat and affected both promoters and enhancers (Fig 4J). Gene expression levels of 17% of all nascent transcripts were significantly up- or downregulated (Appendix Fig S4E). After 2 h of panobinostat treatment, only chromatin modifying processes were affected (e.g., upregulation of histone acetylation). At the 24-h time point, transcription of genes changed that were associated with RNA metabolism, possibly compensating the HDAC inhibition-related gene expression changes. Notably, a number of genes involved in BCR signaling and apoptosis were affected. These changes included reactivation of *BCL2L11, CDKN1A, PIK3CB1, NFKBIA, MEF2C,* and *IRF3* in CLL, while *BCL2, ZAP70, LEF1, ETS1,* and *RUNX1* were repressed (Dataset EV11). The complex gene expression response after panobinostat treatment can be rationalized by considering that pan-HDAC inhibitors like panobinostat not only induce histone hyperacetylation, but also affect posttranslational acetylation of non-histone proteins including TFs. For example, it has been shown that acetylation changes the activity of NF-κB (Chen *et al*, 2001), enhances DNA binding and transactivation of MEF2C (Ma *et al*, 2005), and decreases DNA binding and transcription activity of SP1/3 (Braun *et al*, 2001; Waby *et al*, 2010).

## Differential occupancy of TF binding sites at enhancers

We identified TF binding motifs in ATAC-seq peak regions (median size 327 bp) for different potential enhancer loci and excluded a ± 1 kb region around promoters. The annotations included differential ATAC signals (Dataset EV10) at ChromHMM states 1, 8, 9, and 11 (Fig 5A and C), states 1 and 9, which both carry a strong H3K27ac signal (Appendix Fig S5A), and states 8 and 11 for potential poised or weak enhancers (Appendix Fig S5B). In addition, the consensus ATAC peaks were intersected with DMRs for states 1, 8, 9, and 11 (Fig EV2F) as well as differential H3K27ac regions either genome-wide (excluding promoters; Appendix Fig S5C) or for SEs called as described above (Appendix Fig S5D). These different

approaches showed a large overlap of the TF motifs retrieved. Motifs recognized by E protein (including TCF4), TCF7/LEF, NFAT, EGR, and Forkhead TF families were gained in CLL, while motifs that were lost in CLL were mostly from the EBF, ETS, NF-κB, and JUN/FOS (AP-1) families (Appendix Table S2). PAX5, which has been recently identified within CLL subgroups as an essential super-enhancer factor for CLL cell survival (Ott *et al*, 2018), was no top hit in our differential TF motif analysis of SEs in CLL vs. NBCs. For the largest set of potential enhancer loci (ChromHMM states 1, 8, 9, 11), NFAT and E2A (E protein family) motifs were identified for sites that gained ATAC signal in CLL (Fig 5A). In addition, a total of 279 potential enhancer loci simultaneously exhibited significant changes of their H3K27ac signal at DMRs. For 256 (92%) of these sites, H3K27ac increased while meC was lost, suggesting that meC loss was correlated with enhancer activation. As an example, NFAT binding motifs are displayed, which revealed both a decrease in DNA methylation and an enrichment of the H3K27ac mark in CLL cells (Fig 5B). This finding points toward activation of this TF in CLL cells as shown previously (Oakes *et al*, 2016) in dependence of DNA methylation and H3K27ac. Next, we performed a corresponding TF motif analysis within regions with lost ATAC-seq signal in CLL at predicted enhancer loci (ChromHMM states 1, 8, 9, 11). The most frequently lost motifs were those of EBF/EBF1 and CTCF (Fig 5C). The differential gain or loss of binding of selected TFs was further analyzed by ATAC-seq footprinting at the genomic location of the motif sequences from Fig 5A and C as shown for E2A, EBF, and CTCF (Fig 5D). While the fine-scale pattern also reflected the sequence preference of the Tn5 transposase, the depth of the footprint and the differential accessibility around the aligned motifs confirmed the gain/loss of occupancy in CLL precisely at the target sequence.

The differential CTCF occupancy as inferred from ATAC-seq was corroborated by a CTCF ChIP-seq analysis and revealed clear differences between CLL cells and NBCs (Fig EV5A). Using DiffBind to extract differentially occupied regions from our CTCF ChIP-seq data, we found that CTCF binding was lost in CLL cells at 5,964 sites and gained at 441 sites (Fig EV5B). Of these lost sites, 93% overlapped with peaks from the ENCODE data set of the B-lymphocyte cell line GM12875 (GEO GSM749670), and with 55 and 47% of the gained and lost CTCF sites, respectively, determined from our ATAC-seq analysis. CTCF was lost in CLL cells mostly at chromatin states predictive for enhancers (Fig EV5C). In addition, while CTCF

---

**Figure 5. ATAC-seq and TF binding motif analysis of enhancers.**

A Plot of the most enriched TF binding motifs in regions that showed gained ATAC-seq signal at ChromHMM states 1, 8, 9, and 11 (predictive for enhancers). Color coding represents different TF classes. The size of the spots is proportional to the percentage of target sequences with a given motif.

B H3K27ac (left) and DNA methylation (right) at NFAT binding sites. CLL cells showed both an H3K27ac enrichment and DNA hypomethylation at NFAT target sites, suggesting a higher activity of TFs from the NFAT family in CLL.

C Same as panel (A) but for lost ATAC-seq signal.

D ATAC footprints for E2A, EBF, and CTCF motifs from the Homer analysis. The E2A motif footprint (binding site of E protein family TFs like TCF4) displayed an increased binding signal in CLL, while sites with the EBF and CTCF motif lost the ATAC signal.

E Expression of the genes nearest to target enhancers with constitutively bound ("stable") CTCF vs. enhancers that lost CTCF in CLL. Loss of CTCF binding correlated with reduced gene expression.

F Correlation matrix of simultaneously open regions computed from the scATAC-seq data. For all loci, the pair-wise correlation coefficients were calculated and plotted to visualize how different loci were wired to each other. As an example, the *TCF4* locus on chromosome 1 is depicted.

G Enhancer–promoter rewiring at the *NFKB2* locus. Top: Clustering of NBC and CLL samples according to gene expression of *NFKB2, PSD* and *FBXL15*. NBC samples were in the left cluster with high NFKB2 expression, which was reduced in the right cluster containing the CLL samples. Bottom: A switch of interactions between the *NFKB2* promoter and two different enhancers in CLL (red line) vs. NBCs (gray line) was observed. Based on the CTCF ChIP-seq analysis, both intronic enhancers at the *NFKB2* and the *FBXL15* gene show constitutively bound CTCF nearby, even though the targets of the two enhancers were switched.

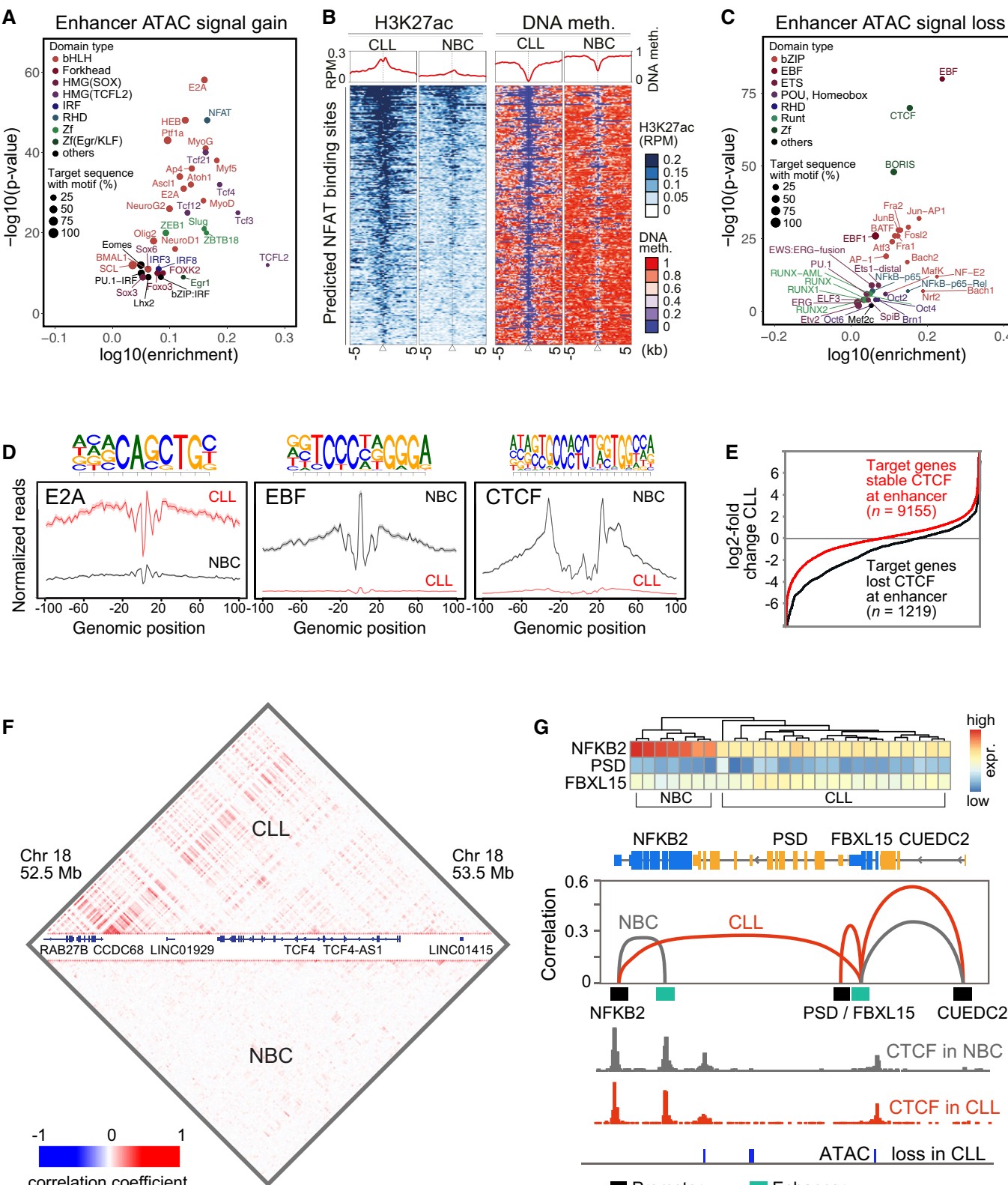

**Figure 5.**

demarcated PMDs as shown above, the PMD boundaries did not colocalize with TAD boundaries (Fig EV5D). Furthermore, loss of CTCF did not occur at TAD boundaries, which were rather occupied by CTCF in both malignant cells and NBCs (Fig EV5E). However, loss of CTCF was linked to the downregulation of the nearest gene in CLL (Fig 5E).

For the further analysis, we compiled lists of "active enhancers in CLL" and "active enhancers in NBCs". They were derived from the consensus of individual ATAC-seq peak regions flanked by active chromatin states 1 and 9 (Dataset EV9). The two lists comprised 10,145 loci in CLL and 7,312 in NBCs of which 4,771 were shared between the two groups after merging. The enhancers defined in this manner showed a large overlap of 85 ± 8% (average and standard deviation for all patient samples) with those identified from a corresponding analysis based on ATAC-seq peaks flanked by H3K27ac peaks instead of using the ChromHMM states. This finding is to be expected since states 1 and 9 carry a strong H3K27ac signal that is obtained from the peak called H3K27ac ChIP-seq used as input for ChromHMM.

### Mapping promoter–enhancer interactions from single-cell ATAC-seq data

Next, we identified accessible chromatin loci in single cells via single-cell ATAC-seq (scATAC-seq), which can resolve the heterogeneity between individual cells (Appendix Fig S6A–C). We analyzed 494 malignant cells from three CLL patients and 343 NBCs from two healthy donors. Distinct signatures of co-occurring open loci were found across different cells with a high overall overlap of open chromatin regions identified by bulk ATAC-seq and scATAC-seq. The heterogeneity of TF binding between single cells was assessed from the integration frequency of ATAC at known TF binding motifs (Appendix Fig S6B). Several motifs like NRF1, NFYA/B, and JUN/FOS (AP-1) identified above from the analysis of deregulated CLL chromatin features at promoters and enhancers displayed also the highest heterogeneous accessibility between individual cells in CLL.

Single-cell ATAC-seq can also reveal co-regulation of genomic loci by correlating concomitant accessibility of loci in heterogeneous cell populations. Based on the scATAC-seq data, we computed correlation matrices that contain information about pairs of genomic regions that are simultaneously open in the same cell (Fig 5F). Correlations between two regions can arise from spatial contacts between them or from other types of co-regulation, e.g., binding of a common regulator to both loci in the same cell. Using the annotated "active enhancers in CLL/NBC" lists (Dataset EV9), about 800 accessible enhancers per cell were detected on an average. With a threshold derived from the correlation coefficients obtained in a permuted data set (Appendix Fig S6D and F), we then identified enhancer–promoter pairs within 100-kb windows (Dataset EV10). We found that 68% of these pairs were also listed as spatial contacts in the 4D genome database (Teng *et al*, 2015), suggesting that 2/3 of the promoter–enhancer pairs involve physical contacts. In total, 3,955 promoter–enhancer pairs were identified, with most promoters being connected to one enhancer at mean and median distances of 32 and 20 kb (CLL) and 23 and 10 kb (NBCs), respectively (Appendix Fig S6E and G). A total of 1,612 pairs were specific for CLL, 1,932 pairs were specific for NBCs, and 411 pairs were present in both CLL and NBCs. In addition, a set of 205 rewired promoters was defined as promoters that do not share any enhancer between CLL and NBCs but were correlated with at least one enhancer in each of the cell types. Within this set of rewired promoters, 70 and 49 were linked to genes down- and upregulated in CLL, respectively. These genes were enriched for pathways involved in signal

transduction, apoptosis, and differentiation. We then evaluated the properties of enhancers that were assigned to different target genes in CLL and NBCs. Based on our CTCF ChIP-seq analysis, we found that at 90% of the rewired enhancers CTCF was stably bound in both cell types (Fig EV5F). As an example, the *NFKB2* locus that is downregulated in CLL is shown (Fig 5G). In CLL, the regular enhancer of *NFKB2* was replaced with an enhancer that was connected to the *FXB15* gene in NBCs. The *FXB15* enhancer had a reduced activity as inferred from the loss of ATAC signal in CLL, potentially causing downregulation of *NFKB2*. Based on these observations, we propose that rewiring between enhancers and their target promoters might be one mechanism that leads to deregulated gene expression in CLL. This process appears to be independent of CTCF binding changes. Rather, it is consistent with the previous conclusion that CTCF insulates larger contact domains, while specific enhancer–promoter interactions might involve other factors like YY1 (Weintraub *et al*, 2017).

### Assignment of TFs to aberrant CLL chromatin features

Aberrant regulatory epigenetic signals were detected at 81% of the transcriptionally deregulated genes in CLL (Fig 6A, Datasets EV11 and EV12). Based on our TF motif analysis at promoters and enhancers, we selected the most relevant TFs that displayed differential binding with respect to two or more of our chromatin readouts as described in Materials and Methods (Appendix Table S2). These TFs were assigned to four different deregulated CLL chromatin features at promoters or enhancers (Fig 6B). A gene set enrichment analysis of targets of these TFs retrieved pathways that are highly relevant for the CLL pathophysiology (BCR, NF-κB, and MAPK signaling; Fig 6C, Appendix Fig S7). Relevant links in this context were, for example, from SYK (Benkisser-Petersen *et al*, 2016) to NFATC1, BCL2 (Roberts *et al*, 2016) to TCF4, and NOTCH1 (Fabbri *et al*, 2017; Ryan *et al*, 2017) to NF-κB. Using the connections provided by our gene regulatory network, specific chromatin modifiers were identified that had deregulated activities and could be involved in the observed changes of meC, H3K4me3, H3K27me3, and H3K27ac at promoters and enhancers (Appendix Fig S7). The resulting deregulated TF network thus rationalizes how epigenetic dysregulation in CLL could be linked to changes in TF activity. This TF network can be used to generate hypotheses on how these activity changes are driven by external signaling and how the chromatin aberrations feedback to signaling.

### Integrative gene regulatory enhancer network analysis

The pair-wise correlations between enhancers and promoters computed from the scATAC-seq data was combined with the B-cell GRN to derive a "gene regulatory enhancer containing network" (GREN; Datasets EV13 and EV14). From the complete GREN, a CLL-specific GREN was extracted that includes the connected network derived from the CLL-specific TF list, their target genes, as well as linked chromatin modifiers that affect the aberrant chromatin features in CLL. As an additional requirement for this selected network part, all included factors needed to be deregulated in their activity/expression between CLL and NBCs. A part of this network that is centered around TCF4 and EBF1 is shown in Fig 6D. To validate the ATAC-based motif analysis for EBF1 (Fig 5C) and

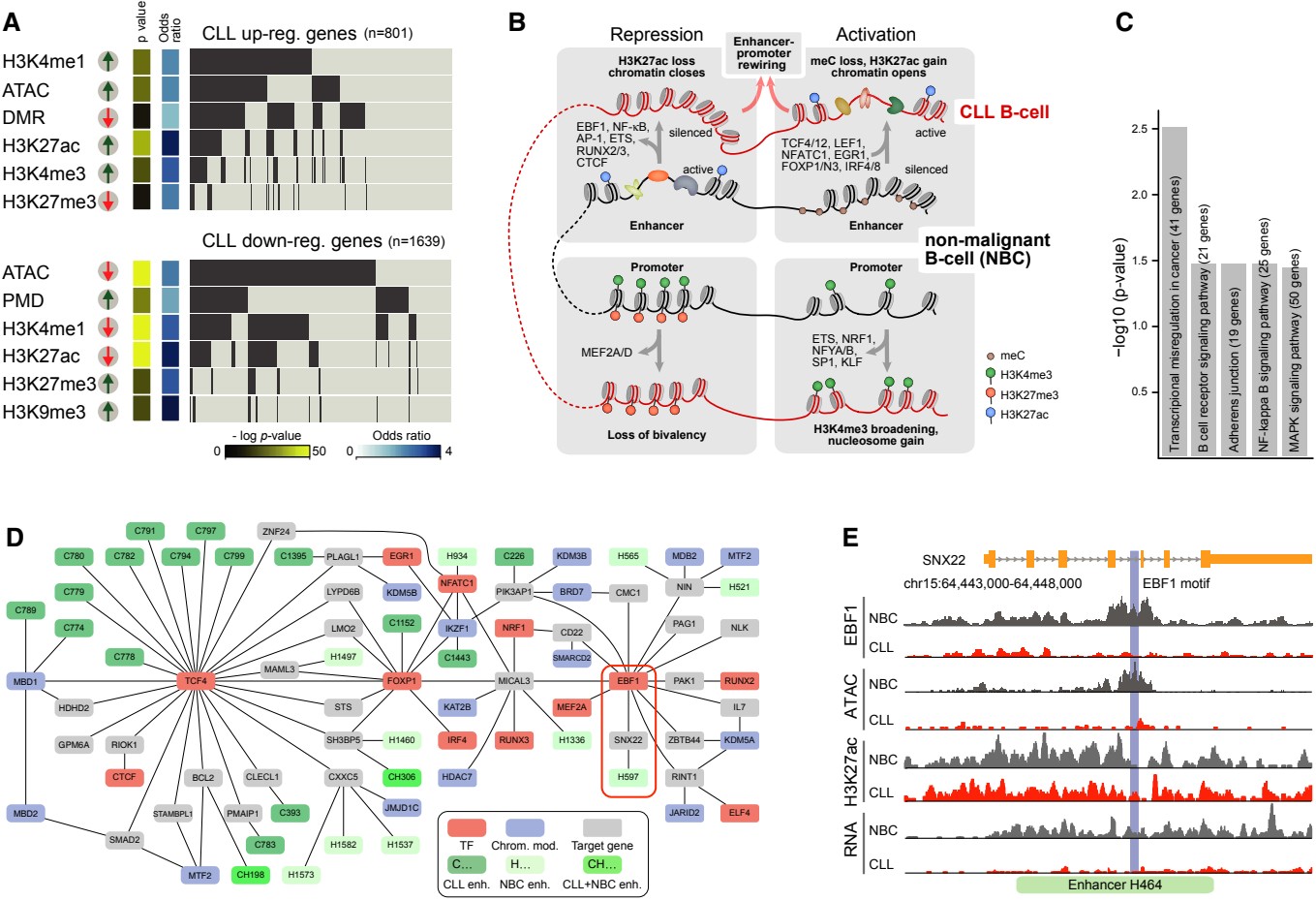

**Figure 6. Integration of chromatin state changes, TF binding, and gene expression.**

A   Relation of chromatin states and gene expression changes. For 81% of differentially expressed genes, a change in a regulatory chromatin feature was observed.

B   Scheme of chromatin feature changes and associated TFs identified in this study (Appendix Table S2).

C   Pathway analysis of identified core TFs and their target genes from our B-cell GRN. Significantly enriched gene sets (*P* < 0.05) relevant to CLL pathophysiology were retrieved (BCR signaling, NF-κB signaling, and MAPK signaling).

D   CLL GREN. TFs identified here, associated chromatin modifiers, and differentially expressed target genes were integrated into a network. A part that includes TCF4 and EBF1 is shown. The different TSSs of the *TCF4* gene (Fig 1A) were combined into a single gene target for the corresponding SE. Several enhancers of EBF1 target genes were active in NBCs (H402, H441, H464, H1000) but became silenced in CLL. As an example, H464 and SNX22 are highlighted by a red rectangle. These enhancers represent candidate enhancers for involving EBF1 binding. Color code: active enhancers, dark, light, and medium green for CLL only ("C"), only in NBCs ("H"), and both cell types ("CH"), respectively; TFs, red; target genes, gray; chromatin modifier, blue.

E   Analysis of regulation of *SNX22* by the intronic H464 enhancer shown in panel (B). This enhancer lost its ATAC signal at the predicted EBF1 binding site. EBF1 ChIP-seq analysis validated that EBF1 is indeed lost at this site. Although H3K27ac at the locus was only slightly reduced, transcription of *SNX22* was largely reduced. These findings are consistent with a mechanism where EBF1 binding drives gene expression of *SNX22* by binding to H464.

predictions from our CLL GREN, we conducted a ChIP-seq analysis of EBF1. Binding of EBF1 was lost at 826 sites and gained at 173 sites in CLL, separating malignant and NBC samples and showing a very good overlap with the ATAC analysis (Appendix Fig S8A–D). Interestingly, genes that in CLL lost EBF1 binding at their enhancers could be clustered into the functional groups of immune response and cell activation, two pathways of central relevance in the pathophysiology of CLL (Appendix Fig S8E). We then applied our CLL-specific GREN to dissect regulation by EBF1, which can act as both an activator and repressor (Boller *et al*, 2018). Several enhancers of EBF1 target genes were predicted to be active only in NBCs (H402, H441, H464, H1000) and silenced in CLL. These enhancers represent

candidates for being driven by EBF1, which is lost in CLL. As an example, H464 and its target gene *SNX22* are highlighted (Fig 6D). We confirmed the predicted loss of EBF1 at the intronic H464 enhancer by ChIP-seq and determined a global CLL-specific EBF1 binding pattern (Fig 6E, Appendix Fig S8 and Table S3). Although H3K27ac at the *SNX22* locus was only slightly reduced, transcription of the *SNX22* gene was largely decreased (average log fold changes −5.1) as evident from the RNA-seq tracks (Fig 6E). Furthermore, enhancers H1000 (MICAL3 gene) and H441 (NIN gene) also showed the expected loss of EBF1 binding, while little change was observed for H402 (NIN gene, not shown; Appendix Fig S8F). Thus, for three out of four enhancers in our CLL-specific gene regulatory network, a

predicted loss of EBF1 binding in CLL was experimentally validated by the EBF1 ChIP-seq.

## Discussion

Here, we dissected the aberrant epigenetic circuitry in primary CLL cells vs. NBCs by conducting a comprehensive analysis of chromatin modifications. Deregulated chromatin features were uncovered that included both localized changes at regulatory enhancer and promoter elements as well as large-scale chromatin reorganization on the Mb scale and above. The most striking example for the latter was the massive accumulation of large PMDs. These domains correlated with repressive chromatin and gene silencing as in other cancers (Berman *et al*, 2011; Hon *et al*, 2012; Hovestadt *et al*, 2014; Schultz *et al*, 2015). They mapped to lamina-associated domains (Guelen *et al*, 2008) and Hi-C B compartments (Fortin & Hansen, 2015; Fig EV2C), reflecting an enhanced heterochromatic chromatin organization in CLL. Interestingly, boundaries of PMDs were enriched for occupied CTCF sites, which might indicate a role of CTCF to limit the expansion of PMDs. This view is supported by the finding that CTCF binding sites can act as bifurcation points for differential DNA methylation and that CTCF binding can be modulated by DNA demethylation activity (Teif *et al*, 2014; Wiehle *et al*, 2019). Our integrative analysis of a large set of readouts revealed altered chromatin features at promoter/enhancer elements for 81% of the differentially regulated genes in CLL cells (Fig 6A). The underlying CLL-specific TF network that we derived from aberrant chromatin features displayed strong connectivity with epigenetic readers and writers such as SIN3 and the NuRD and SWI/SNF chromatin remodeling complexes that were deregulated in CLL (Appendix Fig S7, Dataset EV14). While little is known about the function of remodeling complexes in CLL, the NuRD and SWI/SNF remodelers play an important role in hematopoiesis and differentiation and have been implicated in oncogenesis and cancer progression in numerous other entities (Lai & Wade, 2011; Kadoch & Crabtree, 2015; Prasad *et al*, 2015). Furthermore, it is noted that the ability of these complexes to translocate nucleosomes might be crucial to modulate chromatin accessibility at enhancers and involve their targeting by histone modifications like H3K4me1 (Local *et al*, 2018). In line with these considerations, we observe striking changes in nucleosome positioning and occupancy at B-cell-specific genes in CLL. In our promoter analysis, changed nucleosome occupancy at active promoters resulted in a loss of accessible chromatin and additionally modified histones. This change seems somewhat different from the broadened H3K4me3 regions observed previously in other entities (Chen *et al*, 2015) as the extension observed in CLL comprised only 1–2 nucleosomes. Interestingly, extension of the H3K4me3 mark via gain of nucleosomes at transcriptional start sites and loss in ATAC-seq signal was linked to an enrichment of TF binding motifs for NFYA, SP1, and KLF proteins. This finding may point to a loss in TF binding due to the higher nucleosome density that also could contribute to the extension of the H3K4me3 signal. A bivalent signature switch at CLL promoters characterized by a loss of the active H3K4me3 mark points to a reduced developmental plasticity of CLL cells. According to our analysis, loss of H3K4me3 is predicted to occur via MEF2 TFs (Aziz *et al*, 2010; Di Giorgio *et al*, 2017) and reduced KMT2 activity. Furthermore, we find a number

of additional links in our core TF network to modifiers of H3K4me3 that included, for example, KMT2E, KDM5A, SETD7 (Appendix Fig S7, Dataset EV14).

Extensive changes of enhancer activity were observed that allowed us to clearly discriminate between CLL and NBCs, with a pronounced loss of enhancer activity in CLL. It could be related to an elevated HDAC activity measured in CLL samples and was reverted in part by panobinostat treatment that led to increased H3K27ac at enhancers. Our scATAC-seq analysis revealed correlations between *bona fide* active sites and led us to propose that enhancer-mediated changes of transcription activity occurred via rewiring to different target promoters in CLL cells. A similar data set does not exist for CLL, as only direct physical contact have been mapped at low resolution by Hi-C (Beekman *et al*, 2018) or for selected interactions by 4C (Ott *et al*, 2018).

From our chromatin feature maps, we derived a highly connected CLL-specific network centered around the TFs targeting 17 central binding motifs and an enrichment of BCR signaling genes (Fig 6B and C, Appendix Fig S7 and Table S2, Dataset EV14). These central motifs include gained motifs in CLL for NFAT, TCF4, and LEF1 and lost motifs for EBF1 and AP-1, which have similarly been reported in two other studies (Oakes *et al*, 2016; Beekman *et al*, 2018). The integrated view on the interplay of TFs, chromatin modifiers, and associated target genes derived here provides a rich resource to generate hypotheses for novel molecular links to the CLL pathophenotype. For example, the regulation of BCL2 via TCF4 predicted in our GREN (Appendix Fig S7) is of special interest since it is the target of venetoclax, a highly effective treatment option in CLL (Roberts *et al*, 2016). The importance of enhancers as drivers of deregulated gene expression in CLL is another notable finding emerging from our work. Since a large number of enhancers change their activity state and/or their target genes, a promoter centric view will be insufficient to rationalize the global reprogramming of the CLL transcriptome. Finally, given the complex interlinked structure between a large number of TFs, chromatin modifiers, and their target genes, it appears quite clear that simple linear models will not be able to fully grasp essential parts of the pathomechanism. Accordingly, we envision that the approach of developing integrated gene regulatory enhancer containing networks will prove to be valuable for therapy response prediction and patient stratification for CLL. Furthermore, the comprehensive data sets created here and in another study (Beekman *et al*, 2018) provide a rich resource for CLL researchers. It will largely facilitate studies that involve clinically relevant disease phenotypes with deregulated molecular mechanisms, which are reflected by the multitude of aberrant features present in the CLL epigenome.

## Materials and Methods

### Patient samples

All CLL patients analyzed in this study gave written informed consent (ethics committee approval) and were characterized in the diagnostics department of the University Hospital Ulm (Appendix Table S1). The median age of the CLL patients was 62 years (range, 47–79). NBC control samples were enriched from peripheral blood of in total 65 age-matched donors (median,

57 years, range, 51–70) that were collected in 17 sample pools. For all samples from CLL patients and from NBC donors, CD19-positive B cells were purified, yielding a CD19-positive fraction of 98 ± 2% in NBC donor and 99 ± 1% in CLL patient samples. A fraction of 79% of CLL patients had favorable genetic aberrations [del(13q)] as sole abnormality associated with increased survival times, 21% had normal karyotype. 61% of the patients were diagnosed with hyper-mutated IGHV associated with good prognosis. Due to the high numbers of B cells required for multiple sequencing, we selected asymptomatic CLL patients who had been approached with the watch-and-wait strategy. None of the patients analyzed had previous treatment of the CLL disease with high numbers of CLL cells unaffected by the potential impact of prior therapy in the peripheral blood with a median leukocyte count of $101.4 \times 10^9$/l (range, 37.5–$280.6 \times 10^9$/l).

### Whole-genome bisulfite sequencing

DNA was extracted with the AllPrep DNA/RNA Mini Kit (Qiagen), spiked with unmethylated lambda DNA (Promega) and sonicated to achieve an average size of 250–300 bp. After bisulfite conversion using the EZ DNA Methylation Kit (Zymo), strand-specific WGBS-seq libraries were prepared as described previously (Kretzmer et al, 2015). For each library, three lanes with 100 bp paired-end reads were sequenced on the Illumina HiSeq 2000 system.

### ChIP-seq of histone modifications and transcription factors

Patient-derived cells were cross-linked with 1% methanol-free formaldehyde for 10 min. After quenching with glycine, cells were washed three times with PBS and the cell pellet was frozen in liquid nitrogen. For analysis, the cell pellet was thawed and treated with four units MNase per $1 \times 10^6$ cells for 15 min. MNase was stopped with 10× covaris buffer, and the chromatin was sheared for an additional 15 min with the S2 covaris device. The soluble chromatin was then recovered and subjected to a background-minimizing pre-clearing step with an unspecific IgG antibody. For each ChIP assay, an equivalent of $3 \times 10^6$ cells was used. After the IP, chromatin was digested with RNaseA and proteinase K. From the purified DNA sequencing, libraries were generated with the NEBNext Ultra library preparation kit (NEB). ChIP-seq of both CTCF and EBF1 was done with the SimpleChIP-seq kit from Cell Signaling Technology according to the manufacturer's instructions. After purification of the DNA, libraries were cloned with the NEBNext Ultra II library kit (NEB). The antibodies used for ChIP-seq were H3K4me1 (Abcam ab8895), H3K4me3 (Abcam ab8580), H3K9ac (Active Motif 39137), H3K9me3 (Abcam ab8898), H3K27ac (Abcam ab4729), H3K27me3 (Abcam ab6002), H3K36me3 (Abcam ab9050), H3 (Abcam ab1791), CTCF (Active Motif 61311), and EBF1 (Sigma SAB2501166) and are listed in Appendix Table S4.

### ATAC-seq

Nuclei from formaldehyde-fixed or viable frozen (one out of seven NBC samples) cells were isolated with Nuclei EZ lysis buffer (Sigma-Aldrich), washed once in ATAC-seq lysis buffer (10 mM Tris–HCl, pH 7.4, 10 mM NaCl, 3 mM $MgCl_2$, and 0.1% IGEPAL CA-630) and incubated with Tn5 transposase (Illumina) for 30 min

at 37°C or viable frozen cells (three out of seven NBC samples) were directly incubated with Tn5 for 30 min at 37°C in the presence of 0.1% Igepal CA-630 (Sigma-Aldrich). After incubation, Tn5 was stopped with EDTA, for previously fixed samples, cross-links were reversed, and the DNA was purified with AMPure beads (Beckman Coulter). Barcodes were added by PCR, and the libraries were purified again with AMPure beads. The scATAC-seq sequencing libraries were generated with the Fluidigm C1 system as described previously (Buenrostro et al, 2015). In brief, viably frozen, primary B cells were loaded on a 96-chamber C1 flow cell and captured cells were analyzed by microscopy to exclude doublets. After harvesting the pre-amplified single-cell ATAC-seq libraries, dual external barcodes were added by PCR and all libraries from one C1 flow cell were pooled. Libraries were 50-bp paired-end sequenced on Illumina HiSeq 2000 (all scATAC-seq samples) or 4000 systems.

### RNA-seq

For RNA-seq, short and long RNA fractions were isolated with the miRNeasy Mini Kit (Qiagen), RNA was digested by DNase I (Promega) for 30 min at 37°C and depleted from rRNA with Ribo-Zero Gold, and finally, directional cDNA synthesis and sequencing library preparation were conducted according to the NEBNext Ultra Directional RNA Library Prep Kit (NEB). All libraries were 50-bp single-end sequenced on the HiSeq 2000 machine.

### Panobinostat treatment and HDAC activity assay

Primary CLL cells from three treatment-naïve patients and NBCs in a pool of 28 healthy donors were treated with 5 nM panobinostat (Biozol) ex vivo. Cells were sampled before as well as 2 and 24 h after initiation of treatment. Western blots were conducted to evaluate changes in the abundance of H3K9ac and H3K27ac. The samples were subject to ChIP-seq experiments for seven histone modifications and the H3 control and also to RNA-seq as described below. Malignant and NBCs of each of four individual CLL patients or healthy donors, respectively, were enriched for CD19-positive. A total of 10,000 cells were seeded in serum-free medium and incubated for 30 min with DMSO as mock control or different concentrations of panobinostat (25 pM–250 nM). HDAC-Glo I/II assay (Promega) was performed in technical triplicates according to manufacturer's protocol and measured with the Promega GloMax device.

### DNA methylation analysis

Whole-genome bisulfite sequencing data were processed with MethylCtools as previously described (Hovestadt et al, 2014). In addition to the default pipeline of MethylCtools, an updated version of BWA-MEM (v0.7.12-r1039) was used to align reads to the human reference genome, build hg19 (hs37d5) from the 1000 Human Genome project (Li & Durbin, 2010). DNA methylation levels (β-values) were calculated for more than 23 million CpG sites covered with at least five reads and supported to be a correct CpG by the sequencing data in individual samples (> 75% of reads; Hovestadt et al, 2014). The non-conversion rates were below 0.25% for all samples as determined from the spiked-in lambda phage genome DNA. Partially methylated domains were identified for each sample

using an approach described previously (Berman *et al*, 2011; Hovestadt *et al*, 2014). In brief, the genome was scanned with a 10-kb sliding window (R caTools package, v1.17.1) to retrieve windows with average methylation levels < 0.65. The resulting 10-kb windows were joined together if overlapping, and the merged regions over 100 kb were called as partially methylated domains. The consensus PMDs were generated as the union of PMDs present in at least half of the CLL samples analyzed ($\geq$ 6). Differentially methylated regions between CLL and NBCs were identified using the R Bioconductor (Gentleman *et al*, 2004) package, DSS (v2.10.0; Feng *et al*, 2014), and $P < 0.05$, the minimum length of 200 base pairs and the differential methylation threshold of 0.3. The DMRs and PMDs overlapping with the UCSC "gap" table, comprising centromeres, telomeres or regions with no sequence information, were discarded from further analysis. Normal B-cell programming might account for some of the methylation changes from the comparison of CLL to NBCs (Oakes *et al*, 2016). To account for this, DMRs were filtered out if they show "similar" methylation changes (< 0.2 β-value) also in the comparison of high-maturity memory B cells to naive B cells as described previously (Oakes *et al*, 2016). Methylation profiles were employed to assign CLL samples into three distinct CLL subtypes (low, intermediate, and high) programmed using a reference 450K methylation data from 185 CLL cases. Raw 450K data were processed using the Minfi Bioconductor package (v1.16.1; Aryee *et al*, 2014). Phylogenetic analysis was carried out as previously reported (Brocks *et al*, 2014; Oakes *et al*, 2016) on the 450K data of CLL cases and normal B-cell subsets together with the WGBS data of CLL cases from our study. As a validation, a support vector machine (SVM) model (Queiros *et al*, 2015) was adopted to classify CLL samples into three distinct subtypes, confirming our previous results. To exclude any label switching, genetic fingerprinting was performed to all the samples profiled by WGBS, ChIP-seq, ATAC-seq, and RNA-seq, validating the sample origin.

### ChIP-seq data analysis

After mapping with bowtie to the hg19 genome assembly with decoy regions (options –best -strata -v 1 -m 1), peaks were called with MACS for H3K4me1, H3K4me3, H3K9ac, and H3K27ac using H3 as a background control. For H3K9me3, H3K27me3, and H3K36me3, SICER was used to determine enriched regions with histone H3 serving again as a control. In order to computationally validate the quality of the ChIP-seq experiments, we clustered the different IP samples using their Pearson correlation. Additionally, quality scores suggested by the ENCODE project were used to assess the overall quality of each ChIP-seq data set (Dataset EV4). Peaks were then used for chromatin state analysis with ChromHMM as stated below. Topology and size distributions of modified chromatin regions were calculated with MCORE as described in Molitor *et al* (2017) with the modification that histone H3 was used as a background file for normalization. CTCF and EBF1 ChIP-seq sequencing libraries were mapped as described above, and peaks were called with MACS2 using standard parameters for narrow peak detection.

Differential histone modification and TF binding analysis from ChIP-seq data were performed using the peaks called for each sample and the corresponding reads counts within the alignment files using DiffBind (Dataset EV8). Read counts were scaled to the control, and normalization was done relative to the number of reads within peak regions. Differential enhancers were called using edgeR with an FDR threshold of 0.01 within DiffBind. For the 24-h mock treatment and panobinostat samples, paired analysis was performed using DESeq2 with an FDR threshold of 0.01 within DiffBind. The panobinostat differential H3K27ac regions were annotated as promoter overlapping with respect to any TSS from Gencode v17 and annotated as enhancer overlapping with respect to the enhancers of combined CLL occurring in at least three untreated or two 24-h-treated samples. For both CTCF and EBF1, differential regions between CLL and NBCs were computed form a consensus peak list within DiffBind.

### Genome segmentation and chromatin annotation

The chromatin was segmented and annotated using a multivariate Hidden Markov model with the ChromHMM software (Ernst & Kellis, 2012). The model was trained using seven histone modification marks (H3K4me1, H3K4me3, H3K27ac, H3K27me3, H3K9ac, H3K9me3, and H3K36me3) across all samples where histone modification ChIP-seq data were of sufficient quality for all histone marks (samples H1, H3–5, H8, CLL1–3, CLL5–11). The model was learned by using the merged peak called with MACS or SICER from the ChIP-seq data as binarized input for ChromHMM and allowing for a maximum of 400 iterations. We generated and compared models with 7–50 states and decided to use 12 states (Datasets EV6 and EV7). This model captured all combinations of biologically interpretable states with median correlation of 0.72 to the 50-state model and displayed high concordance with the Roadmap 15-state model. For further analysis, the 12-state ChromHMM model was used to segment the genome of each individual sample (including the panobinostat-treated samples not used for generating the model).

### Nucleosome occupancy maps

Paired-end sequenced histone H3 ChIP-seq libraries from MNase-digested chromatin were used to determine nucleosome positions in malignant and NBCs. Reads were mapped with Bowtie as described above and processed using NucTools (Vainshtein *et al*, 2017) to generate genome-wide nucleosome occupancy landscapes, extract individual genomic regions, and calculate the nucleosome repeat length, as described previously (Teif *et al*, 2012, 2014; Vainshtein *et al*, 2017). Aggregate nucleosome occupancy profiles around genomic features were calculated using HOMER (Heinz *et al*, 2010) and NucTools (Vainshtein *et al*, 2017). The pathway enrichment analysis was performed with DAVID and EnrichR.

### ATAC-seq data analysis

Both scATAC-seq and bulk ATAC-seq were processed in a similar manner. Reads were trimmed using Trimmomatic version 0.36 (Bolger *et al*, 2014) and mapped to the human genome (hs37d5) using bowtie2 (Langmead *et al*, 2009) with the options "–very-sensitive -X 2000". After removing reads which in regions blacklisted by ENCODE (https://sites.google.com/site/anshulkundaje/projects/blacklists), duplicate reads were discarded using rmdup of SAMtools version 1.3 (Li *et al*, 2009) and reads mapping to the mitochondrial

genome were also removed. For further analysis and calculation of coverage tracks, all reads were shifted to be centered on the cutting position of the transposase and extended to a total of 29 bp to represent the region occupied by the transposase (Adey et al, 2010). For bulk ATAC-seq, peak calling was done with MACS2 version 2.1.1 (Zhang et al, 2008) with the parameters "–nomodel –shift -10 – extsize 28 –broad". Differential ATAC-seq signals were calculated with DiffBind (Ross-Innes et al, 2012), within peak regions called in at least four replicates (Dataset EV10.1). Fold change thresholds were determined as the intersections of three Gaussian functions fitted to the distribution of log2 fold changes from DiffBind (Fig EV1E). The FDR threshold used for the differential ATAC-seq signal analysis was set to the $x$-value of the inflection point of the number of differentially accessible regions as function of FDR threshold. The analysis was also conducted with only fixed frozen NBC samples to ensure that the results were not affected by differences between fixed and viably frozen control samples (Dataset EV10.2, used for TF motif analysis in Appendix Fig S5A and B).

For scATAC-seq, typically 40,000–100,000 unique integrations per cell were obtained. Cells with very few (less than 0.2-times the average) or very frequent (more than three standard deviations above the average) integrations were removed from the analysis. Variations in accessibility for different TF binding motifs were determined with chromVAR (Schep et al, 2017). Correlation coefficients for simultaneous insertions across all cells at promoters and enhancers were computed with the R-package RWire, which has been deposited at https://github.com/FabianErdel/RWire. Target promoters of enhancers within TAD-sized regions were identified based on the respective correlation values using a threshold (0.22) derived from the spurious correlations obtained after randomly permuting insertions (see Appendix Fig S6 for details). The resulting set of promoter–enhancer pairs was used for downstream analysis and integrated into the gene regulatory network.

### Enhancer annotation

Different sets of putative enhancers in CLL and NBCs were compiled that all excluded promoter regions as defined by a region of ± 1 kb around the TSS (RefSeq): (i) In order to identify TF motifs gained or lost in CLL within the ATAC peak regions, the differential ATAC signal was intersected with either all four potential enhancer containing ChromHMM states 1, 8, 9, and 11 (Fig 5A and C) or only with the predicted active enhancer states 1 and 9, which carry a strong H3K27ac signal (Appendix Fig S5A) or with the predicted poised/weak enhancers from states 8 and 11 (Appendix Fig S5B). (ii) Active enhancer lists were based on either ATAC signal or bidirectional transcription (Bidi) as active marks. ATAC-seq peaks or sites of bidirectionally expressed RNA loci identified in both replicates of a given patient or NBC control sample were selected. A ± 1 kb region around the ATAC/Bidi signal center was intersected with the predicted intergenic and genic active enhancer states 1 and 9. The size of ± 1 kb was selected based on the average extension of key enhancer marks (H3K27ac, H3K4me1, p300, Bidi; Chen et al, 2018). Subsequently, overlapping regions were merged. Consensus active enhancer lists for CLL and NBCs were generated from individual patient or NBC control lists by selecting regions which were identified in at least three of the available CLL patient sample and one (ATAC) or two (Bidi) of the NBC controls, for which both the

ChromHMM annotation and ATAC-seq/Bidi data were available. (iii) SEs were predicted with the ROSE tool (Whyte et al, 2013) by using the H3K27ac read signal within the overlap region of ATAC and H3K27ac peaks. The Gencode v17 genes were defined as a background gene model for filtering transcription start sites within ROSE. Differential analysis of H3K27ac at enhancers was performed using DiffBind on the enhancer regions occurring in at least three samples with the original H3K27ac data for the individual samples. Differential loci were called using DiffBind with an FDR threshold of 0.01. Functional assignment of differential SEs was done with GREAT (McLean et al, 2010) with modified parameters of 10 kb upstream, 5 kb downstream, and 2 Mb distal which annotates cis-regulatory regions.

### TF motif analysis

To identify TF motifs in regions of interest with gained or lost occupancy, we used the HOMER package with appropriate background controls for each individual data set (Heinz et al, 2010). All TF motif analysis was conducted within the ATAC peak regions. For the TF footprinting analysis, loci with a certain motif were retrieved with HOMER's "annotatePeaks.pl" and "known motifs" analysis. Coverage tracks ± 100 bp around the center of the motifs were generated with ngs.plot (Shen et al, 2014) using combined reads from all CLL patients or NBC controls collapsed to the center of the transposase binding event. In Appendix Table S2, the TF sequence-binding motifs and selected factors are listed that were associated with at least one additional aberrant chromatin feature in CLL in addition to chromatin accessibility changes as detected by ATAC-seq. These chromatin features included histone H3K4me3 broadening, nucleosome gain or loss of the bivalent H3K4me3-H3K27me3 mark at promoters, or, at enhancers, differential DNA methylation or H3K27ac changes. Only TFs were included that showed a significant differential protein activity (or gene expression for network target genes) as computed from our B-cell-specific gene regulatory network (Fig 1E). The latter served to select specific TFs from TF families that recognize the same DNA sequence motif. For example, a number of the ETS family TFs recognize the same sequence motif (Wei et al, 2010) but many of these TFs did not display differential activity as computed from their target gene expression data.

### RNA-seq data analysis

Sequence reads were mapped to the human genome (hs37d5,) using STAR version 2.3.0 (Dobin et al, 2013) with the parameters: –genome-Load LoadAndRemove –alignIntronMax 500000 –alignMatesGapMax 500000 –outSAMunmapped Within –outFilterMultimapNmax 1 – outFilterMismatchNmax 3 –outFilterMismatchNoverLmax 0.3 – sjdbOverhang 50 –chimSegmentMin 15 –chimScoreMin 1 –chimScoreJunctionNonGTAG 0 –chimJunctionOverhangMin 15. RPKM and TPM values were computed for the long and the total RNA. The mapped reads were quantified using HTSeq-count from the HTSeq framework version 0.6.0 using the intersection_nonempty option in a stranded fashion according to the gencode17 annotation (Anders et al, 2015). Normalization and differential expression analysis were performed using DESeq2 (Love et al, 2014; Dataset EV11). Bidirectional expression was computed within a 1-kb sliding window. Loci

were selected that had at an expression level of at least 20% of the reads of the higher transcribed strand produced from the other strand and displaying a bimodal shape of the differently directed reads. Regions overlapping transcription start sites or exons on both strands were excluded.

## Gene regulatory network construction and activity calculation

A human B-cell regulatory network was constructed from 264 publicly available samples (Basso *et al*, 2010) including normal B cells, B-cell lymphomas, and cell lines by using the ARACNe-AP algorithm (Lachmann *et al*, 2016) with default parameters (MI $P = 10^{-8}$, 100 bootstraps, and permutation seed = 1). Regulatory proteins were taken from a previously compiled list of 5,927 proteins (TFs, transcriptional co-factors, and signaling pathway-related genes etc.) based on gene ontology annotations (Alvarez *et al*, 2016). From this list, 3,862 proteins were present in the B-cell network, which additionally contained 12,119 target genes and 214,405 interactions. The average and median of target genes per regulator were 56 and 45, respectively. The activity values of regulators were calculated based on our RNA-seq data set by using the VIPER algorithm (Alvarez *et al*, 2016). For this, the raw RNA-seq counts were normalized by variance-stabilizing transformation using DESeq2 (Love *et al*, 2014). Activities for 2,804 of the 3,862 regulators could be computed. To identify regulators which showed a significantly different activity between the CLL samples and the NBC samples, a two-sided Student's *t*-test was performed and the *P*-values were adjusted for multiple testing (Benjamini & Hochberg, 1995) yielding 1,378 differentially active regulators with an adjusted *P*-value < 0.05. Differential expression of regulators target genes of the regulators was determined with DESeq2. Gene set enrichment analysis was performed using the R-package "gProfileR" (Reimand *et al*, 2016) with the B-cell network as universe to identify significantly enriched KEGG pathways (Kanehisa & Goto, 2000) for the identified core TFs and their target genes. The network was visualized using Cytoscape version 3.5.1 (Shannon *et al*, 2003). Regulators with a differential activity change between the CLL and the NBC samples (*P* < 0.05) and all differential expressed non-regulators (*P* < 0.01) with a log fold change of $-1.7 < \text{LFC} > 1.7$ were selected for further analysis. To identify aberrant chromatin modifier in CLL vs. NBCs, the list of deregulated regulators/genes was intersected with the EpiFactors database (Medvedeva *et al*, 2015). This list was then reduced to those genes that were linked to the deregulated chromatin features identified in our study (meC loss, H3K27me3 gain, H3K27ac loss/gain, and H3K4me3 loss/gain).

All enhancer–promoter pairs with a correlation of 0.22 or higher in the scATAC-seq analysis (Appendix Fig S6D and F) were used from the CLL and the NBC samples. To remove duplicates, overlapping regions were put together and the gene promoters were mapped to gene symbols. The enhancer–promoter pairs were grouped into CLL only (C), NBCs only (H), and found in both groups (CH). For all these cases, the enhancers were numbered consecutively (C1-C1208 in case of the CLL-specific enhancers) and the information about the genomic positions was added as a node attribute. For all the genes in the B-cell network, the enhancer–gene interactions were integrated. For the CLL-specific network, the interactions between the core TFs (Appendix Table S2), their deregulated target genes, and the selected chromatin modifier as well as the enhancers were extracted from the B-cell network (Datasets EV13 and EV14). Exemplarily, the part of the network around TCF4 and EBF1 is shown in Fig 6D. In Appendix Fig S7, the linked part around the core TFs is shown, and here, some chromatin modifiers were grouped into complexes.

# Data availability

The data and computer code produced in this study are available from the following sources:

All original sequencing data have been deposited at European Genome-phenome Archive under the accession number EGAS00001002518 (http://www.ebi.ac.uk/ega/).

Processed WGBS, ChIP-seq, (sc)ATAC-seq, and RNA-seq data are available under GSE113336 (https://www.ncbi.nlm.nih.gov/geo/query/acc.cgi?acc=GSE113336) at Gene Expression Omnibus as full bigBed/BigWig files.

Metadata from the comprehensive analyses are provided as Datasets EV1–EV14 with the manuscript. They are described in Appendix Table S6, and additional data are available via the web page http://www.cancerepisys.org/data/cancerepisys_data/.

Software used for the data analysis for the different sequencing readouts is listed in Appendix Table S7. Custom analysis scripts and tools are available from Github at https://github.com/CancerEpiSys/Mallm-et-al-processing-scripts, and the R-package RWire has been deposited at https://github.com/FabianErdel/RWire.

**Expanded View** for this article is available online.

# Acknowledgements

We thank the members of our laboratories and the CancerEpiSys and PRECiSe consortia for help and discussion. The main funding of this project was from the CancerEpiSys (0316049) and PRECiSe (031L0076) grants of the German Federal Ministry of Education and Research (BMBF) in the e:Bio program. The DKFZ-Heidelberg Center for Personalized Oncology (DKFZ-HIPO) provided additional sequencing (HIPO_H038). Further support was via BMBF project CancerTelSys (01ZX1302B, 01ZX1602B) to KR and RK, Wellcome Trust grant 200733/Z/16/Z to VBT and DFG SFB1074 (subprojects B1 and B2) to SS and DM, and the NCT 3.0 project ENHANCE to PL and KR. We thank Andre Holzer, Martina Seiffert, and Yevhen Vainshtein for help and discussions; Gideon Zipprich, Jules Kerssemakers, and Nick Kepper for data management; and the DKFZ Genomics and Proteomics Core Facility for sequencing services.

# Author contributions

Acquisition of patient samples: SJK, SSt, and DM; Acquisition of data: J-PM, SSc, and SJK; Analysis of data: J-PM, MI, NI, LCK, JMM, VBT, AMP, SG, FE, DT, SDK, MZ, and KR; Development of integrative model: J-PM, MI, NI, DM, and KR; Drafting of manuscript: J-PM, MI, NI, DM, and KR; Supervision and manuscript reviewing: BB, RK, DR, SSt, PL, MV, MZ, DM, and KR; Study design and coordination: DM and KR.

# Conflict of interest

The authors declare that they have no conflict of interest.

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
