## [Review Process File · Molecular Systems Biology]

Linking aberrant chromatin features in chronic lymphocytic leukemia to transcription factor networks

Jan-Philipp Mallm, Murat Iskar, Naveed Ishaque, Lara C. Klett, Sabrina J. Kugler, Jose M. Muino, Vladimir B. Teif, Alexandra M. Poos, Sebastian Großmann, Fabian Erdel, Daniele Tavernari, Sandra D. Koser, Sabrina Schumacher, Benedikt Brors, Rainer König, Daniel Remondini, Martin Vingron, Stephan Stilgenbauer, Peter Lichter, Marc Zapatka, Daniel Mertens, Karsten Rippe.

Review timeline:

Submission date:	28 th March 2018
Editorial Decision:	27 th April 2018
Revision received:	17 th October 2018
Editorial Decision:	19 th December 2018
Revision received:	2 nd April 2019
Accepted:	17 th April 2019

Editor: Maria Polychronidou

Transaction Report:

1st Editorial Decision

27th April 2018

Thank you again for submitting your work to Molecular Systems Biology. We have now heard back from the three referees who agreed to evaluate your study. As you will see below, the reviewers appreciate the extensive amount of data generated in this work and mention that the study is a potentially relevant contribution to the field. They raise however a series of concerns, which we would ask you to address in a major revision of the manuscript.

I think that there is no reason to repeat the points listed below, since they are rather clear. Overall the reviewers mention that additional analyses need to be included in to better support the main conclusions and strengthen the study. They provide several constructive suggestions in this regard. Please let me know in case you would like to discuss further any of the comments of the reviewers.

REFeree REPORTS.

Reviewer #1:

In the paper of Mallm et al., the epigenomic analysis of B-cells of 19 CLL patients and 9 healthy donors is described. The authors use a variety of different assays to analyze DNA methylation, chromatin states, nucleosome positioning, accessibility and the transcriptome to provide new insights into CLL pathogenesis. One of the main findings is a derived gene regulatory network (GRN) constructed from (public) RNA-seq data. The manuscript is rich in data that can be further explored by the community and most computational analysis is on par with the standards in the field. However, the current manuscript presents some weaknesses. First, the relation and the agreement between the RNA-seq derived network and the TF motif analysis is not clear. Are some of the TF motifs associated with alternative promoters or are CLL-specific enhancers also major

regulators in the GRN? Second, many of the key findings are based on computational analysis for which no validation (e.g. ChIP-seq or 4C-seq) was provided. The manuscript needs to be strengthened by validation of the findings.

Major comments

The GRN is based on information theoretic analysis (ARACNE-AP) of published RNA-seq data and supplemented with current knowledge of known regulators in the field. How well does the derived model predict these known regulators? E.g. a ROC curve would provide a measure of the "quality" of this network.

The abstract states that 'based on DNA-binding motifs, TFs were integrated into the GRN.' How was this done? How do the TF motifs relate to the VIPER-derived activity scores? It seems that the authors can use differential motif analysis to constrain their GRN to direct targets that are differential between healthy and CLL donors. This is however not clear from the methods section, and some figures (e.g. Fig 1E) appear to be based solely on the RNA-seq data.

The authors state that loss of DNA methylation and the accompanied loss of RNA-expression could be explained by accumulation of H3K9me3 and H3K27me3. This is not apparent from Fig 2D and Fig S2G. In Fig S2G, the accumulation of H3K27me3 seems much higher in the non-PMDs.

Moreover, the PRC/repressed states do not cover 50% of the genome, unlike the PMDs. Can the authors show a negative correlation between DNA methylation and occupancy of these marks?

Also, H3K36me3 is generally associated with gene transcription. In Fig 1A for CLL TCF4 seems high expressed according to the RNA-seq results, but H3K36me3 is low, while it is high in non-malignant cells that have no signal for TCF4 in the RNA-seq track.

The authors identify novel TFs related to alternative promoter usage (FOXA1, LEF1, POUF3 OR REPIN1), differential acetylation (MEF2a/d, Klf4) and CLL-specific enhancers (e.g. NFkB, TCF4, ATF) but no validation was provided. ChIP-seq in healthy vs CLL for one or two of these TFs would significantly advance the current manuscript. Similarly, the promoter-enhancer rewiring inferred from single cell co-accessibility is interesting. These claims would be substantially more valuable when validated in differential setting for some loci of interest using a chromatin conformation assay like 4C-seq. Along the same lines: on page 10: "distinct patterns of spatially co-regulated activity hubs ... affect central pathways in the CLL pathogenesis". If the differential occupancy of CTCF and promoter-enhancer rewiring is so essential to CLL pathogenesis, computational derivation from ATAC-seq is insufficient. Again, such claims would be substantially more valuable when CTCF occupancy was measured using ChIP-seq and chromatin interactions for some loci were analyzed in a differential setting using 4C-seq.

In Fig 7 the authors essentially show that HDACi's are doing what they are expected to do (reduce HDAC activity and increase H3 acetylation) at the molecular level. What happens at the cell biological to these treated cells? Are they differentiating (as would be expected from the loss of the CLL heterochromatinized state claimed by the authors) or do these cells have altered viability? Is this specific for the CLLs, or are also the non-malignant cells responding?

Minor comments

Please clarify Fig 1B. The legend refers to log change and $-\log_{10}$ p-values but how are they represented in the figure?

How many of the BCR signaling components are occupied by the FOXA1, LEF1, POUF3 OR REPIN1? I.e. what is the consensus between the functional network analysis and the motif analysis? Fig 1C: Can the authors include the 9 normal B cells for comparison?

Fig 3B: Do we see H3K4me3 broadening or stronger nucleosome positioning? The H3K4me3 signal to the right seems to disappear in CLLs.

Fig 3C, D: If you repeat the analysis for the non-malignant cells (extract promoters with extended H3K4me3 for non-malignant cells), do you get the same patterns? And what are the numbers? Are these lower as compared to non-malignant cells? How would your observation of H3K4me3 broadening in CLL fit with the reduced expression of MLLs (Fig 3K).

Page 7: Fig 3I at the bottom should be 3K.

Fig. 6A, B: In the text 5 deregulated chromatin features are mentioned, but in 6B the authors show 6. Please clarify.

Reviewer #2:

Summary

This work represents an effort to comprehensively characterize differences between chronic lymphocytic leukemia cells and normal B cells with respect to genome-wide chromatin state, DNA methylation, gene expression, and genomic element regulation by DNA sequence-specific transcription factors. CLL-associated epigenetics changes identified by the authors include differences in large-scale repressive chromatin domains with respect to histone and DNA methylation, altered nucleosome positioning and histone modification distribution at active promoters, altered combinatorial histone modifications at repressed promoters, altered enhancer activation, and altered enhancer-promoter relationships. The authors used differential gene expression analysis between CLL and normal B cells, transcription factor binding motif analysis of genomic features highlighted in their epigenetic analysis, and a regulatory network model generated from published B cell transcriptional profiles to create a network model for mechanisms of epigenetic and transcriptional dysregulation in CLL. After identifying increased histone deacetylase activity in CLL cells versus normal B cells, they also used an HDAC inhibitor to study differences in the way enhancers and promoters are affected by HDAC perturbation in CLL and normal B cells, and relate these findings to their model.

General Remarks

In this work, Mallm, Iskar, Ishaque and colleagues present the results of an integrative genome-wide comparison of peripheral blood CLL cells to normal peripheral blood B cells with respect to a diverse set of features related to chromatin and transcriptional regulation. These include whole-genome assessment of DNA methylation, RNA-sequencing, chromatin accessibility (in bulk and at the single-cell level), diverse histone modifications, and nucleosome positioning. The number of subjects in each category were sufficient for valid statistical assessment of significant differences between CLL and normal B cells with regard to these features, and the analyses presented clearly demonstrate the high technical quality of these datasets, with few exceptions (the H3K9ac data appears to be of somewhat lesser quality than other datasets, for example). Together, this represents one of the most impressive and comprehensive epigenetic profiling efforts I have seen for any cancer type, and these datasets will certainly be an invaluable resource for the CLL field, and as a model for future investigations in other cancers. The authors are also to be applauded for their efforts in making not only raw and processed data, but also their custom analysis scripts available to the community.

With regards to the analysis and biological findings presented here, they are primarily of a descriptive and hypothesis-generating nature. Given the ambitiously broad scope of the profiling and analysis portions of the project, the value of their well-characterized datasets as a resource, and the challenges inherent in studying CLL for which few faithful *in vitro* or *in vivo* functional models exist, this does not overly detract from the value of the work. However, the authors should more clearly acknowledge this as a caveat in describing their findings. It would be best to avoid language that appears to claim an unproven functional consequence of chromatin mark associations in the abstract and discussion (e.g. "...loss of bivalent promoters indicated a reduced developmental capacity"). As another example, their network model predicts a functional association between TCF4 activation and BCL2 activation in CLL (described in the abstract, results, and discussion sections), but no functional experiments are performed to directly test this prediction. If the authors wish to highlight this hypothesized relationship in the abstract, they should make it clear that this is an example of the hypothesis-generating power of their data, rather than an experimentally demonstrated mechanism.

Another important caveat is warranted in presenting this data. It is well known that CLL proliferation, and activation of many transcriptional regulatory pathways critical for CLL pathogenesis (BCR signaling, MYC, Notch) occur primarily in tissues (lymph nodes & other secondary lymphoid organs, bone marrow, etc). See for example Herishanu et al 2014, cited by the authors. However, all samples studied in this work are (for practical reasons) from peripheral blood. The authors should discuss the inherent limitations of studying CLL peripheral blood cells with regard to understanding the biology that sustains these tumors.

Major points

The following points should be addressed in revisions in order for this paper to be acceptable for publication.

- Overall, this is a very impressive and comprehensive description of the epigenetic regulatory "landscape" of CLL as contrasted with normal mature B cells. However, in the absence of hypothesis-driven experiments, the authors seem to stretch their model and the literature excessively in some areas in order to construct mechanistic explanations for all of their major findings and purported relevance to CLL pathogenesis and therapy. The paper could be improved by de-emphasizing these pat explanations, and instead suggesting broad categories of future hypothesis-driven investigation that could be enhanced by using these impressive datasets as a resource.
- Figure 1b requires far more explanation for the reader to understand. Four different heatmap legends are provided, but I only see corresponding data for the top two (which have very similar color schemes).
- Figure 3 / S3 - Broadening of H3K4me3 domains is a phenomenon that has been studied in a number of developmental and cancer contexts, and has been linked to altered transcriptional elongation, and enhancer-dependent regulation, for example. Here, the authors seem to link this phenomenon to alternate TSS / promoter usage, but it's not clear that these phenomena are related. The authors should not rely on chromatin state (HMM) calls alone to claim that alternate TSS's are being used, but should support this by demonstrating the presence of transcripts with alternate 5' ends, either through RNA-Seq analysis or more targeted experiments, to disambiguate true alternate TSS usage from broadening H3K4me3. For example, it is not clear in figure 3F whether the gene on the bottom is truly using the lower TSS, or if it merely has H3K4me3 extending into the area of that annotation. Definitions of "active promoter" used in each sub-panel should be explicitly stated, and should be based on demonstration of alternate RNA transcripts for at least some of these figures.
- With regards to claims of TF motif enrichment at broadened H3K4me3 promoters (Fig S3C) and loss of promoter bivalency in CLL (Fig 3J), much more information needs to be provided about these analyses (merely showing the HOMER library PWM motif logos is not informative). Were these findings supported by both de novo and known motif analyses? What background regions were used? What was the statistical significance and frequency of these motif associations, and how much more significant were they than other enriched motifs (it is assumed that these were the MOST enriched motifs in these analyses, but that is not explicitly stated).
- The author's hypothesized association between MEF2 factors and MLL enzymes that generate H3K4me3 is poorly supported by the single citation provided, a review of MEF2 factors that in turn cites a single paper that linked MEF2C/D phosphorylation to recruitment of KMT2D, which is generally thought to be an enhancer-associated generator of H3K4me1 (not a promoter-associated generator of k4me3). Figure 3K uses the deprecated gene symbols for the MLL genes -this is ambiguous and should be avoided since "MLL2" has been used for both KMT2B and KMT2D (though generally the latter in humans). It is unclear why "MLL4" (presumably KMT2B) is not included in this figure. Given the very weak literature support for a direct connection between MEF2 factors and H3K4 trimethylation, this figure might better be omitted, and the corresponding claims significantly modified.
- Figure S4A is not mentioned in the text, and needs to be better explained with respect to which ENCODE TF ChIP-Seq datasets were analysed (B-lymphoblastoid lines only? Other cell types?) and how the TFBS clusters were generated.
- Figure 4B requires much better explanation of what is being depicted. Each row seems to represent the regions in a given state in CLL that had a different state in normal B cells, correct? And then whether they were enriched for that same state (or for p300) in ENCODE or FANTOM cell lines? It is surprising that so many of the different FANTOM and ENCODE lines showed enrichment for the enhancer state in CLL enhancers, given how cell type specific enhancers are. I would have expected significant enhancer state enrichment for ENCODE B cell lines (e.g. GM12878), but not others - it would be helpful to label the cell lines. It is unclear what "inactivation of enhancers in CLL occurred

mostly via the bivalent state" means - does this reflect the fact that the author's ChroHMM "bivalent" state shows more H3K4me1 / H3K27me3 bivalency (a variant "enhancer bivalent state" - Fig1B), rather than the classic k4me3 / k27me3 usually described at promoters? In its current state, it is unclear what points are being made in this figure.

- 4G - What background was used for the motif analysis of DMR-associated enhancers? Are enriched motifs in these enhancers significantly different from motifs enriched in a control set of CLL enhancers that are not associated with DMR's or differential acetylation?
- 4H - How were "NFATC1 binding sites" identified? Are these ChIP-Seq-proven binding sites (e.g. from an ENCODE GM12878 dataset) or simply enhancers that contained the NFAT motif? This figure is barely mentioned in the text and needs to be better explained.
- 5E - To my eye, there doesn't seem to be any relationship between the scATAC-Seq correlation matrix and the overlaid Hi-C contact domains. Perhaps a different locus, different scale, or genome-wide statistical measure would make this point better? As it stands, the author's claim that ATAC-Seq peak co-regulation correlated with topological features is not supported by the data presented.
- S5E - The depiction of co-regulated scATAC-Seq regions on chromosome 1 is perplexing. Enhancer-promoter loops do not operate on the scale of 10's to 100's of megabases as shown by the arcs in this figure. While the text briefly points out that these correlations could be driven by co-regulation in trans, the figure legend seems to describe them as enhancer-promoter pairs, which is highly unlikely. Needs clarification.
- The main text (page 10) also describes an enhancer-promoter pair analysis within windows of 200 kb, followed by identification of "re-wired" pairs in CLL versus normal cells, but no specific figures or tables are associated with this analysis (other than simple lists of features). CTCF site chromatin accessibility near these "re-wired pairs" is supposedly lost in CLL, but no analysis is presented as to whether this is a statistically significant association vs appropriate comparator genes. It is unclear whether the authors hypothesize that these CTCF pairs directly facilitate enhancer-promoter looping in the normal B cells, or rather represent topological domain boundaries (e.g. insulators) that are selectively lost in CLL and thus facilitate "re-wiring" across TAD boundaries. This is a complex topic, and should either be investigated in detail with appropriate statistical analysis and figures presented or perhaps omitted entirely and explored in a future paper.

Discussion:

- The authors mention several times (discussion and introduction) that BCR signaling pathway genes are not recurrently altered by mutations in CLL, but this is not true in an important sense. They should refer to the extensive literature on immunoglobulin gene VDJ stereotypy in CLL, which results in genetically encoded B cell receptors capable of autonomous signaling (e.g. PMID 22885698), or remove this claim.
- It is not at all clear how the papers cited support the authors' MEF2 / KMT2D / bivalent domain loss model (Ge et al, 2011; Kurt et al, 2017; Long et al, 2013) - Mef2 factors are not mentioned in any of these papers. This explanation needs to be removed or better supported.
- In the HDAC / enhancer section of the discussion the author's claim that MEF2 factors are connected to HDAC's "via BCOR" is also unusual (no reference is provided). MEF2 factors bind directly to Class II HDACs (HDAC4/5/7/9) via a well-characterized interface - crystal structures of this interaction are available. This interaction is mutually exclusive with Mef2 binding to p300 acetyltransferase or the CABIN1 corepressor and is likely controlled by post-translational modifications. The authors should incorporate a more conventional understanding of MEF2 factor function into their model.

Minor points

The following points suggest areas where improvements would enhance the paper's acceptability for

publication.

- The introduction could benefit from some re-writing for clarity and coherence.
- Figure 1a - The authors provide sample tracks for one CLL and one normal sample at the TCF4 locus, which is helpful for gauging the quality of their many datasets, although inclusion of their peak calls for each track might provide more insight into their genome-wide analysis. The figure legend should contain an appropriate reference to the chromHMM color code & corresponding states (e.g. chromHMM states are color coded as in Figure 1B)
- Figure 1C - Here, the authors appear to be using their data to reproduce a previously published result regarding DNA methylation-defined CLL subtypes. It's not clear whether there is functional evidence for a "developmental block" as the authors claim, except perhaps as explored in that prior paper.
- Figure 2 / S2 - The characteristics described for the CLL PMDs (100's of kb to mb scale, enrichment for repressive chromatin marks, boundary enrichment for CTCF or active promoter marks) are reminiscent of topologically associated domains (TADs) as defined by Hi-C studies, which the authors discuss later in the paper. If the CLL PMD's in fact correspond to repressed TADs (or perhaps series of TADs within larger structures such as Hi-C "B" compartments) this would be an interesting point to discuss.
- 4B The analysis of bidirectional transcription is quite interesting, but it's unclear how this was achieved. The methods describe separation of small and large RNAs prior to ribodepletion and library construction from the latter. Was sequencing of short RNA's performed (which would presumably contain most eRNA's), or is the bidirectional signal coming from the long RNA pool? Please clarify in methods and results text.
- 4C - I would expect a null signal along the diagonal for the matrix of state transitions. How is it possible to transition to the same state?
- 4D/E - There seems to be a typo in the legend, as the "number of sites with the motif" is not found on the y axis. What is the significance of dot size and color intensity? Many of the motifs in the HOMER library are highly redundant (e.g. ETS factors, which make up most of the significant signals in the downregulated chart, or different ATF factors). It is unlikely that these very similar motifs accurately distinguish between these different members of the same TF family. Perhaps a more parsimonious analysis could be shown for unique motifs detected in de novo HOMER motif analysis of the union enhancers, which would highlight more distinct motifs rather than many variants of the same top motifs.
- S4D/E - The legend mentions depiction of TFs with enriched motifs in super enhancers, but most of the genes highlighted in the hockey plots are not TF's but rather selected loci that are discussed in the text as "particularly relevant" for various reasons. It's not entirely clear why the TCRA locus is of interest (T cell receptors are not expressed in CLL), and the BCL2 super-enhancer, and those of several other genes highlighted in the discussion, are not indicated in the hockey plots.
- 4I - It is mentioned in the text that 6.3% of CLL super-enhancers were "unique to this study", but no citations are provided as to the prior super-enhancer papers or databases from which this comparison is made.
- Figure 5/S5 - While the "footprinting" analysis shown in 5B is intriguing, the fine-scale pattern may be largely driven by the sequence preferences of the TN5 transposase, which should be mentioned as a caveat. The overall differential signal levels are also not informative, since the sites were selected on that basis. The motif enrichment plots in S5C are much more compelling evidence for a specific role for these TF's.
- 5D - The discussion of the TF motif heterogeneity analysis is difficult to follow - the list of highly heterogeneous motifs is very enriched for Fos/Jun sites, which are not mentioned in the text, while SP1 and MYC (not in the list of heterogeneous motifs) are discussed in the text as possible interactors of NFYA/B. The text should probably follow more closely to the data, or be better

explained.

- Figures 6 and S6 provide an integrative framework that tie together the differential epigenetic and transcriptional features of CLL versus normal B cells into a model of TF-driven gene dysregulation in CLL. It is interesting that there is almost no overlap between TF's frequently mutated in CLL and the TF's identified as driving differential chromatin and transcriptional regulation in this study. For example, gain-of-function mutations in the TF NOTCH1 are the most common recurrent gene mutations in CLL, and Notch dysregulation has also been demonstrated in most NOTCH1 wild-type CLL samples, but no signature of Notch/RBPJ differential activity was described in this work. Does this reflect a limitation of the approaches used, or does the network model provide a possible explanation?

- Figure 7D - This figure shows significantly increased H3K27ac at promoters enriched for SP1, E2F, and KLF family members. The corresponding text describes ETS motif enrichment, which is not shown in the figure.

Discussion.

- CLL-specific large PMD's associated with repressive histone modifications and gene silencing are described, but confusingly are then associated with genes (IRF4 HIST1H1E, NOTCH1, IGLL5) that are mutated but also expressed in CLL. Are these two distinct classes of PMD's (associated with repression or alternately with expressed, recurrently mutated genes)? Needs clarification.

Supplementary data

- Some of the supplementary data Excel sheets suffer from conversion of certain gene names to dates (e.g. SEPT11 has been converted to the date "Sep-11")

Reviewer #3:

General comments.

The identification of partially methylated domains in CLL is a very interesting observation, but the manuscript is crowded with other minor and sometimes dubious observations. There is a lot of valuable data in this study that needs publishing once it is better analysed with more rigorous curation of the data, but it cannot be published in this form. The methods used to identify high confidence regulatory elements are well below the required standard and need repeating with more stringent criteria for defining them. The enhancer identification data set is flooded with background made up of huge regions of modified histones, probably representing entire domains or active chromatin. To be meaningful, the study should focus strictly on narrowly defined open chromatin regions (ATAC) linked to flanking histone regions. It should not define strings of modified nucleosomes as enhancers as this is not where the transcription factors are bound. Instead of being the starting point for identifying enhancers, the ATAC data is rarely used effectively, and this greatly undermines the whole analysis. Rather than coming first, ATAC appears in fig 5 in the context of footprinting, not to define enhancers in the first place. By focussing on histone modifications to define enhancers, the authors are actually identifying nucleosomal regions flanking open chromatin regions and not the enhancers themselves.

The authors could expand the ATAC motif analysis to get the most useful information from this study, and leave out more dubious stuff. Some of the most valuable information is effectively lost in fig S5C. More convincing footprints at these motifs would be good.

The study equates modified chromatin domains with enhancers, and then counts up TF motifs in these domains, which is something you can only do with enhancers. They need to separate these entities.

The HDAC experiments are probably providing valuable information, but the gene regulation networks pointing to HDACs may not be real if the analyses were flawed.

The authors may be trying too hard to make the analyses increasingly more intricate and more novel, but this runs the risk of losing sight of the basic biology, and moving away from our current definitions of what an enhancer is. An enhancer is more than a chromatin state as defined here.

Overall there is a sense that the authors are using complex tools to extract more information than the

data and the methods actually allow with any confidence. The end result is that the several good points of the study are lost in a morass of unreliable analyses and predictions based on these analyses.

Specific comments:

(1) The manuscript contains too few examples of primary data showing specific gene loci and showing individual tracks for the individual patients. For 2 or 3 key genes it would be good to have e.g. the ATAC tracks for all samples where ATAC was performed. The one example shown in Fig 1A does not allow such a comparison, and as a 1 Mb window the resolution is too low to fully interpret the patterns. Finer resolution is needed to see what is being measured. Given the problems illustrated below, it would be helpful to have regions called as enhancers marked by bars under the tracks as the EXCEL file suggests that many are not true peaks but broad regions of histone modification and not the open chromatin regions needed to specific enhancers where factors are likely to be bound.

(2) The TCF4 locus in Fig 1A has sharp peaks of H3K4me3 in both the transcribed and non-transcribed state. This is a little unusual, and might raise questions about the specificity of the antibody. This modification is normally associated with active promoters. Are the authors suggesting that these peaks are instead poised or bound by Polycomb?

(3) The text refers to an enhancer-like region identified at the TCF4 locus in Fig 1A based on peaks of H3K4me1, but the profiles actually shows several broad stretches of H3K4me1 and H3K4me27, and not sharp peaks. It is hard to see how this data can be used to identify a discrete enhancer. It would have helped if this putative element was labelled in Fig 1A.

(4) DNA regions should not be defined as "enhancers" based entirely on histone ChIP data. These "enhancer-like" regions can be defined as active chromatin regions, but should not be referred to as enhancers without more data, and not if there is no ATAC peak at the same site. E.g. One study found that only 26% of predicted enhancers had any actual enhancer activity (Kwasnieski JC, Fiore C, Chaudhari HG, Cohen BA. High-throughput functional testing of ENCODE segmentation predictions. *Genome Res.* (2014) 24:1595-602.). The text also refers to "genic enhancers" but it is unclear what distinguishes these as a group from poised and active enhancers. Does genic mean regions inside transcribed genes that also have H3K36me3? If so, it is a bit artificial to separate intergenic and exonic enhancers into 2 groups as they do the same things, and exon enhancers often do not regulate the gene where they reside, and may not then have H3K36me3.

(5) The data in table 8 lists 141,000 enhancers present in B cells and 238,000 enhancers in the combined data set. This is vastly more than you would reasonably expect to find in one cell type if meaningful criteria were used to delimit the data. A stringent analysis might identify 30,000 high confidence peaks that have the potential to be enhancers. It has become a problem in the field that some studies greatly over-estimate of the number of total enhancers in the genome (some claim more than 500,000) or in any one cell type. This arises by including too much low quality data or unsuitable definitions. It appears that the Roadmap consortium data cited here to validate the study also over-estimates numbers of enhancers, claiming 10% of the genome in B cells, so this is not helpful.

In the present study the huge number of "peaks" indicates that insufficient culling of insignificant peaks has been performed, and the wrong criteria used to define enhancers. The table also suggests that normal B cells have over 1000 enhancers greater than 10 kb in length. This means that the authors are defining entire active chromatin domains, not discrete enhancers. At the very least the authors should ensure that a distinct open chromatin region is present at the site where they define an enhancer. These should be discrete ATAC peaks, typically ~ 200-500 bp across, defined on the basis of a minimum meaningful threshold. My analysis of a random selection of the 140,000 enhancers identified in B cells raised many concerns. A good example highlighting the problems are 2 enhancers defined by the authors at the NFATC2IP locus in hg19 as

16 28961200 28962000

16 28963000 28965000

If these are viewed on the ENCODE GC B cell and CD20 B cell DNaseI tracks it is clear that these represent modified histones on either side of a sharp DNaseI peak between these two coordinates.

In this case the defined enhancers excluded the open chromatin region at the promoter in between. Next to this promoter is a sharp DNaseI peak that is called as an enhancer at

16 28956400 28957800

In this case the called region is in open chromatin, but this is a discrete conserved CTCF binding site and is most likely an insulator, not an enhancer.

At the CD20 locus (MS4A1) the entire coding region and promoter is defined as a series of enhancers spanning 15 kb at

11 60220600 60226000

11 60227000 60231600

11 60232000 60232800

11 60234600 60235400

At this locus a very probable enhancer exists as a very sharp DNaseI peak downstream but is called as a 3 kb enhancer at

11 60398800 60402200

While it is not useful to focus on just a few specific examples, this does leave the reviewer with the firm conviction that the entire enhancer data set needs redefining with more stringent criteria to include discrete regions of less than 1 kb, that are validated as open chromatin, with minimum peak heights defined on an empirical basis, so that they have a good chance of representing true regulatory elements and not just broad zones of active chromatin. In the current data set, 61,000 of the defined enhancers are greater than 1000 bp whereas the typical enhancer should be ~300 bp.

(6) The enhancer data file should include at least some numerical data on peak values, so that it would at least be possible for the reader to interpret the significance of each peak, without having to re-process and upload the data on a genome browser. In this case, a link to a genome browser session would have been a good idea so that the reviewers and the end reader can see the data for themselves.

(7) The Hidden Markov modelling of chromatin states may not be valid if the identification of chromatin features is over-stated to start with.

(8) In contrast to the claim in the text relating to fig 1D, it is not obvious that there is a significant increase in enhancers in CLL.

(9) Page 5 states that "we identified 1378 regulators ($p < 0.05$) with differential activity" which I think is far too many. This p value may mean that 5% of the 23,000 genes in the genome will be identified by chance, which is equivalent to 1150 genes, which could account for most of the regulators identified. I suggest that a lower p value cut-off would give more meaningful data. It would be better if they focussed on e.g. the top 100 differential regulators for this to be meaningful.

(10) Fig 2D would benefit by adding an analysis centred on open chromatin regions near the histone peaks at the boundaries of PMDs. This may show that an active promoter or enhancer also blocks spreading of repressive marks, maybe more so than CTCF. It is not acceptable that ATAC is left out of the PMD analysis.

(11) In fig 2E the authors should indicate whether the increased mutations are occurring in PMDs that gained (or lost) meCG compared to normal B cells. If methylation is gained at these sites, it could be that meCG is transmuting to TG or CA during replication. How many of these mutations involve these bases?

(12) The extension of H3K4me3 domains in fig 3A is rather minor and may depend on how this data is normalised. This analysis does not add much and is difficult to interpret. The claims made in 3C and 3D about gain of nucleosomes would need a high resolution map of average nucleosome spacing for e.g. the first 5 nucleosomes from the TSS, or the centred on the first nucleosome. Fig 3D is only one example and is too low resolution to see anything meaningful.

(13) The link between the gain of one modified promoter nucleosome, and the gain of alternate promoters is greatly over-stated given that alternate promoters are many kb away in fig 3F, not 200 to 400 bp away.

(14) In fig 4A the authors incorrectly use total Mb of chromatin as a way of defining enhancer quantities. This is totally inappropriate given the methodology used. An enhancer should be an entity at an open chromatin region, not a vast expanse of modified chromatin. For example, a 10 kb stretch of transcribed modified nucleosomes should not be called an enhancer, but this is what some loci in the attached enhancer table look like. This figure needs to state the number of discrete open chromatin regions with the right flanking modification, not Mb of chromatin of highly variable lengths.

The whole discussion on page 8 becomes invalid once the reader realises that discrete rigorously defined enhancers were not identified in the first place, so I will not comment on it at length. You cannot do a motif analysis of chromatin regions 1 to 10 kb long. There are many of these, and some are much longer. Motifs concentrate within 100 to 150 bp of an ATAC peak summit, a much smaller region than most enhancers defined here. The background must be huge.

I assume that some of the super-enhancers will also include the tracts of ~50 kb of modified chromatin that are in some places defined as single enhancers. This is not a great definition.

(15) Fig S5A cannot be interpreted without an inbuilt legend to the graphs. The ATAC data should show average profiles for CLL-specific ATAC peaks in CLL and in normal B cells. Fold change is not enough. We need an indication of peak height and width. The data here is over-processed to the point where you no longer know what it looks like. Where is some primary data?

It needs stating if the X axis in S5A is a natural or a log₂ scale. If they are using a 1.13 threshold, this has to be a log scale = ~2.3 actual difference.

(16) Fig 5B does not necessarily show TCF occupancy, it more likely shows sequence specific cleavage around a central TCA motif. This pattern is characteristic of the sequence bias of nucleases. Footprints normally look very different and are wider with a deeper drop at the footprint. The EBF data is less clear but this may also be influenced by sequence specificity which could explain the pattern. If we do not know how rigorous the calling is, it is hard to have confidence in 5C and 5D.

A simple table of actual motifs identified, and the % occurrence would help to make fig S5C more meaningful.

(17) I find it hard to know whether the gene regulation pairs defined by co-occurrence of sequence reads in scATAC are reliable or not. This is an interesting approach, but it is hard to see how they can reliably get the pairing right. In a uniform population of cells, expressing the same genes, there is no reason why they would detect an active promoter/enhancer pair in one cell and not another. It seems there is a lot of room for error here. These pairings should be mostly within 100 kb, rarely more than 1 Mb apart, but the pairings shown in S5E are typically 100 Mb apart. This seems to confirm that the method is not very reliable.

(18) On page 10, the authors seem to be using differential footprinting to assess CTCF occupancy. Differential occupancy analysis here would actually be much more reliable if they simply mapped ATAC sites at known CTCF sites or motifs. An insulator has one or more sharp ATAC peaks with a single CTCF motif in it. Footprinting is not needed here and is more likely to generate false differentials, because footprinting has more limitations than ATAC peak calling.

(19) The motif analysis in fig 7E is not valid, because the enhancers were not correctly identified. Stretches of modified nucleosomes in coding regions should not have more motifs than random sequence. It is possible that the analysis ends up biased towards simple motifs that exist everywhere. To interpret this, a motif table is needed with the actual motifs and with % found, and the predicted % background values shown. P values become unreliable when dealing with what seems to be 10% of the genome because of the vast number of motifs found.

(20) On page 4 and on page 11 the authors seem to want the region downstream of TCF4 to be both an enhancer and the co-regulated LINC01929 gene. This may actually be an enhancer-derived transcript. Maybe this should be clarified, and the predicted enhancer better defined.

(21) For the above reasons, the 231 Mb of novel genic enhancers (8% of the genome) referred to on page 13 are more likely to be just active gene coding regions and not enhancers. The authors will need to redo their analyses of histone modifications and ATAC peaks to separate active genes (with e.g. H3K27Ac) away from promoters and enhancers defined as open chromatin regions.

Minor comments

- (a) The citations to Puente et al and Horton et al are incomplete.
- (b) The link to MACS seems to be the wrong address.
- (c) Hyphens are missing in the protein names for IL-4 an NF- κ B, and kappa should be a Greek letter, and Sp1 and Sp2 have p in lower case.
- (d) The methods text and the data files should include information as to which build of the genome is being used to define chromosomal coordinates. This seems to be hg19.
- (e) The heat map in fig 2A should have a colour code.
- (f) On page 7, "B-cells lost H3K4me3 in CLL (Fig. 3H, I, see also Fig. 1C, state 7)" should refer to 1B.
- (g) The description of the nucleosome positioning method is inadequate, referring simply to H3 libraries. The methods should specify MNase-H3 ChIP-Seq as in the results section.
- (h) The vertical legend on fig S2I should be non-malignant, not "on malignant".
- (i) Fig 3K is not cited on page 7.
- (j) On page 11 a reference to figs 7C, S7A should be to 7C, S7B.
- (k) The description of Fig 7D on page 12 is incorrect as promoters with Sp and KLF motifs are changing, not ETS motifs.

1st Revision - authors' response

17th October 2018**Response to reviewer comments for manuscript MSB-18-8339****General comments (GCs) to the revised manuscript**

We are highly grateful to all reviewers for the significant work that they have put into the critical evaluation of our study and for the detailed, specific and constructive suggestions and comments to improve it. We have thoroughly revised our work and have addressed the issues as described in the point-by-point response to the specific reviewer comments below (highlighted in blue and renumbered) and feel that these revisions have significantly strengthened our manuscript. In addition, we have summarized how we have addressed more general issues that have been raised at more than one occasion by the reviewers.

GC1. Workflow and integration of different readouts

To better explain how the different readouts were integrated we have now included an additional work-flow scheme in Appendix Fig. S1B. It gives an overview about how data, methods and results are connected. In addition, we have clarified three issues throughout the manuscript. (i) The core set of TFs associated with deregulated chromatin features (Supplementary Table S2) was always based on TF binding motif analysis of differential ATAC-seq signal which is therefore the most fundamental data layer. Subsequently, the chromatin context was used to annotate the corresponding locus. (ii) The scATAC-seq analysis yields promoters and enhancers that showed high correlations for being simultaneously open in the same cell. The resulting connections expressed as a set of pairwise correlation coefficients yield an enhancer-promoter network. This network has now been integrated into the B cell gene regulatory network (GRN) to create what we have termed a "gene regulatory enhancer containing network" (GREN, Supplementary Data). (iii) From the complete GREN, a "CLL specific GREN" was extracted. It includes the connected network part that contains TFs from our CLL TF list, their target genes as well as linked chromatin modifiers that affect the

chromatin features aberrant in CLL. An additional requirement for this reduced network was that all included factors were deregulated in their activity/expression between CLL and non-malignant B cells.

GC 2. Additional experimental datasets for validation of model predictions.

We have clarified in the abstract and in the manuscript that we generate a number of predictions and hypotheses that await further experimental studies. Furthermore, in several instances we have improved the “quality control” of our analysis and validations with existing data. For example, the scATAC-seq based analysis of promoter-enhancer correlations includes now additional controls in Appendix Fig. S5 and the results are confirmed with interactions listed in the 4D nucleome database (Teng et al, 2015). To further validate our general approach and to demonstrate the predictive power of our model and analysis we have now included ChIP-seq experiments for the CTCF and EBF1 transcription factors that were conducted for four samples from healthy donors and eight CLL patients. Motifs of CTCF and EBF1 were among the top hits for TF binding motifs that were lost at CLL enhancers (Fig. 5C). We now confirm by ChIP-seq that these factors are indeed lost at enhancers as predicted from the ATAC-seq based binding motif analysis and apply the additional datasets in two exemplary analyses for the integrative assessment of CLL specific deregulation. In Fig. 5G we dissect the role of CTCF in promoter-enhancer wiring at the *NFkB2* locus and in Fig. 6D and 6E we exploit the EBF1 ChIP-seq data to validate interactions and changes in enhancer activity that are predicted by our CLL specific GREN. In addition, we would like to note that a study published during the revisions of our manuscript comes to similar conclusions about deregulated chromatin features and TFs in CLL as mentioned now in the revised text (Beekman et al, 2018).

GC 3. Organization of results

Our manuscript is associated with a large set of data. We understand that the high information density and the way we have presented them in the initial manuscript made it complicated to understand easily. In the revised manuscript we have now, based on the assessment of the reviewers, expanded or reduced relevant parts (see below). Furthermore, we have reordered the presentation of results into main figures, expanded view figures and appended supplementary figures to show results from the workflow in Appendix Fig. S1B and to better explain how the different parts of our results provide complementary information. Specifically, we have done the following revisions: (i) In Fig. 3 we now focus on the localized extension of H3K4me3 and its loss at bivalent promoters in CLL. (ii) The analysis of enhancers has been separated into two Fig. 4 and 5. A more general part in Fig. 4 compares the different enhancer definitions and integrates the contribution of the increased HDAC activity in CLL from the previous Fig. 7. The ATAC based analysis of TF binding motifs gained or lost in enhancers is now presented as Fig. 5. (iii) Fig. 6 is expanded and focuses on the integration of the enhancer-promoter correlations from the scATAC-seq into the GREN and testing the resulting predictions by ChIP-seq. (iv) The revised Fig. 6 integrates the main finding of our study. The associated “hand-made” schemes of the deregulated TF network in CLL and its associated links to chromatin modifiers (previous Figs. 6C and Fig. S6) have been replaced by the network scheme in Appendix Fig. S7. This scheme is directly derived in a clearly defined manner from the CLL specific GREN and the corresponding full network is provided as supplementary Cytoscape and xls files.

GC 4. Access to data access and analysis software

We provide original data and analysis software via different sources:

1. Original sequencing data for all samples are available at the European Genome-phenome Archive (<http://www.ebi.ac.uk/ega/>) under the accession number EGAS00001002518.
2. Metadata from the analyses are listed in Appendix Table S7 and are available as Datasets EV01-14 that have been uploaded to MSB with the manuscript.
3. Full bigBed and bigWig files for individual samples can be downloaded from the GEO archive at accession number GSE113336 with reviewer token abixoakhtytzeb .
4. Additional data are available at http://www.cancerepisy.org/data/cancerepisy_data/ , which will be updated and expanded in the future.
5. The software used is listed in Appendix Table S6. Custom analysis scripts and tools can be obtained from Github at <https://github.com/CancerEpiSys/Mallm-et-al-processing-scripts>.

Reviewer**#1:**

1. In the paper of Mallm et al., the epigenomic analysis of B-cells of 19 CLL patients and 9 healthy donors is described. The authors use a variety of different assays to analyze DNA methylation, chromatin states, nucleosome positioning, accessibility and the transcriptome to provide new insights into CLL pathogenesis. One of the main findings is a derived gene regulatory network (GRN) constructed from (public) RNA-seq data. The manuscript is rich in data that can be further explored by the community and most computational analysis is on par with the standards in the field.

We thank the reviewer for the overall positive evaluation of our work.

2. However, the current manuscript presents some weaknesses. First, the relation and the agreement between the RNA-seq derived network and the TF motif analysis is not clear. Are some of the TF motifs associated with alternative promoters or are CLL-specific enhancers also major regulators in the GRN?

We have addressed the above issues and clarified in the manuscript, how we derived the set of TFs listed in Appendix Table S2 that were associated with deregulated CLL chromatin features and other results (see GC 1 and point 4 below). The enhancers in CLL and non-malignant B cells have now been explicitly integrated in the CLL specific part of the GRN that we identified in our analysis, resulting in a gene regulatory enhancer containing network (GREN). While an interesting finding, the alternative promoter usage is more difficult to associate with a specific TF motif due a relatively low number of loci. Accordingly, these motifs were not included in the selection of our TF set.

3. Second, many of the key findings are based on computational analysis for which no validation (e.g. ChIP-seq or 4C-seq) was provided. The manuscript needs to be strengthened by validation of the findings.

We agree with the reviewer that our findings are based on our integrative analysis. We have now performed additional experiments (e.g., ChIP-seq of CTCF and EBF1) and extended the integrative analysis to validate our predictions as described below and in GC 2.

Major comments

4. The GRN is based on information theoretic analysis (ARACNE-AP) of published RNA-seq data and supplemented with current knowledge of known regulators in the field. How well does the derived model predict these known regulators? E.g. a ROC curve would provide a measure of the "quality" of this network. The abstract states that 'based on DNA-binding motifs, TFs were integrated into the GRN.' How was this done? How do the TF motifs relate to the VIPER-derived activity scores? It seems that the authors can use differential motif analysis to constrain their GRN to direct targets that are differential between healthy and CLL donors. This is however not clear from the methods section, and some figures (e.g. Fig 1E) appear to be based solely on the RNA-seq data.

We have now clarified the workflow (see GC 1 and Appendix Figure S1B). The TFs that we identified from the combination of ATAC-seq based Homer analysis, aberrant chromatin context and deregulated gene expression were used to select a connected part of the network that contained these TFs (shown in part in Fig 6D). We further used a ROC curve analysis to see how well our TF set derived from chromatin features (Appendix Fig S2) rationalizes deregulation on the transcriptome level in CLL. For this we ranked all TFs based on the significance of their activity changes between CLL and the B cell samples and then checked at which positions are the TFs from the independent motif analysis. This leads to the ROC curve shown below (**Figure R1**) and an AUC of 0.73. Thus, our TF set, which has been selected based on a binding motif analysis at promoters and enhancers with aberrant chromatin features in CLL, makes a significant contribution to transcriptional deregulation.

Figure R1. ROC curve for the contribution of TFs linked to aberrant chromatin features in CLL to changes in activity. Regulators were ranked according to the significance of their activity changes computed with VIPER between CLL and the B cell samples. Then it was checked at which positions the TFs from the motif analysis can be found to compute the ROC.

The quality of the GRN has been evaluated in two ways: First, we compared our B cell network with three publicly available ARACNE networks for prostate cancer, glioblastoma (GBM) and acute myeloid leukemia (AML) available in the R-package “aracne.networks” and calculated the activities of all regulators (transcription factors, transcriptional co-factors and signaling pathway related genes etc.) with our CLL RNA-seq data. All regulators with a significant activity change (p -value < 0.05) between the CLL and the non-malignant B-cell samples were then used for a pathway analysis. This analysis yielded “B cell receptor signaling pathway” as top hit for the B cell network as opposed to the other entities (prostate cancer, pos. 65; GBM, pos. 27; AML, pos. 6). This suggests that the network we have computed captures more specific B cell target genes than the three other networks. In addition, we calculated transcription factor activities with the MIPRIP framework we have developed. MIPRIP is an R-package to predict the most important transcription factors for the regulation of a gene of interest by using all the regulators binding to a gene’s promotor (Poos et al, 2018; Poos et al, 2016) (MIPRIP webpage: <https://www.leibniz-hki.de/en/miprip.html>). It uses a generic human network derived from different cell types and thus provides a global view of the transcriptional and regulatory circuitry. However, the MIPRIP method is restricted to transcription factors as regulators, while our ARACNE based B cell network includes also regulation of target genes by chromatin modifiers, signaling proteins and is thus preferable for our current analysis. We compared the computed transcription factor activities from MIPRIP with those obtained with VIPER/ARACNE and with the conventional differential gene expression analysis. There was a high overlap in the significantly up- or downregulated TF activities computed with the two different methods. Accordingly, we conclude that the ARACNE B cell network used in our study is well-suited to analyze deregulation of transcription factors and other regulators and their linked target genes.

5. The authors state that loss of DNA methylation and the accompanied loss of RNA-expression could be explained by accumulation of H3K9me3 and H3K27me3. This is not apparent from Fig 2D and Fig S2G. In Fig S2G, the accumulation of H3K27me3 seems much higher in the non-PMDs. Moreover, the PRC/repressed states do not cover 50% of the genome, unlike the PMDs. Can the authors show a negative correlation between DNA methylation and occupancy of these marks?

We agree with the reviewer that the enrichment of H3K27me3/H3K9me3 among PMDs is not apparent from chromatin states and are grateful for the additional analysis suggested. We calculated the average DNA methylation and H3K27me3 signal for each genomic region divided by PMDs. Appendix Fig S2E shows the negative correlation ($r = -0.3$) of DNA methylation with the H3K27me3 signal. We have rephrased the results section to describe the enrichment of repressive histone marks in PMDs.

6. Also, H3K36me3 is generally associated with gene transcription. In Fig 1A for CLL TCF4 seems high expressed according to the RNA-seq results, but H3K36me3 is low, while it is high in non-malignant cells that have no signal for TCF4 in the RNA-seq track.

We thank the reviewer for pointing this out. TCF4 is actually expressed in both CLL and non-malignant cells but the expression level in CLL is considerably higher. As the same track heights

were used in the figure the expression of TCF4 but was hardly visible for non-malignant cells. To visualize that TCF4 is indeed expressed we now show the RNA-seq tracks at different scales, which is also mentioned in the figure legend.

7. The authors identify novel TFs related to alternative promoter usage (FOXA1, LEF1, POUF3 OR REPIN1), differential acetylation (MEF2a/d, Klf4) and CLL-specific enhancers (e.g. NFkB, TCF4, ATF) but no validation was provided. ChIP-seq in healthy vs CLL for one or two of these TFs would significantly advance the current manuscript. Similarly, the promoter-enhancer rewiring inferred from single cell co-accessibility is interesting. These claims would be substantially more valuable when validated in differential setting for some loci of interest using a chromatin conformation assay like 4C-seq. Along the same lines: on page 10: "distinct patterns of spatially co-regulated activity hubs ... affect central pathways in the CLL pathogenesis". If the differential occupancy of CTCF and promoter-enhancer rewiring is so essential to CLL pathogenesis, computational derivation from ATAC-seq is insufficient. Again, such claims would be substantially more valuable when CTCF occupancy was measured using ChIP-seq and chromatin interactions for some loci were analyzed in a differential setting using 4C-seq.

This point is well taken and we have performed ChIP-seq for EBF1 and CTCF for four samples from healthy donors and eight CLL patients. We could confirm the predictions from the ATAC-seq based TF motif analysis for the two factors and that CTCF binding to enhancers is correlated with changes of genes expression of the next nearest gene. Via the additional analysis of both the CTCF ATAC and ChIP, we found that 90% of enhancers that change in promoter-enhancer correlation had CTCF stably bound to them (Fig. 6G and associated text). We agree with the reviewer that we do not show physical interactions but nevertheless we are convinced that we indeed detect co-regulated open chromatin regions. The loss of this co-regulation (either by loss of physical contact or loss of a transcription factor targeting multiple sites) occurs at genes relevant to the CLL pathogenesis. This has been clarified in the main text.

8. In Fig 7 the authors essentially show that HDACi's are doing what they are expected to do (reduce HDAC activity and increase H3 acetylation) at the molecular level. What happens at the cell biological to these treated cells? Are they differentiating (as would be expected from the loss of the CLL heterochromatinized state claimed by the authors) or do these cells have altered viability? Is this specific for the CLLs, or are also the non-malignant cells responding?

We did not check for a differentiation phenotype as the differentiation process takes substantially longer (24 hrs) than induction of apoptosis (6 hrs) in primary CLL cells. With respect to apoptosis however, CLL cells are much more sensitive to HDACi than non-malignant B-cells. We have included exemplary data in Appendix Figure S4 D & E.

Minor comments

9. Please clarify Fig 1B. The legend refers to log change and $-\log_{10}$ p-values but how are they represented in the figure?

Thank you for pointing this out. We have now simplified the legend to this figure.

10. How many of the BCR signaling components are occupied by the FOXA1, LEF1, POUF3 OR REPIN1? I.e. what is the consensus between the functional network analysis and the motif analysis?

The analysis of FOXA1, TCF4/LEF1, POUF3 and REPIN1 was simply based on the correlation between the expression of these TFs and the expression of alternative transcripts. Following the excellent suggestion of the reviewers, we have constructed a novel gene regulatory enhancer containing network (GREN, see general comment GC 1 above, exemplary part shown in Figure 6D). Using the core TFs identified in the GREN and all their target genes, the KEGG pathway "B-cell receptor signaling" was significantly enriched. The consensus between our motif analyses and the new GREN are described in Figure 6.

11. Fig 1C: Can the authors include the 9 normal B cells for comparison?

Thank you for this suggestion. The normal B cells are now included in the analysis depicted in EV Figure 1A.

12. Fig 3B: Do we see H3K4me3 broadening or stronger nucleosome positioning? The H3K4me3 signal to the right seems to disappear in CLLs. Fig 3C, D: If you repeat the analysis for the non-malignant cells (extract promoters with extended H3K4me3 for non-malignant cells), do you get the same patterns? And what are the numbers? Are these lower as compared to non-malignant cells? How would your observation of H3K4me3 broadening in CLL fit with the reduced expression of MLLs (Fig 3K).

It is now described in the text that 2639 out of 2785 promoters with the extended H3K4me3 promoter signal also gain nucleosomes according to the MNase H3 ChIP-seq mapping. The different nucleosome occupancy gain patterns at promoters are now depicted in Fig EV3B. The numbers of extended H3K4me3 signals are now stated in the paper for both malignant and non-malignant cells. In short, we only detect two extended H3K4me3 regions in non-malignant cells. We link reduced MLL (or KMT as now stated in both text and figure) expression to loss of bivalency in line with previous findings. The extension of the H3K4me3 signal is probably simply caused by the positioning of additional nucleosomes that are then modified alongside the neighboring ones. These two separate ways to change the H3K4me3 signal are now stated in the text accompanying Fig. 3.

13. Page 7: Fig 3I at the bottom should be 3K.

The error has been corrected in the revised manuscript.

14. Fig. 6A, B: In the text 5 deregulated chromatin features are mentioned, but in 6B the authors show 6. Please clarify.

The error has been corrected in the revised manuscript.

Reviewer #2:

Summary

This work represents an effort to comprehensively characterize differences between chronic lymphocytic leukemia cells and normal B cells with respect to genome-wide chromatin state, DNA methylation, gene expression, and genomic element regulation by DNA sequence-specific transcription factors. CLL-associated epigenetic changes identified by the authors include differences in large-scale repressive chromatin domains with respect to histone and DNA methylation, altered nucleosome positioning and histone modification distribution at active promoters, altered combinatorial histone modifications at repressed promoters, altered enhancer activation, and altered enhancer-promoter relationships. The authors used differential gene expression analysis between CLL and normal B cells, transcription factor binding motif analysis of genomic features highlighted in their epigenetic analysis, and a regulatory network model generated from published B cell transcriptional profiles to create a network model for mechanisms of epigenetic and transcriptional dysregulation in CLL. After identifying increased histone deacetylase activity in CLL cells versus normal B cells, they also used an HDAC inhibitor to study differences in the way enhancers and promoters are affected by HDAC perturbation in CLL and normal B cells, and relate these findings to their model.

General Remarks

15. In this work, Mallm, Iskar, Ishaque and colleagues present the results of an integrative genome-wide comparison of peripheral blood CLL cells to normal peripheral blood B cells with respect to a diverse set of features related to chromatin and transcriptional regulation. These include whole-genome assessment of DNA methylation, RNA-sequencing, chromatin accessibility (in bulk and at the single-cell level), diverse histone modifications, and nucleosome positioning. The number of subjects in each category were sufficient for valid statistical assessment of significant differences between CLL and normal B cells with regard to these features, and the analyses presented clearly demonstrate the high technical quality of these data sets, with few exceptions (the H3K9ac data appears to be of somewhat lesser quality than other data sets, for example). Together, this represents one of the most impressive and comprehensive epigenetic profiling efforts I have seen for any cancer type, and these data sets will certainly be an invaluable resource for the CLL field, and as a

model for future investigations in other cancers. The authors are also to be applauded for their efforts in making not only raw and processed data, but also their custom analysis scripts available to the community.

We would like to thank the reviewer for the enthusiasm about our findings and have continued to work on making the data and software accessible (see GC 4).

16. With regards to the analysis and biological findings presented here, they are primarily of a descriptive and hypothesis-generating nature. Given the ambitiously broad scope of the profiling and analysis portions of the project, the value of their well-characterized data sets as a resource, and the challenges inherent in studying CLL for which few faithful *in vitro* or *in vivo* functional models exist, this does not overly detract from the value of the work.

We would like to thank the reviewer for acknowledging the value of our findings for the general readership of *Molecular Systems Biology*.

17. However, the authors should more clearly acknowledge this as a caveat in describing their findings. It would be best to avoid language that appears to claim an unproven functional consequence of chromatin mark associations in the abstract and discussion (e.g. "...loss of bivalent promoters indicated a reduced developmental capacity"). As another example, their network model predicts a functional association between TCF4 activation and BCL2 activation in CLL (described in the abstract, results, and discussion sections), but no functional experiments are performed to directly test this prediction. If the authors wish to highlight this hypothesized relationship in the abstract, they should make it clear that this is an example of the hypothesis-generating power of their data, rather than an experimentally demonstrated mechanism.

We agree and have removed the above predictions from the abstract and have modified results and discussion sections to make clear what are model derived hypotheses (e.g., stating that the functional connection between TCF4 and BCL2 is a prediction derived from our network model) as opposed to a prediction from the network model that has been backed up by additional experimental data (e.g., EBF1 driving an enhancer of the SNX22 gene). Wherever relevant we no longer suggest causative links but rather clearly state that our observations are frequently of correlative nature and use them to generate novel hypotheses.

18. Another important caveat is warranted in presenting this data. It is well known that CLL proliferation, and activation of many transcriptional regulatory pathways critical for CLL pathogenesis (BCR signaling, MYC, Notch) occur primarily in tissues (lymph nodes & other secondary lymphoid organs, bone marrow, etc). See for example Herishanu et al 2014, cited by the authors. However, all samples studied in this work are (for practical reasons) from peripheral blood. The authors should discuss the inherent limitations of studying CLL peripheral blood cells with regard to understanding the biology that sustains these tumors.

We agree with the reviewer that the origin of the samples should be mentioned and discussed in the context of our results. To this end, we now included a passage in the first paragraph of the results section to point out this shortcoming.

Major points

The following points should be addressed in revisions in order for this paper to be acceptable for publication.

19. Overall, this is a very impressive and comprehensive description of the epigenetic regulatory "landscape" of CLL as contrasted with normal mature B cells, However, in the absence of hypothesis-driven experiments, the authors seem to stretch their model and the literature excessively in some areas in order to construct mechanistic explanations for all of their major findings and purported relevance to CLL pathogenesis and therapy. The paper could be improved by de-emphasizing these pat explanations, and instead suggesting broad categories of future hypothesis-driven investigation that could be enhanced by using these impressive data sets as a resource.

We feel that the paper is indeed a valuable resource and we thank the reviewer for his/her appreciation of our collected data (see GC 4). Though still including some hypotheses directly

derived from our work in the paper we have made it clear now that our data do not provide detailed mechanistic explanations of CLL pathogenesis. Rather, we have now revised the discussion to emphasize more on findings directly derived from our study and conceptual advancements in our work that could guide further investigations of experimental validation of CLL disease features.

20. Figure 1b requires far more explanation for the reader to understand. Four different heatmap legends are provided, but I only see corresponding data for the top two (which have very similar color schemes).

We thank the reviewer for pointing this out. We have removed the additional 2 legend items, changed the color scale of the emission heatmap, have made the panel less cluttered to make it easier to understand, and have revised the figure legend.

21. Figure 3 / S3 - Broadening of H3K4me3 domains is a phenomenon that has been studied in a number of developmental and cancer contexts, and has been linked to altered transcriptional elongation, and enhancer-dependent regulation, for example. Here, the authors seem to link this phenomenon to alternate TSS / promoter usage, but it's not clear that these phenomena are related. The authors should not rely on chromatin state (HMM) calls alone to claim that alternate TSS's are being used, but should support this by demonstrating the presence of transcripts with alternate 5' ends, either through RNA-Seq analysis or more targeted experiments, to disambiguate true alternate TSS usage from broadening H3K4me3. For example, it is not clear in figure 3F whether the gene on the bottom is truly using the lower TSS, or if it merely has H3K4me3 extending into the area of that annotation. Definitions of "active promoter" used in each sub-panel should be explicitly stated, and should be based on demonstration of alternate RNA transcripts for at least some of these figures.

We have clarified this section stating that the H3K4me3 extension of 1-2 nucleosomes observed is not directly linked to alternative promoter usage, which is in turn was identified from the emergence of H3K4me3 at additional transcription start sites.

22. With regards to claims of TF motif enrichment at broadened H3K4me3 promoters (Fig S3C) and loss of promoter bivalency in CLL (Fig 3J), much more information needs to be provided about these analyses (merely showing the HOMER library PWM motif logos is not informative). Were these findings supported by both de novo and known motif analyses? What background regions were used? What was the statistical significance and frequency of these motif associations, and how much more significant were they than other enriched motifs (it is assumed that these were the MOST enriched motifs in these analyses, but that is not explicitly stated).

Thank you for pointing to the missing information. We have updated the figure to include the statistical significance and frequency as requested. The background, which in this case contains all non-extended promoters, is now stated in the text.

23. The author's hypothesized association between MEF2 factors and MLL enzymes that generate H3K4me3 is poorly supported by the single citation provided, a review of MEF2 factors that in turn cites a single paper that linked MEF2C/D phosphorylation to recruitment of KMT2D, which is generally thought to be an enhancer-associated generator of H3K4me1 (not a promoter-associated generator of H3K4me3). Figure 3K uses the deprecated gene symbols for the MLL genes -this is ambiguous and should be avoided since "MLL2" has been used for both KMT2B and KMT2D (though generally the latter in humans). It is unclear why "MLL4" (presumably KMT2B) is not included in this figure. Given the very weak literature support for a direct connection between MEF2 factors and H3K4 trimethylation, this figure might better be omitted, and the corresponding claims significantly modified.

We agree with the reviewer that the naming of histone modifiers is ambiguous and have clarified the naming in the figure and included KMT2B as suggested. Still we believe that the expression levels of histone modifiers should be displayed in the manuscript. As the reviewer rightly commented we can only speculate about the relationship between MEF2C/D and KMTs but have now included

more references that would further support our hypothesis. The text was significantly changed as recommended and the heatmap moved to the supplements.

24. Figure S4A is not mentioned in the text, and needs to be better explained with respect to which ENCODE TF ChIP-Seq datasets were analysed (B-lymphoblastoid lines only? Other cell types?) and how the TFBS clusters were generated.

We apologize for the lack of clarity. This figure has been moved to Figure EV4B. We have amended the manuscript mentioning that transcription factor binding sites are enriched in the TSS, enhancer active and bivalent chromatin states. The heatmap was simplified into clusters of transcription factors as originally there were too many TFs to meaningfully visualize. The cluster and the description how they have been generated is now given in Appendix Table S4 and the figure is now referenced in the main text.

25. Figure 4B requires much better explanation of what is being depicted. Each row seems to represent the regions in a given state in CLL that had a different state in normal B cells, correct? And then whether they were enriched for that same state (or for p300) in ENCODE or FANTOM cell lines? It is surprising that so many of the different FANTOM and ENCODE lines showed enrichment for the enhancer state in CLL enhancers, given how cell type specific enhancers are. I would have expected significant enhancer state enrichment for ENCODE B cell lines (e.g. GM12878), but not others - it would be helpful to label the cell lines. It is unclear what "inactivation of enhancers in CLL occurred mostly via the bivalent state" means - does this reflect the fact that the author's ChroHMM "bivalent" state shows more H3K4me1 / H3K27me3 bivalency (a variant "enhancer bivalent state" - Fig1B), rather than the classic k4me3 / k27me3 usually described at promoters? In its current state, it is unclear what points are being made in this figure.

We thank the reviewer for pointing out the lack of clarity regarding Fig. 4B. The so called "bivalent" chromatin state (also referred to as a "poised" state) is transcriptionally silent and involves presence of the repressive mark H3K27me3 alongside an active mark of H3K4me1/2/3. The earlier chromatin segmentation models used in the ENCODE study originally showed the bivalent states being primarily a combination of H3K27me3 and H3K4me1 (Segway segmentation, state "EnhancerPoised") but also H3K27me3 with mainly H3K4me2 and H3K4me3 (ChromHMM segmentation, state "PromoterPoised"). Thus, the concept of the combination of H3K27me3 and H3K4me1 to describe bivalent marks in enhancers is not new. This was recapitulated in the Roadmap Project 15 state segmentation which also had a specific "Bivalent Enhancer" chromatin state. Figure 4B shows the overlap enrichment of other enhancer sets to our identified chromatin states using the ChromHMM OverlapEnrichment tool to show enrichment of these enhancer calls in our active enhancer and TSS states as well as in our bivalent state. From this we would infer that since the bivalent state is essentially inactive, bivalency is the most enriched state for inactive enhancers. To clarify what is being shown, we have revised Fig 4B (now Fig. EV4C) its legend and the methods description. Like the reviewer, we were expecting many of the FANTOM5 enhancer to be mainly enriched in repressed chromatin due to the cell specificity of these enhancers. However, there are still many overlaps between different FANTOM enhancer sets as, for example, 109 of the 209 FANTOM large intestine enhancers overlap with FANTOM blood enhancers.

26. 4G - What background was used for the motif analysis of DMR-associated enhancers? Are enriched motifs in these enhancers significantly different from motifs enriched in a control set of CLL enhancers that are not associated with DMR's or differential acetylation?

We apologize for this omission. We now improved our TF-binding site analysis of DMRs by focusing on the ATAC-seq sites overlapping with the DMRs. All the remaining ATAC-seq sites not overlapping with the DMRs were treated as the background. We have modified the figure legend of Figure EV2 to clarify this point.

27. 4H - How were "NFATC1 binding sites" identified? Are these ChIP-Seq-proven binding sites (e.g. from an ENCODE GM12878 dataset) or simply enhancers that contained the NFAT motif? This figure is barely mentioned in the text and needs to be better explained.

These are binding sites identified using the HOMER tool, which in turn uses the NFATC1 binding motif and have modified the text to reflect this. We attempted to experimentally validate the results by CHIP-seq of NFATC1 in our patient samples but the signal-to-noise ratio of the sequencing results was not of sufficient quality to draw any meaningful conclusions, most likely due to the quality of the antibodies tested.

28. 5E - To my eye, there doesn't seem to be any relationship between the scATAC-Seq correlation matrix and the overlaid Hi-C contact domains. Perhaps a different locus, different scale, or genome-wide statistical measure would make this point better? As it stands, the author's claim that ATAC-Seq peak co-regulation correlated with topological features is not supported by the data presented.

We have rephrased the text to make clear that the scATAC-seq correlations indeed do not fully reproduce the Hi-C contact domains and have removed this from the figure. This difference could be present for two reasons: First, the Hi-C domains were from a lymphoid cell and not from primary CLL cells. We also investigated the recently published HiC dataset for CLL cells but found its resolution insufficient for our purposes (Beekman et al, 2018). Second, correlations in accessibility derived from the scATAC-seq reflect co-regulation of promoter/enhancer sites. These can involve physical contact as in HiC but can also arise from co-regulation *in trans*, for example by the over-expression of a TF within a given cell that binds two co-regulated sites. Accordingly, some TAD boundaries show a good overlap with the scATAC-seq pattern, while other correlated open sites retrieved from scATAC-seq do not arise from physical interactions within a TAD. In the revised manuscript we have clarified this issue. Furthermore, we find that 68% of the promoter-enhancer pairs identified from scATAC-seq are also found in the 4D nucleome database (Teng et al, 2015). It is noted that the pair-wise correlations derived in our analysis make no assumptions about the underlying molecular mechanism and are conceptually equivalent to the pair-wise correlations used to describe the B cell GRN. Accordingly, these pair-wise correlations can be used to extend the GRN to include enhancer-gene linkages as done in our study.

29. S5E - The depiction of co-regulated scATAC-Seq regions on chromosome 1 is perplexing. Enhancer-promoter loops do not operate on the scale of 10's to 100's of megabases as shown by the arcs in this figure. While the text briefly points out that these correlations could be driven by co-regulation *in trans*, the figure legend seems to describe them as enhancer-promoter pairs, which is highly unlikely. Needs clarification.

We agree with the reviewer that the previous Fig. S5E (now removed) should probably not be mentioned in the context of promoter-enhancer connections. We do not want to make the claim that the long-range correlations shown in the figure are promoter-enhancer connections (which are indeed expected to be more short-range) but are more likely to occur *in trans* via binding of the same TF (see response to point 28). This has been addressed by limiting the maximal distance between promoters and enhancers to 100 kb. The length distribution of wired promoters and enhancers is now also included in the manuscript clearly showing a preference for shorter distances within the 100 kb window (Appendix Figure S5G).

30. The main text (page 10) also describes an enhancer-promoter pair analysis within windows of 200 kb, followed by identification of "re-wired" pairs in CLL versus normal cells, but no specific figures or tables are associated with this analysis (other than simple lists of features). CTCF site chromatin accessibility near these "re-wired pairs" is supposedly lost in CLL, but no analysis is presented as to whether this is a statistically significant association vs appropriate comparator genes. It is unclear whether the authors hypothesize that these CTCF pairs directly facilitate enhancer-promoter looping in the normal B cells, or rather represent topological domain boundaries (e.g. insulators) that are selectively lost in CLL and thus facilitate "re-wiring" across TAD boundaries. This is a complex topic, and should either be investigated in detail with appropriate statistical analysis and figures presented or perhaps omitted entirely and explored in a future paper.

Since we have performed CTCF CHIP-seq we included these datasets in the context of the re-wiring analysis. We found that 90% of re-wired enhancers are stably bound by CTCF within the enhancer in both malignant and non-malignant cells. This suggests that CTCF is needed for promoter-

enhancer contacts and that the re-targeting is driven either by variable CTCF binding sites outside of enhancers or by other factors such as YY1 or cohesin.

Discussion

31. The authors mention several times (discussion and introduction) that BCR signaling pathways genes are not recurrently altered by mutations in CLL, but this is not true in an important sense. They should refer to the extensive literature on immunoglobulin gene VDJ stereotypy in CLL, which results in genetically encoded B cell receptors capable of autonomous signaling (e.g. PMID 22885698), or remove this claim.

We have now included the landmark paper by Hassan Jumaa and colleagues and have in addition pointed out in the introduction that the BCR itself is subject to biased usage of IGHV genes and that somatic hypermutation is strongly correlated with prognosis, underlining its relevance in the pathomechanism of CLL.

32. It is not at all clear how the papers cited support the authors' MEF2 / KMT2D / bivalent domain loss model (Ge et al, 2011; Kurt et al, 2017; Long et al, 2013) - Mef2 factors are not mentioned in any of these papers. This explanation needs to be removed or better supported.

We apologize for not referencing the correct papers here. We have now included additional references that would further support our hypothesis.

33. In the HDAC / enhancer section of the discussion the author's claim that MEF2 factors are connected to HDAC's "via BCOR" is also unusual (no reference is provided). MEF2 factors bind directly to Class II HDACs (HDAC4/5/7/9) via a well-characterized interface - crystal structures of this interaction are available. This interaction is mutually exclusive with Mef2 binding to p300 acetyltransferase or the CABIN1 corepressor and is likely controlled by post-translational modifications. The authors should incorporate a more conventional understanding of MEF2 factor function into their model.

The panobinostat section has been moved to our general enhancer section and has also been completely revised. Now we point to changes in the reactivation of enhancers by panobinostat and affected pathways, rather than investigating specific transcription factors being involved in the changes of the histone acetylation since this is, as suggested by the reviewer, indeed very complex.

Minor points

The following points suggest areas where improvements would enhance the paper's acceptability for publication.

34. The introduction could benefit from some re-writing for clarity and coherence.

We have simplified sentences and structure of the introduction to improve readability.

35. Figure 1a - The authors provide sample tracks for one CLL and one normal sample at the TCF4 locus, which is helpful for gauging the quality of their many datasets, although inclusion of their peak calls for each track might provide more insight into their genome-wide analysis. The figure legend should contain an appropriate reference to the chromHMM color code & corresponding states (e.g. chromHMM states are color coded as in Figure 1B)

The figure legend of Figure 1A is now revised to include a reference for the chromHMM color codes. To address the valid point about peak calls all the track files together with the peak files are made accessible from the NCBI GEO and CancerEpiSys website (see GC 4). Thus, it is possible to overlay these tracks locally for any gene of interest using IGV.

36. Figure 1C - Here, the authors appear to be using their data to reproduce a previously published result regarding DNA methylation-defined CLL subtypes. It's not clear whether there is functional evidence for a "developmental block" as the authors claim, except perhaps as explored in that prior paper.

We agree with the reviewer in that we reproduce the correlative results of previous publications by showing similarity of DNA-methylation patterns of CLL cells with non-malignant B-cells of different stages. Although we strongly believe that this is representative of a developmental block at least for the IGHV non-mutated more aggressive subtype of CLL, we have no functional proof and have thus removed this statement.

37. Figure 2 / S2 - The characteristics described for the CLL PMDs (100's of kb to mb scale, enrichment for repressive chromatin marks, boundary enrichment for CTCF or active promoter marks) are reminiscent of topologically associated domains (TADs) as defined by Hi-C studies, which the authors discuss later in the paper. If the CLL PMD's in fact correspond to repressed TADs (or perhaps series of TADs within larger structures such as Hi-C "B" compartments) this would be an interesting point to discuss.

We are thankful for this suggestion. To address this, we calculated the overlap of HiC B compartments among PMD regions. As shown in Figure EV2, we found both HiC B compartments and lamina-associated domains to be enriched in PMD regions. We revised the manuscript to include a section discussing the findings.

38. 4B The analysis of bidirectional transcription is quite interesting, but it's unclear how this was achieved. The methods describe separation of small and large RNAs prior to ribodepletion and library construction from the latter. Was sequencing of short RNA's performed (which would presumably contain most eRNA's), or is the bidirectional signal coming from the long RNA pool? Please clarify in methods and results text.

Indeed, we agree with the reviewer that the analysis of the bidirectional transcription is interesting. We have improved our methods description on how this was achieved. In short, we have conducted separate sequencing of both long and short RNAs and used the combined data to map bidirectional transcript. We have also added an additional supplementary figure (Appendix Fig. S4A) which describes the chromatin states at the identified bi-directionally transcribed loci.

39. 4C - I would expect a null signal along the diagonal for the matrix of state transitions. How is it possible to transition to the same state?

We would like to thank the reviewer to point this out. The zero diagonal is now present due to an additional modification: only the transition of the consensus non-malignant and consensus CLL state was considered whereas in the previous version the matrix of all non-malignant to CLL transitions were included. Thus, there was previously a constraint that the consensus CLL and non-malignant state had to be different. We expect the current version of the panel to be more intuitive.

40. 4D/E - There seems to be a typo in the legend, as the "number of sites with the motif" is not found on the y axis. What is the significance of dot size and color intensity? Many of the motifs in the HOMER library are highly redundant (e.g. ETS factors, which make up most of the significant signals in the downregulated chart, or different ATF factors). It is unlikely that these very similar motifs accurately distinguish between these different members of the same TF family. Perhaps a more parsimonious analysis could be shown for unique motifs detected in de novo HOMER motif analysis of the union enhancers, which would highlight more distinct motifs rather than many variants of the same top motifs.

The typo has been removed. As we always show the results from the known motif search we also do this in case of the enhancers for consistency. It is true however, that we cannot discriminate between very similar motifs but we have focused this type of analysis by using our ATAC-seq data and color-coded TFs from the same family with similar binding motifs for clarity (compare Figure 5A and 5C).

41. S4D/E - The legend mentions depiction of TFs with enriched motifs in super enhancers, but most of the genes highlighted in the hockey plots are not TF's but rather selected loci that are discussed in the text as "particularly relevant" for various reasons. It's not entirely clear why the TCRA locus is of interest (T cell receptors are not expressed in CLL), and the BCL2 super-enhancer, and those of several other genes highlighted in the discussion, are not indicated in the hockey plots.

The hockey plots have been removed from the manuscript.

42. 4I - It is mentioned in the text that 6.3% of CLL super-enhancers were "unique to this study", but no citations are provided as to the prior super-enhancer papers or databases from which this comparison is made.

We have now mentioned the database used in our study in the methods section which was "dbSuper".

43. Figure 5/S5 - While the "footprinting" analysis shown in 5B is intriguing, the fine-scale pattern may be largely driven by the sequence preferences of the TN5 transposase, which should be mentioned as a caveat. The overall differential signal levels are also not informative, since the sites were selected on that basis. The motif enrichment plots in S5C are much more compelling evidence for a specific role for these TF's.

We now mention the TN5 sequence preference as a caveat in the manuscript and the motif analysis has been moved to the main figure panel to highlight the more convincing finding as suggested by the reviewer.

44. 5D - The discussion of the TF motif heterogeneity analysis is difficult to follow - the list of highly heterogeneous motifs is very enriched for Fos/Jun sites, which are not mentioned in the text, while SP1 and MYC (not in the list of heterogeneous motifs) are discussed in the text as possible interactors of NFYA/B. The text should probably follow more closely to the data, or be better explained.

We agree with the reviewer and the text has now been fully restructured to follow the findings presented.

45. Figures 6 and S6 provide an integrative framework that tie together the differential epigenetic and transcriptional features of CLL versus normal B cells into a model of TF-driven gene dysregulation in CLL. It is interesting that there is almost no overlap between TF's frequently mutated in CLL and the TF's identified as driving differential chromatin and transcriptional regulation in this study. For example, gain-of-function mutations in the TF NOTCH1 are the most common recurrent gene mutations in CLL, and Notch dysregulation has also been demonstrated in most NOTCH1 wild-type CLL samples, but no signature of Notch/RBPJ differential activity was described in this work. Does this reflect a limitation of the approaches used, or does the network model provide a possible explanation?

As the reviewer pointed out, the majority of patients with or without NOTCH1 mutations show activation of the NOTCH1 signaling cascade. Consistent with this expectation we find NOTCH1 as a deregulated gene in our study and it is also present in the CLL specific GREN that we derive to integrate our finding (see GC1 and Fig. R2 below).

Figure R2. Part of the CLL specific GREN that includes NOTCH1. Color code: active enhancers, dark, light and medium green for CLL only ("C"), only in non-malignant B cells ("H"), respectively; TFs, red; target genes, grey; chromatin modifier, blue.

Furthermore, it is noted that we find a loss of EBF1 at enhancers as one of our most prominent features, which was validated. As discussed previously, EBF consensus binding sites overlap with high-affinity RBPJ sites and it has been concluded that RBPJ complexes can bind at these sites only after EBF1 dissociation (Miele, 2011). It is noted that EBF1 can act as a repressor of NOTCH1/RBPJ genes (Banerjee et al, 2013). Thus, the activation of the NOTCH1 signaling cascade in the absence of NOTCH1 mutations could be linked to the loss of EBF1. These considerations are now mentioned in the discussion.

46. Figure 7D - This figure shows significantly increased H3K27ac at promoters enriched for SP1, E2F, and KLF family members. The corresponding text describes ETS motif enrichment, which is not shown in the figure.

The claims described by the reviewer were removed from the manuscript.

Discussion.

47. CLL-specific large PMD's associated with repressive histone modifications and gene silencing are described, but confusingly are then associated with genes (IRF4 HIST1H1E, NOTCH1, IGLL5) that are mutated but also expressed in CLL. Are these two distinct classes of PMD's (associated with repression or alternately with expressed, recurrently mutated genes)? Needs clarification.

The reviewer's concern is valid that these example genes are rather exceptions within the PMDs. We investigated these genes in detail and found these to be only partially overlapping with the PMDs. We agree with the reviewer that these may not be the best representative cases. Therefore, we decided to remove this section from the discussion.

Supplementary data

47. Some of the supplementary data Excel sheets suffer from conversion of certain gene names to dates (e.g. SEPT11 has been converted to the date "Sep-11")

Thanks for pointing this out. The gene names in the excel lists have been corrected.

Reviewer #3:**General comments**

48. The identification of partially methylated domains in CLL is a very interesting observation, but the manuscript is crowded with other minor and sometimes dubious observations. There is a lot of valuable data in this study that needs publishing once it is better analysed with more rigorous curation of the data, but it cannot be published in this form.

We are glad to learn that this reviewer sees a lot of valuable data in our study. We have carefully considered his/her criticism. In several instances we have improved the data analysis as described below and corroborated our conclusions with additional experiments (GC2).

49. The methods used to identify high confidence regulatory elements are well below the required standard and need repeating with more stringent criteria for defining them. The enhancer identification dataset is flooded with background made up of huge regions of modified histones, probably representing entire domains or active chromatin. To be meaningful, the study should focus strictly on narrowly defined open chromatin regions (ATAC) linked to flanking histone regions. It should not define strings of modified nucleosomes as enhancers as this is not where the transcription factors are bound. Instead of being the starting point for identifying enhancers, the ATAC data is rarely used effectively, and this greatly undermines the whole analysis. Rather than coming first, ATAC appears in fig 5 in the context of footprinting, not to define enhancers in the first place. By focusing on histone modifications to define enhancers, the authors are actually identifying nucleosomal regions flanking open chromatin regions and not the enhancers themselves.

Defining enhancers based on an ChromHMM segmentation of histone modifications is a well-established approach that has been used in many studies, for example in the Roadmap Epigenomics Consortium (Roadmap Epigenomics et al, 2015) and in a recently published study on chromatin features of CLL (Beekman et al, 2018). Accordingly, we disagree that this type of analysis would be “well below the required standard” but rather consider it an important as a reference approach for the annotation of chromatin states. It is also noted that no consensus exists about the best way to map the “real” enhancer sites. Other studies, like those by the FANTOM consortium, use neither ATAC/DNase I nor histone modifications but base their enhancer assignment on bidirectional transcription and p300 binding (Andersson et al, 2014). While we fully agree that defining active enhancers based on the enrichment of H3K4me1 and H3K27ac results in rather broad regions we believe that it would not be justified to conclude that the resulting larger size of enhancers has no biological relevance. In fact the concept of super-enhancers with typical extension of ≥ 10 kb, while controversially discussed, explicitly claims that these particular large enhancer tracts have specific functions by (i) increasing cooperative binding of transcription factors (Loven et al, 2013), (ii) the local accumulation of H3K27ac reader proteins like BRD4 (Loven et al, 2013) or (iii) inducing a liquid-liquid phase separation process to form functionally distinct nuclear subcompartments (Hnisz et al, 2017; Sabari et al, 2018).

That being said the point of the reviewer is well taken with that it is also important to narrow down enhancers for certain types of analysis especially for the TF motif analysis (s. point 50 below) and to focus more on the ATAC-seq analysis. Accordingly, we now introduce the ATAC-seq analysis already in Fig. 1D and EV1D-F and extended our enhancer annotation with an ATAC centric approach as follows: Based on the extension of enhancer profiles reported previously (Chen et al, 2018) and excluding sites within promoters we select a +/- 1 kb window around either the ATAC peak center or the bidirectional start sites (bidirectional transcription is from our strand specific RNA-seq datasets of both short and long RNAs). The ATAC sites that overlap with an active ChromHMM enhancer state (E1 or E9) within this 2 kb region are used to define a set of “focused” active enhancers loci that represent 20-40% of the active ChromHMM enhancers. In this manner we now provide different enhancer sets, which we believe will all be valuable to other researchers. Furthermore, alternative strategies to map enhancers from a combination of DNA methylation, histone marks, ATAC peaks and bidirectional transcription can be implemented as desired with the data provided in our study.

50. The authors could expand the ATAC motif analysis to get the most useful information from this study, and leave out more dubious stuff. Some of the most valuable information is effectively lost in fig S5C. More convincing footprints at these motifs would be good.

We would like to emphasize that the TFs listed in Table S2 in the previous and revised manuscript were based on a motif analysis within the regions of differential ATAC signal. The other readouts were used to assign these loci with different types aberrant chromatin features in CLL. Thus, the requirement was that the motif was associated with both a differential ATAC signal and an aberrant chromatin feature and that the given TF displayed gain/loss of activity/gene expression. Additional motif analyses presented in the manuscript in Fig. EV2F and Fig. 3E are intersections with the ATAC-signal, while the analysis in Fig. EV4G and EV4H was conducted within the region of aberrant chromatin features, which is now clear from the figure legends.

51. The study equates modified chromatin domains with enhancers, and then counts up TF motifs in these domains, which is something you can only do with enhancers. They need to separate these entities.

See response to point 49 and 50.

52. The HDAC experiments are probably providing valuable information, but the gene regulation networks pointing to HDACs may not be real if the analyses were flawed. The authors may be trying too hard to make the analyses increasingly more intricate and more novel, but this runs the risk of losing sight of the basic biology, and moving away from our current definitions of what an enhancer is. An enhancer is more than a chromatin state as defined here. Overall there is a sense that the authors are using complex tools to extract more information than the data and the methods actually allow with any confidence. The end result is that the several good points of the study are lost in a morass of unreliable analyses and predictions based on these analyses.

To address this point for the HDAC experiment we have now simplified the analysis of this part and focused on the local increase of H3K27ac before and after inhibition. This is complemented by a differential RNA-seq analysis extracting genes that respond to treatment. The HDAC inhibitor-dependent TF motif analysis was removed and we now focus on making the following points: (i) HDAC activity is globally increased in CLL. (ii) It can be efficiently reduced to normal levels with panobinostat. (iii) The drug treatment leads to increased histone acetylation and a time dependent response on gene expression where the initial effect is counteracted by changes of RNA metabolism and chromatin acetylation related activity. (iv) The loss of H3K27ac at some CLL enhancers can be reverted by panobinostat treatment.

Specific

comments:

53. The manuscript contains too few examples of primary data showing specific gene loci and showing individual tracks for the individual patients. For 2 or 3 key genes it would be good to have e.g. the ATAC tracks for all samples where ATAC was performed. The one example shown in Fig 1A does not allow such a comparison, and as a 1 Mb window the resolution is too low to fully interpret the patterns. Finer resolution is needed to see what is being measured. Given the problems illustrated below, it would be helpful to have regions called as enhancers marked by bars under the tracks as the EXCEL file suggests that many are not true peaks but broad regions of histone modification and not the open chromatin regions needed to specific enhancers where factors are likely to be bound.

We have included several new tracks in the manuscript with finer resolutions as suggested. Enhancer / promoter regions are also marked as bars if appropriate (see Figures 1,3-5, Fig. S2,S3, S5). In addition, we refer to the additional tracks at GEO or at our web page as described above in GC 4.

54. The TCF4 locus in Fig 1A has sharp peaks of H3K4me3 in both the transcribed and non-transcribed state. This is a little unusual, and might raise questions about the specificity of the antibody. This modification is normally associated with active promoters. Are the authors suggesting that these peaks are instead poised or bound by Polycomb?

TCF4 has multiple transcription start sites and is transcribed at significantly lower levels also in non-malignant cells and thus the H3K4me3 peaks are consistent with that. The RNA track is hardly visible (see response to point 6). This has been addressed by presenting the RNA tracts at different scales.

55. The text refers to an enhancer-like region identified at the TCF4 locus in Fig 1A based on peaks of H3K4me1, but the profiles actually shows several broad stretches of H3K4me1 and H3K4me27, and not sharp peaks. It is hard to see how this data can be used to identify a discrete enhancer. It would have helped if this putative element was labelled in Fig 1A.

The ChromHMM is included in the Fig 1A giving the chromatin state and thus also the information about putative enhancer elements or active chromatin regions (in light and dark orange – please refer to the legend in panel Fig 1B). In addition, the element mentioned in the text has now been clearly labeled.

56. DNA regions should not be defined as "enhancers" based entirely on histone ChIP data. These "enhancer-like" regions can be defined as active chromatin regions, but should not be referred to as enhancers without more data, and not if there is no ATAC peak at the same site. E.g. One study found that only 26% of predicted enhancers had any actual enhancer activity (Kwasnieski JC, Fiore C, Chaudhari HG, Cohen BA. High-throughput functional testing of ENCODE segmentation predictions. *Genome Res.* (2014) 24:1595-602.). The text also refers to "genic enhancers" but it is unclear what distinguishes these as a group from poised and active enhancers. Does genic mean regions inside transcribed genes that also have H3K36me3? If so, it is a bit artificial to separate intergenic and exonic enhancers into 2 groups as they do the same things, and exon enhancers often do not regulate the gene where they reside, and may not then have H3K36me3.

The definition of "genic enhancer" is in line with definitions of chromatin states as defined by the ROADMAP consortia. It is the co-occurrence of the transcription elongation mark H3K36me3, the enhancer mark H3K4me1, and the active mark H3K27ac. These regions are not annotated by gene models and instead only use histone modification signals. Therefore, these are not annotated specifically to lie within genes, exon or introns or other gene features defined by gene models. For the other aspects of defining enhancers we refer to our response to points 49 and 50.

57. The data in table 8 lists 141,000 enhancers present in B cells and 238,000 enhancers in the combined data set. This is vastly more than you would reasonably expect to find in one cell type if meaningful criteria were used to delimit the data. A stringent analysis might identify 30,000 high confidence peaks that have the potential to be enhancers. It has become a problem in the field that some studies greatly over-estimate of the number of total enhancers in the genome (some claim more than 500,000) or in any one cell type. This arises by including too much low quality data or unsuitable definitions. It appears that the Roadmap consortium data cited here to validate the study also over-estimates numbers of enhancers, claiming 10% of the genome in B cells, so this is not helpful.

We agree with the reviewer that a focused enhancer set is needed as described in response to point 49 and 50. The comparison to the roadmap data, however, is still useful as the same histone marks in our study were used for the enhancer definition. Thus, the roadmap data set confirmed our data quality per se, even if the enhancer definition has to be more stringent.

58. In the present study the huge number of "peaks" indicates that insufficient culling of insignificant peaks has been performed, and the wrong criteria used to define enhancers. The table also suggests that normal B cells have over 1000 enhancers greater than 10 kb in length. This means that the authors are defining entire active chromatin domains, not discrete enhancers. At the very least the authors should ensure that a distinct open chromatin region is present at the site where they define an enhancer. These should be discrete ATAC peaks, typically ~ 200-500 bp across, defined on the basis of a minimum meaningful threshold. My analysis of a random selection of the 140,000 enhancers identified in B cells raised many concerns. A good example highlighting the problems are 2 enhancers defined by the authors at the NFATC2IP locus in hg19 as

16 28961200 28962000

16 28963000 28965000

If these are viewed on the ENCODE GC B cell and CD20 B cell DNaseI tracks it is clear that these represent modified histones on either side of a sharp DNaseI peak between these two coordinates. In this case the defined enhancers excluded the open chromatin region at the promoter in between.

Next to this promoter is a sharp DNaseI peak that is called as an enhancer at

16 28956400 28957800

In this case the called region is in open chromatin, but this is a discrete conserved CTCF binding site and is most likely an insulator, not an enhancer.

At the CD20 locus (MS4A1) the entire coding region and promoter is defined as a series of enhancers spanning 15 kb at

11 60220600 60226000

11 60227000 60231600

11 60232000 60232800

11 60234600 60235400

At this locus a very probable enhancer exists as a very sharp DNaseI peak downstream but is called as a 3 kb enhancer at

11 60398800 60402200

While it is not useful to focus on just a few specific examples, this does leave the reviewer with the firm conviction that the entire enhancer data set needs redefining with more stringent criteria to include discrete regions of less than 1 kb, that are validated as open chromatin, with minimum peak heights defined on an empirical basis, so that they have a good chance of representing true regulatory elements and not just broad zones of active chromatin. In the current data set, 61,000 of the defined enhancers are greater than 1000 bp whereas the typical enhancer should be ~300 bp.

It is unclear to us where the 300 bp size of a typical enhancer would come from. We have now included a more stringent list of enhancers based on an intersection of ATAC signal and active ChromHMM enhancer states. For the extension of “typical” enhancer regions we use the aggregated profiles for a set of crucial enhancer marks from a recent study (Chen et al, 2018), which yields regions of +/- 1kb around an ATAC peak or a bidirectional TSS (see response to point 49 and 68).

59. The enhancer data file should include at least some numerical data on peak values, so that it would at least be possible for the reader to interpret the significance of each peak, without having to re-process and upload the data on a genome browser. In this case, a link to a genome browser session would have been a good idea so that the reviewers and the end reader can see the data for themselves.

Bigwig files giving the enrichment of reads per region are available for further data analysis for all readers via the NCBI GEO accession number GSE113336. In addition, all the intermediate peak and ChromHMM segmentation files were shared for individual samples from the CancerEpiSys website (<http://www.cancerepisys.org/>).

60. The Hidden Markov modelling of chromatin states may not be valid if the identification of chromatin features is over-stated to start with.

The ChromHMM analysis was done in accordance to other studies (e.g., (Beekman et al, 2018; Roadmap Epigenomics et al, 2015). As outlined above in response to point 49 and 57, we consider it informative to relate our data in this manner to existing work as long as it is unclear what a “better” consensus approach would be.

61. In contrast to the claim in the text relating to fig 1D, it is not obvious that there is a significant increase in enhancers in CLL.

We apologize for the incorrect statement, which has been removed from the manuscript. As apparent from Fig. 1D there is in fact a loss of ATAC signal at enhancer regions in CLL which is in line with a globally increased HDAC activity.

62. Page 5 states that "we identified 1378 regulators ($p < 0.05$) with differential activity" which I think is far too many. This p value may mean that 5% of the 23,000 genes in the genome will be identified by chance, which is equivalent to 1150 genes, which could account for most of the regulators identified. I suggest that a lower p value cut-off would give more meaningful data. It would be better if they focused on e.g. the top 100 differential regulators for this to be meaningful.

We have expanded the relevant part of the Methods section to clarify this issue. For the construction of the B cell network we initially used a precompiled list of 5927 regulatory proteins (TFs, associated co-factors, signaling enzymes etc.) (Alvarez et al, 2016). From this list, 3862 proteins were present in the B cell network. With our RNA-seq data we could compute activities for 2804 of them and found 1378 regulators differentially active with an adjusted p -value < 0.05 . Thus, our reference list does not involve all genes but only 2804 regulators of which 5% would correspond to 140 proteins. Since we provide all relevant data in the xls file "Dataset_EV13-B-cell_network" other thresholds can be set as desired.

63. Fig 2D would benefit by adding an analysis centred on open chromatin regions near the histone peaks at the boundaries of PMDs. This may show that an active promoter or enhancer also blocks spreading of repressive marks, maybe more so than CTCF. It is not acceptable that ATAC is left out of the PMD analysis.

We thank the reviewer for the suggestion. We revised Figure 2D to include open chromatin regions in addition to histone peaks (Fig. EV2D). As pointed out by the reviewer, open chromatin regions show a higher signal around PMD boundaries. In addition, we have conducted ChIP-seq of CTCF and confirm our previous conclusion that it is enriched at the PMD boundaries (Fig. 2D). Thus, the simplest interpretation of our data is it that the enrichment of open chromatin at the PMD boundaries arises from bound CTCF.

64. In fig 2E the authors should indicate whether the increased mutations are occurring in PMDs that gained (or lost) meCG compared to normal B cells. If methylation is gained at these sites, it could be that meCG is transmuting to TG or CA during replication. How many of these mutations involve these bases?

Only 5% of the somatic mutations (Puente et al, 2015) were found to be on CG sites. In addition, PMD regions were mainly characterized by DNA methylation loss rather than DNA methylation gain. Overall, our analysis of dinucleotide frequencies revealed that the increased mutation rate is valid for all somatic mutations and is not specific to CpG dinucleotides.

65. The extension of H3K4me3 domains in fig 3A is rather minor and may depend on how this data is normalised. This analysis does not add much and is difficult to interpret. The claims made in 3C and 3D about gain of nucleosomes would need a high-resolution map of average nucleosome spacing for e.g. the first 5 nucleosomes from the TSS, or the centred on the first nucleosome. Fig 3D is only one example and is too low resolution to see anything meaningful.

Figure EV3 B has now been included that depicts the clustered nucleosome occupancy at all promoters. It can be seen that the bottom cluster shows an increased occupancy signal at the TSS, which reflects the profile of the promoters with the extended H3K4me3 signal depicted in Fig EV3A.

66. The link between the gain of one modified promoter nucleosome, and the gain of alternate promoters is greatly over-stated given that alternate promoters are many kb away in fig 3F, not 200 to 400 bp away.

We agree with the reviewer and we have removed the claims linking these two points.

67. In fig 4A the authors incorrectly use total Mb of chromatin as a way of defining enhancer quantities. This is totally inappropriate given the methodology used. An enhancer should be an entity at an open chromatin region, not a vast expanse of modified chromatin. For example, a 10 kb stretch of transcribed modified nucleosomes should not be called an enhancer, but this is what some loci in the attached enhancer table look like. This figure needs to state the number of discrete open chromatin regions with the right flanking modification, not Mb of chromatin of highly variable lengths.

The whole discussion on page 8 becomes invalid once the reader realises that discrete rigorously defined enhancers were not identified in the first place, so I will not comment on it at length. You cannot do a motif analysis of chromatin regions 1 to 10 kb long. There are many of these, and some are much longer. Motifs concentrate within 100 to 150 bp of an ATAC peak summit, a much smaller region than most enhancers defined here. The background must be huge.

As described in response to point 49 and 50, the TF motif analysis was conducted within the region that carry a differential ATAC signal. As a background unchanged ATAC-seq regions were used as stated in the method section.

68. I assume that some of the super-enhancers will also include the tracts of ~50 kb of modified chromatin that are in some places defined as single enhancers. This is not a great definition.

In our study we follow the approach introduced previously using the ROSE software tool as stated in the manuscript (Loven et al, 2013; Whyte et al, 2013). This leads indeed to some super-enhancer regions of ~50 kb. We refer to our response to point 49 for this issue. We would also like to note that recent studies (Hnisz et al, 2017; Sabari et al, 2018) propose that enhancers assemble into a distinct nuclear subcompartment formed by a liquid-liquid phase separation process. Such a mechanism to form “chromatin bodies” would require large chromatin regions that establish multivalent interactions as opposed to bivalent bridging interactions between specific binding sites as discussed recently (Erdel & Rippe, 2018).

69. Fig S5A cannot be interpreted without an inbuilt legend to the graphs. The ATAC data should show average profiles for CLL-specific ATAC peaks in CLL and in normal B cells. Fold change is not enough. We need an indication of peak height and width. The data here is over-processed to the point where you know longer know what it looks like. Where is some primary data? It needs stating if the X axis in S5A is a natural or a log₂ scale. If they are using a 1.13 threshold, this has to be a log scale = ~2.3 actual difference.

The figure has been moved and updated as requested to Fig EV1 D. Primary ATAC-seq data can be found throughout the manuscript, e.g., in the new Fig EV1F and in Fig 3 and 4.

70. Fig 5B does not necessarily show TCF occupancy, it more likely shows sequence specific cleavage around a central TCA motif. This pattern is characteristic of the sequence bias of nucleases. Footprints normally look very different and are wider with a deeper drop at the footprint. The EBF data is less clear but this may also be influenced by sequence specificity which could explain the pattern. If we do not know how rigorous the calling is, it is hard to have confidence in 5C and 5D. A simple table of actual motifs identified, and the % occurrence would help to make fig S5C more meaningful.

In the case of EBF1 we have performed ChIP-seq to validate our findings as illustrated in Figure 6. The information requested by the reviewer about the motifs is comprehensively displayed now in Figures 4A and 4C including the % of sequences with the corresponding motif. We have now stated in the text that the transposase has a sequence bias which is a caveat of the method, nevertheless we are confident that the differences depicted in the plot are valid.

71. I find it hard to know whether the gene regulation pairs defined by co-occurrence of sequence reads in scATAC are reliable or not. This is an interesting approach, but it is hard to see how they can reliably get the pairing right. In a uniform population of cells, expressing the same genes, there is no reason why they would detect an active promoter/enhancer pair in one cell and not another. It

seems there is a lot of room for error here. These pairings should be mostly within 100 kb, rarely more than 1 Mb apart, but the pairings shown in S5E are typically 100 Mb apart. This seems to confirm that the method is not very reliable.

We have extended the “quality control” of our scATAC-seq analysis in Appendix Fig S5. First, we are convinced of the high quality of our scATAC-seq data as reflected by the high number of integrations per cell and the excellent agreement of the aggregated data with the bulk ATAC-seq profile. As discussed above in the context of reviewer comment 28, scATAC-seq data are qualitatively different from chromosome conformation capture data because correlations can arise from both physical contacts *in cis* or from co-regulation *in trans* and reflect a bona fide active state in contrast to 3C based interactions. That being said, it is noted that 68% of the promoter-enhancer pairs identified from our scATAC-seq are also found in the 4D nucleome database (Teng et al, 2015), which further corroborates the validity of our analysis.

72. On page 10, the authors seem to be using differential footprinting to assess CTCF occupancy. Differential occupancy analysis here would actually be much more reliable if they simply mapped ATAC sites at known CTCF sites or motifs. An insulator has one or more sharp ATAC peaks with a single CTCF motif in it. Footprinting is not needed here and is more likely to generate false differentials, because footprinting has more limitations than ATAC peak calling.

Our data set now includes EBF1 and CTCF ChIP-seq data, which validate our ATAC-seq analysis. As depicted in Figure EV5 and associated text, the conclusions on CTCF occupancy derived from the ATAC-seq data fully validated CTCF ChIP-seq results. Based on the CTCF ChIP-seq analysis CTCF is lost at 5964 sites and gained at 441 sites in CLL with an overlap of 55% and 47% of the sites, respectively, as identified also in the differential ATAC-seq analysis.

73. The motif analysis in fig 7E is not valid, because the enhancers were not correctly identified. Stretches of modified nucleosomes in coding regions should not have more motifs than random sequence. It is possible that the analysis ends up biased towards simple motifs that exist everywhere. To interpret this, a motif table is needed with the actual motifs and with % found, and the predicted % background values shown. P values become unreliable when dealing with what seems to be 10% of the genome because of the vast number of motifs found.

As stated in response to point 52, the TF motif analysis has been removed.

74. On page 4 and on page 11 the authors seem to want the region downstream of TCF4 to be both an enhancer and the co-regulated LINC01929 gene. This may actually be an enhancer-derived transcript. Maybe this should be clarified, and the predicted enhancer better defined.

We have now marked the predicted enhancer downstream of TCF4 in Fig 1A with only a small part overlapping with LINC01929. The weak RNA signal that is detected in the CLL rescaled RNA-seq track is also outside of this gene.

75. For the above reasons, the 231 Mb of novel genic enhancers (8% of the genome) referred to on page 13 are more likely to be just active gene coding regions and not enhancers. The authors will need to redo their analyses of histone modifications and ATAC peaks to separate active genes (with e.g. H3K27Ac) away from promoters and enhancers defined as open chromatin regions.

See response to point 49 and 50.

Minor comments

76. The citations to Puente et al and Horton et al are incomplete.

These errors have been corrected.

77. The link to MACS seems to be the wrong address.

This error has been corrected.

78. Hyphens are missing in the protein names for IL-4 and NF- κ B, and kappa should be a Greek letter, and Sp1 and Sp2 have p in lower case.

We have corrected the spelling of NF- κ B when we refer to the protein complex. For the individual proteins we use the HGNC nomenclature (all letters in uppercase, gene in italics, no hyphens).

79. The methods text and the data files should include information as to which build of the genome is being used to define chromosomal coordinates. This seems to be hg19.

This information is now included in the method section.

80. The heat map in fig 2A should have a colour code.

A color code has been added as suggested.

81. On page 7, "B-cells lost H3K4me3 in CLL (Fig. 3H, I, see also Fig. 1C, state 7)" should refer to 1B.

This error has been corrected.

82. The description of the nucleosome positioning method is inadequate, referring simply to H3 libraries. The methods should specify MNase-H3 ChIP-Seq as in the results section.

The method section has been updated accordingly.

83. The vertical legend on fig S2I should be non-malignant, not "on malignant".

This error has been corrected.

84. Fig 3K is not cited on page 7.

Figure 3 has been revised and all panels are now cited in the text.

85. On page 11 a reference to figs 7C, S7A should be to 7C, S7B.

Figure 7 has been revised and is now integrated in Figure 5. All panels are now correctly cited in the text.

86. The description of Fig 7D on page 12 is incorrect as promoters with Sp and KLF motifs are changing, not ETS motifs.

Figure 7 has now been into Figure 5 and the motif analysis has been removed.

References

- Alvarez MJ, Shen Y, Giorgi FM, Lachmann A, Ding BB, Ye BH, Califano A (2016) Functional characterization of somatic mutations in cancer using network-based inference of protein activity. *Nat Genet* **48**: 838-847
- Andersson R, Gebhard C, Miguel-Escalada I, Hoof I, Bornholdt J, Boyd M, Chen Y, Zhao X, Schmidl C, Suzuki T, Ntini E, Arner E, Valen E, Li K, Schwarzfischer L, Glatz D, Raithel J, Lilje B, Rapin N, Bagger FO et al (2014) An atlas of active enhancers across human cell types and tissues. *Nature* **507**: 455-461
- Banerjee A, Northrup D, Boukarabila H, Jacobsen SE, Allman D (2013) Transcriptional repression of Gata3 is essential for early B cell commitment. *Immunity* **38**: 930-942
- Beekman R, Chapaprieta V, Russinol N, Vilarrasa-Blasi R, Verdaguer-Dot N, Martens JHA, Duran-Ferrer M, Kulis M, Serra F, Javierre BM, Wingett SW, Clot G, Queiros AC, Castellano G, Blanc J, Gut M, Merkel A, Heath S, Vlasova A, Ullrich S et al (2018) The reference epigenome and regulatory chromatin landscape of chronic lymphocytic leukemia. *Nat Med* **24**: 868-880
- Chen H, Li C, Peng X, Zhou Z, Weinstein JN, Cancer Genome Atlas Research N, Liang H (2018) A Pan-Cancer Analysis of Enhancer Expression in Nearly 9000 Patient Samples. *Cell* **173**: 386-399 e312
- Erdel F, Rippe K (2018) Formation of Chromatin Subcompartments by Phase Separation. *Biophys J* **114**: 2262-2270
- Hnisz D, Shrinivas K, Young RA, Chakraborty AK, Sharp PA (2017) A Phase Separation Model for Transcriptional Control. *Cell* **169**: 13-23
- Loven J, Hoke HA, Lin CY, Lau A, Orlando DA, Vakoc CR, Bradner JE, Lee TI, Young RA (2013) Selective inhibition of tumor oncogenes by disruption of super-enhancers. *Cell* **153**: 320-334
- Miele L (2011) Transcription factor RBPJ/CSL: a genome-wide look at transcriptional regulation. *Proc Natl Acad Sci U S A* **108**: 14715-14716
- Poos AM, Kordaß T, Kolte A, Ast V, Oswald M, Rippe K, König R (2018) Modelling TERT regulation across 19 different cancer types based on the MIPRIP gene regulatory network approach *Bioinformatics*: under review
- Poos AM, Maicher A, Dieckmann AK, Oswald M, Eils R, Kupiec M, Luke B, König R (2016) Mixed Integer Linear Programming based machine learning approach identifies regulators of telomerase in yeast. *Nucleic Acids Research* **44**: e93-e93
- Puente XS, Bea S, Valdes-Mas R, Villamor N, Gutierrez-Abril J, Martin-Subero JI, Munar M, Rubio-Perez C, Jares P, Aymerich M, Baumann T, Beekman R, Berver L, Carrio A, Castellano G, Clot G, Colado E, Colomer D, Costa D, Delgado J et al (2015) Non-coding recurrent mutations in chronic lymphocytic leukaemia. *Nature* **526**: 519-524
- Roadmap Epigenomics C, Kundaje A, Meuleman W, Ernst J, Bilenky M, Yen A, Heravi-Moussavi A, Kheradpour P, Zhang Z, Wang J, Ziller MJ, Amin V, Whitaker JW, Schultz MD, Ward LD, Sarkar A, Quon G, Sandstrom RS, Eaton ML, Wu YC et al (2015) Integrative analysis of 111 reference human epigenomes. *Nature* **518**: 317-330

Sabari BR, Dall'Agnesse A, Boija A, Klein IA, Coffey EL, Shrinivas K, Abraham BJ, Hannett NM, Zamudio AV, Manteiga JC, Li CH, Guo YE, Day DS, Schuijers J, Vasile E, Malik S, Hnisz D, Lee TI, Cisse, II, Roeder RG et al (2018) Coactivator condensation at super-enhancers links phase separation and gene control. *Science* **19**: eaar3958-3917

Teng L, He B, Wang J, Tan K (2015) 4DGenome: a comprehensive database of chromatin interactions. *Bioinformatics* **31**: 2560-2564

Whyte WA, Orlando DA, Hnisz D, Abraham BJ, Lin CY, Kagey MH, Rahl PB, Lee TI, Young RA (2013) Master transcription factors and mediator establish super-enhancers at key cell identity genes. *Cell* **153**: 307-319

2nd Editorial Decision

19th December 2018

Thank you again for submitting your work to Molecular Systems Biology. We have now heard back from the two referees who agreed to evaluate your study. I apologize for the slow process, which was due to the late arrival of one of the reports and the need to perform a consultation with the reviewers regarding the remaining issues. As you will see below, the reviewers appreciate that the performed revisions have improved the manuscript. However, they think that several issues remain and as such, we would ask you to address them in a major revision of the manuscript. Typically, our editorial policy is to restrict major revisions to a single round. However, since in this case the reviewers do not request additional experiments and considering that most of the requested additional analyses sound feasible, we would like to offer you a chance to revise the study.

In brief the most fundamental issues that need to be addressed are the following:

- ChromHMM needs to be used with caution when defining enhancers, since strictly speaking it is a tool for detecting genomic states but cannot be used on its own to define enhancers. The reviewers provide constructive suggestions on how to use additional features to define enhancers.
- The motifs used for the motif analysis need to be better specified and shown in the manuscript. Moreover, the TF regulator analyses need to be performed with more stringent criteria and perhaps in more restricted genomic regions, to avoid false positives. Both reviewers provide constructive suggestions in this regard.
- The super-enhancer calling needs to be refined.
- Reviewer #1 also recommends performing further analyses to better support the proposed promoter-enhancer interactions. As the reviewer mentions, confirming the interactions by Hi-C or 4C would be rather challenging, but s/he proposes alternative options for analyses to address this point.
- As a side note, we would recommend further emphasizing the resource value of the study (which was also explicitly acknowledged by reviewer #2 in the initial round of review) e.g. by adding a sentence in the abstract.

All other issues raised by the reviewers would need to be convincingly addressed. Reviewer #3 raises strong concerns mainly regarding the enhancer calling and the motif analysis. We think that all these issues can be addressed by following the recommendations of reviewer #1. During our cross-commenting process (in which the reviewers are given the chance to make additional comments, including on each other's reports), reviewer #1 provided further comments and recommendations on how to address the concerns of reviewer #3 on these two aspects and I pasted below for your reference (see REVIEWER #1 (comments during the referee cross-commenting process)). Please do let me know in case you would like to discuss further any of the comments of the reviewers. We could also have a phone call if you think this would be useful.

REFEREE REPORTS.

Reviewer #1:

Main comments

The authors have substantially improved the organization of the manuscript and consecutive sections are now better connected. The integration of the scATAC-seq with the GRN is interesting,

but the number of identified regulators is very high, suggesting that the integrated method introduces many false positives. The TP's in the ROC curve (shown in the rebuttal) are also based on motif analysis of the same data, rather than true known regulators from the literature, thereby providing little evidence for the predictive accuracy of the method.

The authors made an effort to validate the interactions derived from co-accessibility in scATAC-seq with data from the 4D nucleome database. Unfortunately, no 4C-seq experiments (as suggested) were included for validation. Also, ChIP-seq experiments for CTCF and EBF1 were performed in CLL patient and healthy controls. Taken together, the additional experiments are sufficient to validate some of the results, but were not used for a genome-wide verification. Instead, ChIP-seq analysis is mainly limited to a few examples and the analysis of the newly derived data raises some questions and is not described in the methods section.

The manuscript will substantially benefit from a genome-wide comparison of some of the model/epigenetic data derived hypothesis with the 4D interactions/CTCF and EBF1 ChIP-seq. This could be addressed in a few figure panels. See specific comments below.

Specific comments

page1) 1,378 differential regulators (TF's and chromatin modifiers) out of 2,804 were detected between CLL and healthy controls. This number is extremely high and likely includes many false positives. Can the authors compare these results with some of their collected data to support this claim? E.g. how many of these regulators are (highly) expressed or differentially expressed in/between CLL and healthy controls. How many of the TF motifs are enriched in the differential ATAC-seq peaks compared to all ATAC-seq peaks?

Page 8) Superenhancers (SE) are called using the ROSE program by stitching neighboring TF binding sites that have overall enriched H3K27ac. This means that broad H3K27ac domains need to have multiple TF binding sites in order to be called a "super-enhancer". In the original manuscript (Whyte et al 2013), the authors used Med1 and H3K27ac. Here, the authors could use the ATAC-seq peaks and H3K27ac. By calling every H3K27ac region larger than 10kb (which is not uncommon for some promoters) a SE, many false positives might be introduced. Furthermore, the authors may consider removing SEs that overlap a TSS, to make sure these are genuine enhancers. Lastly, are the 6.3% and 5.5% of the SE unique to the study (compared to what then?) or unique to the CLL/healthy control? The latter makes more sense, and it would be of interest to perform a differential motif analysis in the ATAC-seq peaks of the differential SE's, using all (union CLL and healthy) ATAC-seq peaks within a SE as control.

Page 10) CTCF binding was lost at 5,964 sites and gained at 441 sites. Such a dramatic change in CTCF occupancy within the same cell type is a spectacular result, but may indicate a flaw in the analysis. Are these genuine CTCF binding sites (e.g. called with MACS2 and $p < 1e-8$) that have significant changes (e.g. Diffbind or DESeq2) in occupancy? How many CTCF binding sites were detected in total? Figure EV5B indicates sites with very low CTCF coverage, which are likely background. Please add a description of the analysis (similarly for EBF1) to the methods section.

Page 11) 90% of the enhancers had CTCF stably bound in both cell types. While CTCF certainly overlaps with promoters and enhancers, 90% is a lot. Are the authors sure that these are genuine CTCF binding sites (see comment above)?

Page 11) We found that 68% of these pairs were also listed as spatial contacts in the 4D nucleome database (Teng et al, 2015), suggesting that many promoter-enhancer pairs involve physical contacts. In total 3955 promoter-enhancer pairs were identified, with most promoters being connected to one enhancer at mean and median distances of 32 kb and 20 kb (CLL) and 23 kb and 10 kb (non-malignant B cells), respectively".

The average promoter-enhancer distances are very short and it will be hard to validate these with 4C-seq or Hi-C given the high background from proximity ligation at this distance from the promoter viewpoint. Therefore, it is surprising that 68% of these interactions could be validated. Furthermore, an upper limit of 100kb might be too short. The authors should consider the significant interactions derived from a (B-cell specific) 4D nucleome data sets and set their lower- and upper bound distances to e.g. 10% and 90% of the significant interactions in the 4D nucleome data set. Finally, the authors should describe which datasets were taken from the 4D nucleome, whether these are B-cell specific (or closely related) and how the analysis and comparisons were performed (methods section)

The authors suggest that the correlation in scATAC-seq is predictive of (CTCF) looping, but provide only one example. Given that a relatively high number of differential CTCF sites were found, one would expect a coordinated down-regulation in the scATAC-seq signal of CTCF sites that interact and are downregulated in CLL patients relative to control. Can the authors e.g. show a scatter plot where the correlation in the ATAC-seq signal decrease of two interacting loci is associated with the loss of CTCF occupancy in these two loci, for every differential CTCF binding site overlapping the

scATAC-seq signal? A genome browser view of the ATAC-seq and CTCF signal for an exemplary locus would be a nice illustration for the supplement to really reinforce this point.

Similarly, the EBV1 ChIP-seq was not/barely used to validate some of the results. How many peaks were found in CLL and healthy controls? How many were differential? Were these associated with gain/loss in ATAC-seq and H3K27ac at promoters or enhancers? Were these enhancers connected to the target genes that were predicted from the model and to what extent?

Fig 6G (referred to in the rebuttal) does not exist

CTCF and EBV1 and 4D nuclease analysis are missing in the methods section.

Discussion

"Our integrative analysis .. can explain more than 80% of the transactional variance of CLL cells". This statement indicates that e.g. a multivariate regression using all the epigenetic signals as predictors can explain the RNA-seq change between CLL and healthy subjects with an adjusted R^2 of 0.80. If this is the case, the authors should show that. However, such an accurate prediction of gene expression changes is typically not feasible, even with the high-quality integrative dataset the authors collected. In that case, the statement should be adjusted.

Transcription factor motif analysis

"In addition, only TFs were included that showed a significant differential protein activity (or gene expression for network target genes) as computed from our B cell specific transcriptome network (Fig 1D)"

How do the authors compute protein activity from RNA-seq data?

Text/typos

- while the loss at enhancers might indicate a loss OF enhancer activity (page 1)
- "PMD's were enriched .. conformation capture". Please edit the sentence.
- Infexion point → Inflection point (legend Fig. EV1)

Reviewer #3:

The revised manuscript is improved, but not yet to the point of being acceptable. Some of the deficiencies relate to terminology and can be corrected by reclassifying chromatin features under a different name. However, there are many substantial fundamental defects in both the approach and the analysis in this manuscript which relate primarily to:

Using an inadequate method to define enhancers: The ChromHMM pipeline is used extensively by the Roadmap Epigenetics Consortium, the authors of ChromHMM, but in my view this is a highly unreliable method to use to predict enhancers as it is not centred on any analysis of open chromatin regions. Based on the results of this manuscript it is highly likely that a large fraction of the predicted enhancers or changes in enhancer state relate more to changes in the activities of transcription units than enhancer activity. It may be relevant that ChromHMM was a default track on the hg18 UCSC genome browser, but not on the more recent hg19 and hg38 browsers, maybe because it has gone out of favour as a useful tool. The main flaw in this approach is that active genes are essentially covered in varying degrees of H3K4 me1, me2 and me3 and H3 K27 ac, as follows: H3 K4me3 concentrates +/- ~1 kb, K4me2 extends a few more kb and K4me1 extends over most of the gene. While H4 K27ac is highest on either side of the promoter, it extends over the whole gene. H3K36me3 is concentrated on the 3' half of genes. This means that the three different ChromHMM enhancer states, especially genic enhancer and active enhancer 2, will often represent transcribed chromatin and not enhancers. Without alignment with well curated ATAC data, which excludes minor ATAC peaks associated with active nucleosomes, the ChromHMM data has no predictive value. This is abundantly clear in fig 1D, which does at least filter for overlap with ATAC data, but shows that the modifications predicted for active enhancers (enhancer 1 + H3K27ac) are in the minority. The biggest enhancer groups are defined on the basis of either "active enhancer 2" which is just H3 K4me1, found at the 5' half of genes, or "genic enhancer" which is H3 K36 me3, K4me1 and K27ac, found at the 3' half of genes. My original perusal of the table of 238,000 enhancers suggested that these often comprised whole coding regions. The enhancer state 1, marked by K4 me1 and K27ac, which should be found flanking enhancers but not at the enhancer itself, is the most useful group included. However, this is a tiny fraction of the population of peaks in fig 1D. For confirmation of these views, I suggest that the editor and authors to look at Fig S4 of Wang et al Nature Genetics 40, 897 (2008) and Fig 4 of Barski et al Cell 129, 823, (2007) which were global studies of many histone modifications. The average gene profiles show K27ac, K4me1 and K36me3 on typical transcribed regions. This cannot be used to define enhancers. This is also why many "super-enhancers are defined on regions that include transcribed genes.

Including ATAC data in only some of the analyses performed: Most studies should start by mapping open chromatin, looking at adjacent chromatin marks, and identifying motifs centred on open chromatin. The only time such an analysis is presented is in fig 6E, the final figure in the paper, and

that shows one motif linked to one ATAC peak. For most of the paper, the criteria described on p22 for inclusion as an enhancer are just 2 of 6 patterns that do not have to include open chromatin, which should be obligatory. This leads to swamping of the data pool with active chromatin marks commonly found on active genes. Worse than that, most of the 238,000 active chromatin regions do not appear to contain ATAC peaks, meaning that any motif analysis is being performed on nucleosomes, not enhancers, and is meaningless. Furthermore, if 2/6 chromatin modifications are needed to define enhancers, this excludes enhancers at open chromatin regions where TFs are bound where there are no histones. This means that true enhancers and enhancer state 1 are mutually exclusive. This is also evident in the table of 238,000 enhancers, where some actual potential regulatory elements are excluded.

It is significant that a recent study published by Ott et al on the same subject this month in Nature Genetics used just ATAC and H3 K27ac, and came to some different conclusions. It is at least apparent that the target genes identified by Ott et al seem to be different to those identified here. Flawed motif analysis: From the outset, the construction of a gene regulation network relies on motifs, but the motif table in table S2 is full of errors. If these motifs were used in the motif analyses, then all these analysed are fatally flawed, as follows: (i) The NFAT motif listed is AAGAAGAAA. This will never bind NFAT which binds (A/T)GGAAA, (ii) the authors are confused by the TCF nomenclature whereby for example human TCF3 and TCF4 belong to the HLH E2A CACCTG E-Box class whereas mouse Tcf3 and TCF1/LEF1 are HMG proteins that bind CAAAG. The authors imply that they are including both motifs in their E2A searches, (iii) the wrong RUNX motif is used. The right motif for RUNX proteins is TGTGGT(T/C)(T/A) without upstream CCCC which will greatly skew the analyses, possibly making it 200x less likely to hit the basic motif, (iv) The FOX motif is not right, and at the least should start with G/A and not GA, and (v) the NF-kB motif is better represented by The NF-kB motif is best represented by GGGGAAATCCCC (which has an extra A).

Another fatal flaw, explained above, is that most motifs defined in this study are defined based on groups of modified nucleosomes, where there are no TFs, and the analyses may actively exclude enhancers bound by TFs where there are no modified histones.

Specific comments:

- (1) Regarding terminology, many objections are easily resolved by replacing every single reference to "enhancer regions" on every single occasion, without exception, with a reference to "active chromatin Regions", and also in e.g. Fig 1B, 1D etc etc.
- (2) The main persistent problem that remains is the underlying assumption throughout the manuscript that enhancers can be defined as broad regions of modified chromatin. They must by necessity be defined as small regions of 200-500 bp of open chromatin bound by TFs. This authors methodology sidesteps the most important fundamental concept of gene regulation which is that the genome is reprogrammed by TFs which establish open chromatin regions. The modifications came later. In the case of changes in DNA methylation, sometimes much later.
- (3) Following on from above, any analyses of motifs should focus on +/- 200 bp at most of an ATAC peak. In their response to this suggestion in the original review, the authors question where my previous definition of enhancers as ~300 bp regions comes from (point 58). The answer is very simple and based on our fundamental understanding of mechanisms of gene regulation. For essentially every enhancer defined in the genome, enhancers are occupied by nucleosomes in the inactive or poised state, and are nucleosome-free once TFs have evicted 1 or 2 nucleosomes. 1 nucleosome + 2 linkers occupies $40 + 160 + 40 = 240$ bp. This is why DHSs are typically 250- 300 bp of open chromatin bound by TFs. The TFs do not as a whole bind outside this zone where chromatin is inaccessible. This is why motif analyses must be of ATAC peaks and not H3 peaks. In their rebuttal, the authors say they look at motifs +/- 1 kb from an ATAC peak (point 58). This will create a massive background as it adds 1700 bp of nucleosomal DNA to each 300 bp enhancer regions, where most motifs are +/- 100 bp, the window the authors use in Fig 5D. Why not use the same window to define motifs and thereby get better enrichments,
- (4) All analyses included in this study should be restricted to those samples that included ATAC, which is most, and always use ATAC as one of the two criteria used to define regulatory elements. This still allows for 3 normal samples where both ATAC and ChromHMM are available, which should be sufficient given the differences between these and the 18 CLLs that can be included.
- (5) ATAC should be at the top of the flow diagram in fig S1B, and not on the forth tier where it sits now as an afterthought.
- (6) A major technical deficiency in this manuscript is the inadequate analysis of the TF motifs underlying the chromatin patterns which are presented in great detail. Analyses of motif occurrence is restricted to meta data showing probabilities etc. but not the primary data. To have confidence in the enhancer definitions it is necessary to plot motif positions on the plots of ATAC peaks in the style of the analysis shown in fig 5B. To verify enhancers adequately, the same figure should include the profile of ATAC peaks centred on a window +/- 1 kb, ranked by ATAC enrichment, with flanking histone modifications, and confirming the presence of motifs at the centre of the peaks more enriched in the enriched ATAC peaks. None of this has been done. Fig 5B is aligned on motifs, not ATAC peaks, and even then uses a flawed NFAT motif. The authors should show the matrix logos and provide the PWMs used in this study. If these differ to the same motifs annotated on HOMER, these need to be shown alongside. It would be preferable to simplify the number of motifs depicted in figures such as 3E by restricting

the motifs depicted to a single representative example in cases such as ETV/ELK/ETS and KLF where they all bind the same motifs. This figure probably needs ETS, KLF, E-box, SP1, NFY, NRF and no more. Fig 5C only needs one AP-1 and one RUNX motif etc.

(7) As stated above, there is an over reliance on the ChromHMM gene prediction tool, released in 2010, which has not really gained wide acceptance in the gene regulation field as a front line approach for identifying enhancers. Published evidence from groups such as the Hardison and Cohen group suggests that enhancers should be defined on the basis of transcription factors or on function. Most studies measuring actual enhancer activity find ~10% of active chromatin regions have enhancer activity. While it is helpful to know about chromatin states in the context of gene regulation, over-use of this tool is likely to lead to misleading results. ChromHMM seems to tell us more about whether a region is transcribed or not.

The ChromHMM browser tracks often show enhancers and promoters as 10 kb tracts of chromatin spanning open chromatin regions. As an example, ENCODE shows the first 50 kb of the B cell gene PAX5 at hg18 chr9:36,977,389-37,029,852 is covered by either H3 K4 me3 or me1 in lymphoblastoid cells and is shown as either enhancer or promoter region on the Broad Institute ChroHMM track in the same cells. This is highly misleading. Just because the current authors get the same patterns as the ChromHMM tracks loaded on the UCSC genome browser does not mean that this tool is suitable to use as the front line approach to defining enhancers. It would in fact be a disservice to the gene regulation field to give de facto acceptance of this methodology by prominently presenting it as a state of the art approach to be used to predict enhancers. It is not, it is an adjunct to define the neighbourhood of enhancers.

To illustrate another feature of histone profiles, some ChromHMM data nicely show a gap between the K4me1 and me3 marks, just where K4me2 would be predicted. This is what is expected in a transcription unit, but ChromHMM would show it as a gap between chains of enhancers. Page 8 line 15 refers to transitions from the quiescent state to the K4 me1 E2 state, and from the transcribed K36me3 state to the K36me3/K4me1 genic enhancer state. Both represent gain of K4me1, which may just reflect differences in sensitivity of K4me1 detection, and in the second case depends on an ill-defined boundary between states seen at the 5' and 3' halves of genes. There is plenty of room for over-interpretation of the data here.

(8) The opening figure, Fig 1A, is poorly described and has inconsistencies. The sentence "the histone modifications identified a downstream enhancer, which was activated in CLL cells as judged from the enrichment of H3K4me1 and H3K27ac (Fig 1A)" should be turned the other way around to say that "a potential enhancer region was identified far downstream of the TCF4 gene as a cluster of ATAC peaks embedded within an ~ 60 kb region of chromatin enriched in H3K4me1 and H3K27ac in CLL cells". This approach needs to be taken throughout the manuscript whereby any description of a potential regulatory region must begin with a description of the ATAC data.

The 2 RNA-Seq scales in Fig 1A should have some degree of equalisation across the genome, as it is not acceptable to show normal on a scale of 100 and upregulated on a scale of 8000, as it suggests the opposite of what it is. For example, a uniformly active region of the genome could be used to either normalise the FPKM data. Alternatively the authors could adjust the scales to make them equivalent, and then maintain the same ratio in the scales. Furthermore, (i) some of the ATAC peaks at TCF4 look bigger in normal cells, and (ii) the whole TCF4 locus is plastered with H3K4me1 and H3K27Ac in CLL. Does that mean the whole TCF4 locus is one big super-enhancer? Because the whole gene is involved, this data could be actually be used to undermine the very concept of the "super-enhancer", a process that is already happening in other studies that dispute the relevance of this definition (eg at the alpha globin locus, and in Barakat et al recently in Cell Stem Cell).

(9) The significance of the PMDs is unclear. They were defined at 10 kb resolution which is too high to be meaningful. Either 500 or 1000 bp by would be better, as a 10 kb region can contain one 10% and one 90% methylated region. Fig 2A shows a highly variegated pattern of methylation, so maybe this is indeed the case. It would be better to show verification with single clones from bisulphite sequencing of a few hundred bp where meCG is traditionally shown as black dots in a series of clones to reveal the trend. The 50% could also be a population average. Basic mechanisms of activation and repression tend to drive the unstable 50% meCG state one way or the other, so 50% is unexpected at one discrete region.

On p8 line 11, when the authors say PMD loss and meCG loss correlate with loss of enhancers, I suggest these changes simply correlate with gain in transcription.

(10) the data on nucleosome changes at promoters and could be left out. This section does not provide any clear insights. The data is not convincing.

(11) Some major conclusions in this study are based on the flawed motifs TCF4 (at the centre of the network) and NFAT (e.g. 1E, 5A, EV2F).

(12) The concept of "superenhancers" remains controversial and has not been widely accepted in the basic gene regulation community, even though many groups are now trying to define these regions. Most problematic with this concept is that many of the superenhancers defined include whole transcription units and promoters flanked by enhancers and other regulatory elements. This configuration was originally defined as the "active chromatin domain", and the clusters of enhancers were defined as LCRs. In my view the redefinition of these as superenhancers is a retrograde step. In the original paper, Whyte et al defined 231 superenhancers as a small group of highly specialised LCRs. Even their prime examples depicted in the figures include whole transcription units. This is

also true of two of the three prime examples shown in the present study in fig 4I where the CDKN1a and KLF13 genes make up a big part of the superenhancers. The current authors may also be a bit over-zealous as they defined 829 superenhancers, 4x the expected number.

In their rebuttal (point 49 and 68) the authors use recent descriptions of "liquid phase compartments" from the Sharp lab as evidence that superenhancers are meaningful. I would suggest that this new definition is another way of restating the fact that active genes reside within active chromatin domains, which are more accessible, dynamic and interactive. This concept has been around for decades, and like LCRs and superenhancers, has been reinvented. The study Barakat et al also show most regions within a "superenhancers" do not have enhancer activity. The authors can claim that active chromatin domains enhance TF recruitment at localised enhancers, but not that TFs bind cooperatively all across domains > 10 kb. This is a fallacy.

(13) Fig 6A confirms that most upregulated "enhancers" are marked by H3 K4me1 and not K27ac. This should be seen as a warning sign that these are poorly defined.

(14) In fig EV4a the authors need to refer to Mb of active chromatin, not enhancers.

(15) As mentioned in the original review, the patterns for TCF4 and EBF1 in Fig 5D do not resemble footprints but reflect sequence specificity within a ATAC peak centred on a sequence. Only CTCF shows a true footprint. This figure also highlights the error in the TCF4 motif. This should be the TCF3/TCF4 motif CACCTG, but the authors show the motif for TCF1/LEF1 which is CAAAG. This confirms that the wrong motifs are being assigned,

(16) Too many figures are far too small and could be double or trebled in size, especially in the supplement and EV where space is not an issue. As a general criticism, almost all of the genome browser views are at much too low a resolution. Gene regulation and active chromatin are best studies at the level of 100 kb or less.

(17) Although the authors have addressed some concerns, many of the problems highlighted in the first review remain. These have been addressed above so I will not address the rebuttal letter in detail.

Overall, the authors repeatedly refer to studies from the creators of ChromHMM to validate the use of ChromHMM, which is hardly an independent validation.

If other groups are already using better targeted approaches to define enhancers in CLL, and getting different answers, it does not justify published a less well-designed study that may contradict it.

REVIEWER #1 (comments during the referee cross-commenting process):

Enhancer definition and motif analysis

Rev #3 is correct that chromHMM should not be used to define (active) enhancers; in particular, not if the follow-up purpose is to define enriched motifs in these enhancers. ChromHMM tiles the whole genome into bins with a binary signal (presence/absence) of the chromatin mark. From the individual chromatin marks, a user defined number of compound "states" are assigned. Importantly, even the presence of a very low signal (e.g. low levels of H3K4me1, H3K27ac etc.) may cause that such tiles are defined as "enhancers", without any nucleosome depleted region. The authors can easily verify this by including the chromHMM segmentation in a genome browser. They will see that many enhancers with an active state overlap H3K27ac and ATAC-seq peaks, but that even a higher number of regions with a very low signal will share this state. In short, chromHMM might be useful to segment the genome into different states, but will introduce many false positives when used as a "enhancer detection tool". Moreover, even chromHMM segments that overlap the truly active enhancers are often large regions enriched in H3K4me1/H3K27ac from which the nucleosome depleted regions cannot be easily inferred. This is indeed critical for a reliable motif analysis.

I suggest to define enhancers by calling peaks (e.g. MACS2) on the ATAC-seq data and H3K27ac data (the latter possibly with the -broad setting). In the case where multiple replicates are available, it is recommended to use the IDR framework to make sure peaks are reproducibly found in two replicates. It is recommended to upload the tracks of the ATAC and H3K27ac peaks to a genome browser and verify that many ATAC peaks are fully embedded within a broader H3K27ac region. There will also be ATAC-seq peaks without overlapping H3K27ac, which often correspond to CTCF binding sites or TFs related to repressive marks like H3K27me3. The intersection (e.g. bedtools intersect) of the ATAC-seq and H3K27ac tracks is a good proxy for active promoters and enhancers.

Rather than inferring motifs in all the enhancers, the authors may want to look into differential enhancers between CLL and healthy cells. Such differential enhancers can be defined by significantly altered ATAC-seq levels AT the ATAC-seq peak and/or strongly altered levels of H3K27ac at the H3K27ac regions overlapping the ATAC-seq peaks. Altered enhancer activity is not always evident from an increased/decreased nucleosome depletion and so both methods are worthwhile to investigate. Homer can be used for motif enrichment and all enhancers defined in the above way should be used as a background set.

For the motif analysis, the authors may want to discard regions in close proximity to an annotated TSS, or treat these as active promoters. The ATAC-seq/H3K27ac intersecting regions will typically

be several hundreds of basepairs and can be used for motif analysis. Alternatively, the ATAC-seq summit +/- 150bp can be used (see comments rev #3). Binding sites for CTCF and EBV1 ChIP-seq should also be defined using peak calling, e.g. by MACS2 and not by clustering reads over the whole genome. Altered binding levels should be assessed with an appropriate tool like DESeq or Diffbind. A correct implementation of these basic analyses is critical and should take place before more sophisticated (network) models are pursued.

Normenclature

The nomenclature around Tcf3/Tcf4 can be confusing and can refer to the Tcf/Lef family (official symbols Tcf711 and Tcf712) or the bHLH motif family (E2a). When the authors assign genes/TFs to motifs, official gene symbols should be used. The consensus motifs or logos shown in the paper should be the same as the ones used for the motif analysis. Homer provides these motifs: <http://homer.ucsd.edu/homer/motif/HomerMotifDB/homerResults.html> or they can be reconstructed from the PWMs that were used. The authors should use the official gene symbol annotated with the gene. In the case that the motif is highly degenerate (e.g. AP-1 or ETS motifs) it is helpful to include the RNA-seq data and show the expression pattern for the highest expressed/most differential members of that motif cluster.

Superenhancers

Superenhancers have been defined by the group of Richard Young (Whyte et al. 2013) as H3K27ac regions containing multiple Med1 binding peaks. They are also called stitched enhancers, which is probably a better name, as it better describes their nature: it is a cluster of individual enhancers, not one enhancer with a particular "super" capability of driving gene expression. If there is anything special about them, it is their proximity to each other and to lineage specifying/cell identity genes. In defense of Richard Young's group they have never claimed that interstitial regions are also enhancers and provide the locations of the individually stitched enhancers in their paper. Thus, it is not surprising that interstitial regions have no enhancer activity. In the context of this paper, the individual "super enhancer constituents" should be contained in the set of all enhancers. It is only useful to define superenhancers in the CLL context when these are located near/driving the transcription of key CLL genes. Also here, a differential motif analysis (using the SE embedded ATAC-seq peaks) between CLL and healthy subjects would be of interest. Are these same motifs enriched as in the "normal" enhancers?

Response to reviewer comments for revised manuscript MSB-18-8339R

We appreciate the detailed comments of the reviewers and the editor to our revised study and are particularly grateful to reviewer #1 for the additional work he/she has put into the constructive cross-commenting of the various issues raised. In the General Comments 1-3 below we have summarized how we have addressed recurring issues. In addition, we have carefully considered, discussed and addressed all comments in the point-by-point response (reviewer comment highlighted in blue and renumbered). A number of additional analyses have been included as requested by the reviewers. For the enhancer annotation, different approaches exist as discussed below and we provide additional data and explanations for the workflow we have chosen. After having compared the different methods that have been suggested, we, as the authors of the paper, ask to be given some freedom to select what we consider the best suited method. We would like to emphasize that we make all data accessible so that other scientists can use them to conduct alternative analyses as they wish.

General comment 1, Enhancer annotations

In our opinion, the combination of ChromHMM states and ATAC signal provides additional information over an enhancer annotation based only on H3K27ac and ATAC-seq. It has been shown previously that H3K4me1 and H3K9ac are highly informative about the active enhancer state (Calo & Wysocka, 2013) and that active enhancer may also lack H3K27ac (Pradeepa et al, 2016). Furthermore, the addition of the H3K36me3 mark allows it to distinguish intragenic from intergenic enhancers. Accordingly, we consider it advantageous to annotate potential enhancers with ChromHMM from a combination of peaks called from H3K27ac, H3K4me1, H3K9ac and H3K36me3 data sets and then additionally consider the ATAC signal, bidirectional transcription or differential DNA methylation to annotate predicted enhancers rather than only use H3K27ac and ATAC measurements. Furthermore, there are certain regions we would wish to exclude based on the ChromHMM context (e.g. unannotated promoters or regions where H3K27ac signal occurs with repressive marks, both of which are identified in our analysis).

As proposed by the reviewers we have rephrased the text to clarify that the ChromHMM states alone do not represent an “enhancer definition”. To make this clear, we have also renamed the ChromHMM states in Fig. 1B as “Active 1” (state 10 with H3K4me3, predictive for TSS), “Active 2” (state 9 with H3K4me1, H3K27ac, predictive for active intergenic enhancers), “Active 3” (state 1 with H3K4me1, H3K27ac and H3K36me3, predictive for active intragenic enhancers), “Active 4”: (state 11 with H3K27ac, H3K9ac, predictive for an weakly active enhancer state that lacks H3K4me1) and “Poised” (state 8 with H3K4me1 but no H3K27ac, predictive of a poised intergenic enhancer state). These states were derived from ChromHMM by using binarized data obtained after peak calling of the 7 histone modifications as input. Thus, peak calling is implicitly included in our ChromHMM chromatin state annotation

Additionally, we would like to note that the presence of an ATAC signal is no gold standard for active enhancers in line with comment #9 of reviewer 1: (i) Low nucleosome occupancy (for example at poly(dA:dT) tracts or unusual DNA structures like R-loops) can lead to accessible chromatin regions that are independent of TF binding. (ii) Increased accessibility could also reflect the binding of a repressive complex. (iii) TF binding can also occur without a

concomitant gain of chromatin accessibility as inferred previously for binding of some TFs like Oct4 and Sox2 to sites with high nucleosome occupancy (Soufi et al, 2015; Teif et al, 2012). Furthermore, some promoters are active but show no ATAC signal although they must be bound by multiple general and specific TFs to be transcribed.

General comment 2. Enhancer annotations used in our study

We have carefully considered and discussed the reviewer comments as described in the current revision letter and in the previous one. We selected the following approach for annotation different set of enhancers in CLL and NBCs (NBCs):

- ChromHMM was used to annotate potential enhancer regions as states 1, 8, 9 and 11 (including putative poised or weakly active states) by using peak calling data of histone modifications as input. Different combinations of these states were used to derive different enhancer sets after intersection with the ATAC-seq signal (see below).
- The super-enhancers (SEs) were identified with the ROSE software from the Young lab (Loven et al, 2013; Whyte et al, 2013). It is the standard tool to call SEs and has also been used in a recent study of CLL SEs (Ott et al, 2018). Following the suggestions from both reviewers, we have refined the SE calling with the ROSE tool by using the H3K27ac read signal within the overlap region of ATAC and H3K27ac peaks.
- All promoter regions as defined by a region of ± 1 kb around the TSS according to the RefSeq database were removed from all our enhancer annotations.
- In order to identify TF motifs gained or lost in CLL within the ATAC peak regions, the differential ATAC signal was intersected with either all four states 1, 8, 9 and 11 (Fig. 5A,C) or only with the predicted active enhancer states 1 and 9, which carry a strong H3K27ac signal (Fig. S5 A) or with the predicted poised/weak enhancers from state 8 and 11 (Fig. S5 B).
- For the lists of active enhancers in CLL and in NBCs the ChromHMM states “Active 2” (state 9, H3K4me1, H3K27ac) and “Active 3” (state 1, H3K4me1, H3K27ac and H3K36me3) were intersected with regions of ± 1 kb around the center of ATAC-seq peak. This size was selected based on the average extension of key enhancer marks (H3K27ac, H3K4me1, p300, bidirectional transcription) (Chen et al, 2018). We do not consider it excessively long since the nucleosome-depleted regions can also comprise more than one nucleosome and definition of the histone modification state requires multiple nucleosomes flanking the nucleosome depleted regions. The resulting list of predicted active enhancers comprises 10145 loci in CLL and 7312 in NBCs of which 4771 are shared between the two groups (see Dataset EV09). These enhancer lists were used for further analysis and integration into the regulatory network described in Fig. 5 and 6 and the associated expanded view and supplementary figures.
- The predicted active intronic and intergenic enhancer ChromHMM states (1 and 9, see above) were intersected with sites of bidirectional transcription to yield a set referred to as “transcriptionally active enhancers”.

The part of the manuscript associated with Figs 4, EV4 5, S5 and EV5 has been rewritten to cover these points.

General comment 3, TF motif analysis

We consistently used the full HOMER transcription factor binding motifs. These motifs are experimentally determined and contain the weight of every nucleotide at each position in the motif sequence. Genome-wide binding sites were predicted based on the detection threshold as log odds ratio. We would like to emphasize that the TF motif analysis with Homer was *always conducted only within ATAC peak regions* of 327 bp median size throughout the manuscript (see point #10 below). This has been stated already (e.g., p. 10. “We identified TF binding motifs that were enriched at ATAC-seq peaks gained in CLL enhancers (Fig 5A)) but we have further clarified this issue throughout the text. The TF motif analysis for different enhancer annotations is now shown in Fig. 5A,C and Fig. S5). As proposed by reviewer 1, an additional TF motif analysis was conducted by intersecting the differential H3K27ac peaks (excluding promoters) with ATAC peaks either genome wide (Fig S5C) or for SEs called as described above (Fig S5D).

It is emphasized that we consistently used the full Homer motifs, also for the set of core TFs with gained/lost binding in CLL in Appendix Table S2. The simplified binding motifs listed previously in Table S2 were only intended as a reference to the complete HOMER motifs. We apologize for the confusion this has created. These sequences have now been replaced with full HOMER motifs retrieved from the differential ATAC-seq analysis. For the results of the TF motif analysis, we now consistently used the HOMER motif names while the HGNC nomenclature is applied when referring to specific TFs that recognize a given motif. As described in the text, in cases where multiple TFs would recognize a motif identified by Homer we have selected those TF(s) that showed a change in activity as computed by VIPER between CLL and NBCs that corresponded to the differential ATAC signal.

Editor comments

1. ChromHMM needs to be used with caution when defining enhancers, since strictly speaking it is a tool for detecting genomic states but cannot be used on its own to define enhancers. The reviewers provide constructive suggestions on how to use additional features to define enhancers.

See General Comment 1.

2. The motifs used for the motif analysis need to be better specified and shown in the manuscript. Moreover, the TF regulator analyses need to be performed with more stringent criteria and perhaps in more restricted genomic regions, to avoid false positives. Both reviewers provide constructive suggestions in this regard.

See General Comment 3

3. The super-enhancer calling needs to be refined.

See General Comment 2

4. Reviewer #1 also recommends performing further analyses to better support the proposed promoter-enhancer interactions. As the reviewer mentions, confirming the interactions by Hi-C or 4C would be rather challenging, but s/he proposes alternative options for analyses to address this point.

We have expanded this part as requested. When all interactions present in the 4D Genome data base (<https://4dgenome.research.chop.edu/Download.html>) for Homo sapiens were considered, 72 % (CLL) and 74 % (non-malignant) of the interactions derived from our scATAC-seq analysis were confirmed by the 4D Genome database. These numbers changed to 58 % (CLL) and 56 % (NBCs) when only 4D genome pairs for contacts within the 10%-90% distance range were considered, which corresponds to 7.23 kb to 429 kb. Most of the distances in the 4D Genome database are below 100 kb with the distribution shown below (response to point #21) and similar to that derived for the scATAC-seq data (Appendix Figure S6G). We conclude that our 100 kb cut-off is reasonable.

5. As a side note, we would recommend further emphasizing the resource value of the study (which was also explicitly acknowledged by reviewer #2 in the initial round of review) e.g. by adding a sentence in the abstract.

We have emphasized the resource value of our study in the abstract and the end of the introduction. It is noted that reviewer 2 also commented positively on this aspect of our work: "Together, this represents one of the most impressive and comprehensive epigenetic profiling efforts I have seen for any cancer type, and these data sets will certainly be an invaluable resource for the CLL field, and as a model for future investigations in other cancers. The authors are also to be applauded for their efforts in making not only raw and processed data, but also their custom analysis scripts available to the community." We believe this comment captures an important aspect of our work that should be considered in the context of the discussion of the "best" way to analyze the data. For example, it is straightforward to use the data we provide to annotate predicted enhancer loci with a different approach and then discuss the results in relation to our findings.

6. On a more editorial level, we would like to ask you to make sure that the Author Contributions are correct. Currently SR is listed as a contributor, it seems that this is a typo and it should be DR for Daniel Remondini?

Yes, this typo has been corrected.

Reviewer #1 (comments during the referee cross-commenting process)

7. *Enhancer definition and motif analysis.* Rev #3 is correct that chromHMM should not be used to define (active) enhancers; in particular, not if the follow-up purpose is to define enriched motifs in these enhancers. ChromHMM tiles the whole genome into bins with a binary signal (presence/absence) of the chromatin mark. From the individual chromatin marks, a user defined number of compound "states" are assigned. Importantly, even the presence of a very low signal (e.g. low levels of H3K4me1, H3K27ac etc.) may cause that such tiles are defined as "enhancers", without any nucleosome depleted region. The authors can easily verify this by including the chromHMM segmentation in a genome browser. They will see that many enhancers with an active state overlap H3K27ac and ATAC-seq peaks, but that even a higher number of regions with a very low signal will share this state. In short, chromHMM might be useful to segment the genome into different states, but will introduce many false positives when used as a "enhancer detection tool". Moreover, even chromHMM segments that overlap the truly active enhancers are often large regions enriched in H3K4me1/H3K27ac from which the nucleosome depleted regions cannot be easily inferred. This is indeed critical for a reliable motif analysis.

The rationale for our approach for the enhancer annotation is given in the General Comments 1 and 2. As described above we have clarified that ChromHMM states are not used as enhancer definition. Furthermore, all motif analysis is exclusively based on ATAC-seq peaks (General Comment 3). We would like to point out that peak calling also results in binarized data and that we have used the binarized data obtained after peak calling for the 7 histone modifications as input for our ChromHMM state annotation. Accordingly, only regions of chromatin with significant enrichment of histone modification signal are used for the segmentation. Thus, we disagree with the statement that our analysis would somehow artificially detect "a very low signal (e.g. low levels of H3K4me1, H3K27ac etc.)" as compared to using only peak called H3K27ac data (see also below, response to point #8). With respect to the statement "chromHMM segments that overlap the truly active enhancers are often large regions enriched in H3K4me1/H3K27ac from which the nucleosome depleted regions cannot be easily inferred" we would like to reply the following: Considering our General Comment 1, it is unclear to us how one would define "truly active enhancers" in the absence of data from additional experiments where these regions have been deleted or translocated and the effect on transcription was evaluated. The observation that H3K4me1/H3K27ac regions can cover 10 kb or more does not preclude that this feature is functionally relevant as discussed in point #49 of our previous revision letter. Whenever a motif analysis was conducted in the manuscript we used the ATAC-seq data/peaks, which in itself is comprised of small, defined regions, and used the ChromHMM states only to annotate the ATAC peak context. Thus, an extension of the predicted enhancer region beyond the ATAC peak size is not an issue for the TF motif analysis.

8. I suggest to define enhancers by calling peaks (e.g. MACS2) on the ATAC-seq data and H3K27ac data (the latter possibly with the -broad setting). In the case where multiple replicates are available, it is recommended to use the IDR framework to make sure peaks are reproducibly found in two replicates. It is recommended to upload the tracks of the ATAC and H3K27ac peaks to a genome browser and verify that many ATAC peaks are fully embedded within a broader H3K27ac region. There will also be ATAC-seq peaks without overlapping H3K27ac, which often correspond to CTCF binding sites or TFs related to repressive marks

like H3K27me3. The intersection (e.g. bedtools intersect) of the ATAC-seq and H3K27ac tracks is a good proxy for active promoters and enhancers.

We have conducted the analysis suggested by the reviewer. It yields a very similar set of enhancers that we have derived from the intersection of the predicted active ChromHMM enhancer states (state 1 and 9) and the ATAC-seq signal (see General Comment 2). An exemplary data set for one patient is given in Fig. R1.

Figure R1. Overlap of enhancers called by intersecting ChromHMM states 1 and 9 with ATAC peaks (grey) vs. H3K27ac peaks and ATAC in patient sample CLL1. Averaged over all patients 85 ± 8 % of the ChromHMM enhancers overlap with those from the H3K27ac peaks.

The large overlap of 85 ± 8 % (average and standard deviation for all patients of ChromHMM vs H3K27ac peaks) is to be expected because states 1 and 9 carry a strong H3K27ac signal that is obtained from the peak called H3K27ac ChIP-seq data used as input for ChromHMM. Furthermore, the total number of enhancers is somewhat lower in all samples for the ChromHMM based enhancer calling with values of 4977 ± 1863 vs 6160 ± 2080 (average and standard deviation for all patients) due to the additionally required presence of certain other histone modifications. Thus, the statement made in point #7 that “chromHMM ... will introduce many false positives when used as a “enhancer detection tool” as compared to H3K27ac peaks is not correct for our study. Considering these results, we do not think that the enhancer definition from intersection of ATAC and H3K27ac would be an improvement as also explained above in General Comment 1 and 2 and in response to point #7. The approach to only use H3K27ac is typically applied if information on other relevant histone modifications is lacking.

9. Rather than inferring motifs in all the enhancers, the authors may want to look into differential enhancers between CLL and healthy cells. Such differential enhancers can be defined by significantly altered ATAC-seq levels AT the ATAC-seq peak and/or strongly altered levels of H3K27ac at the H3K27ac regions overlapping the ATAC-seq peaks. Altered enhancer activity is not always evident from an increased/decreased nucleosome depletion and so both methods are worthwhile to investigate. Homer can be used for motif enrichment and all enhancers defined in the above way should be used as a background set.

All of our enhancer TF motif analysis is based on a differential analysis between CLL and NBCs using HOMER. We performed an analysis using differential ATAC-seq in combination with ChromHMM states (Fig. 5A, C, Appendix Fig. S5A, B), a differential H3K27ac signal at ATAC peaks (Appendix Fig. S5C) and also TF motifs at SE ATAC peaks (Appendix Fig. S5D).

We fully agree that altered enhancer activity is not always evident from an increased/decreased nucleosome depletion (see General Comment 2), which is accounted for by the analysis included in Fig. S5C. It is noted that there is a large overlap of the TF motifs retrieved with the different sets in Fig 5 and Appendix Fig S5.

10. For the motif analysis, the authors may want to discard regions in close proximity to an annotated TSS, or treat these as active promoters. The ATAC-seq/H3K27ac intersecting regions will typically be several hundreds of basepairs and can be used for motif analysis. Alternatively, the ATAC-seq summit +/- 150 bp can be used (see comments rev #3).

See General Comment 2 and 3. In all of our enhancer analysis we have removed regions ± 1 kb around the TSS given in the RefSeq database. The TF motif analysis was conducted only in the regions of ATAC-peaks, which were called with MACS2 and differential ATAC-seq signals were calculated with DiffBind as described in the Methods section. Our distribution of ATAC-seq peaks had a median size of 327 base pairs, which is very similar size to the 300 bp suggested by the reviewer and is shown below in Fig. R2. Since the nucleosome depleted region can also comprise multiple nucleosomes, e.g., due to the binding of multiple TFs, some more extended nucleosome depleted region can arise after peak calling. Accordingly, we prefer to use the actual experimentally determined size of a given peak region rather than a fixed value for all peaks.

Figure R2. Histogram of the ATAC consensus peak list obtained from peak calling of the ATAC-seq data with MACS2. The median peak size is 327 bp.

11. Binding sites for CTCF and EBV1 [typo, should be EBF1] ChIP-seq should also be defined using peak calling, e.g. by MACS2 and not by clustering reads over the whole genome. Altered binding levels should be assessed with an appropriate tool like DESeq or Diffbind. A correct implementation of these basic analyses is critical and should take place before more sophisticated (network) models are pursued.

There seems to be a misunderstanding. We have indeed used MACS2 to call peaks for both CTCF and EBF1 and applied DiffBind to the resulting consensus peak list for the identification of differential CTCF and EBF1 binding sites from the ChIP-seq data. This has been clarified in the text and in the method section.

12. *Nomenclature*. The nomenclature around Tcf3/Tcf4 can be confusing and can refer to the Tcf/Lef family (official symbols Tcf711 and Tcf712) or the bHLH motif family (E2a). When the authors assign genes/TFs to motifs, official gene symbols should be used. The consensus motifs or logos shown in the paper should be the same as the ones used for the motif analysis. Homer provides these motifs: <http://homer.ucsd.edu/homer/motif/HomerMotifDB/homerResults.html> or they can be reconstructed from the PWMs that were used. The authors should use the official gene symbol annotated with the gene. In the case that the motif is highly degenerate (e.g. AP-1 or ETS motifs) it is helpful to include the RNA-seq data and show the expression pattern for the highest expressed/most differential members of that motif cluster.

We agree that this issue was presented confusingly and have revised the manuscript accordingly as described in General Comment 3.

13. *Superenhancers*. Superenhancers have been defined by the group of Richard Young (Whyte et al. 2013) as H3K27ac regions containing multiple Med1 binding peaks. They are also called stitched enhancers, which is probably a better name, as it better describes their nature: it is a cluster of individual enhancers, not one enhancer with a particular "super" capability of driving gene expression. If there is anything special about them, it is their proximity to each other and to lineage specifying/cell identity genes. In defense of Richard Young's group they have never claimed that interstitial regions are also enhancers and provide the locations of the individually stitched enhancers in their paper. Thus, it is not surprising that interstitial regions have no enhancer activity. In the context of this paper, the individual "super enhancer constituents" should be contained in the set of all enhancers. It is only useful to define superenhancers in the CLL context when these are located near/driving the transcription of key CLL genes. Also here, a differential motif analysis (using the SE embedded ATAC-seq peaks) between CLL and healthy subjects would be of interest. Are these same motifs enriched as in the "normal" enhancers?

See General Comment 2. We agree with the reviewer that SEs are most interesting in light of their CLL context, i.e. when they are near key CLL genes and have already described examples from Fig. 4D, E in the main text on page 9. In addition, exemplary regions of CLL associated genes with putative SE activation for CREB3L2 (Fig. EV4D), FMOD (Fig. EV4E) are shown. With the refined SE calling (using ATAC peaks in combination with H3K27ac according to the reviewers' suggestion) the list of differential SEs now also includes gain of SE activity near ETV6 and CTLA4, and loss near CDKN1A. A TF motif analysis within ATAC peaks in SEs has been added to the manuscript (Fig. S5D). The motif calling in SEs shows that the top enriched motifs in lost SEs were NFkB-p65 and EBF1 (as seen in the normal enhancers). In the gained SEs we also find the E2A binding motif in agreement with the other enhancer annotations. Thus, a significant overlap of motifs enriched in SEs and in "normal enhancers" is observed. The text has been revised to account for these changes.

Reviewer #1

14. Main comments. The authors have substantially improved the organization of the manuscript and consecutive sections are now better connected. The integration of the scATAC-seq with the GRN is interesting, but the number of identified regulators is very high, suggesting that the integrated method introduces many false positives. The TP's in the ROC curve (shown in the rebuttal) are also based on motif analysis of the same data, rather than true known regulators from the literature, thereby providing little evidence for the predictive accuracy of the method.

We thank the reviewer for the positive feedback on our revised manuscript. The integrated network of regulators, target genes and enhancers ("CLL specific GREN", Dataset EV14) includes only factors that have a differential activity as computed with VIPER or, in case of the target genes, were differentially expressed according to the DESeq2 analysis (see also reply to point #17 on the number of identified regulators). Please note that, as described in Methods, our initial B cell gene regulatory network was computed from a completely independent gene expression data set of 264 publicly available samples including normal B cells, B cell lymphomas (Basso et al, 2010). The previously shown ROC curved compared our selection of TFs from the motif analysis at aberrant chromatin features with the TF ranking based solely on the differential activity in CLL computed with VIPER. Furthermore, the scATAC-seq data were only used to derive the promoter enhancer connectivity for the network from the correlation analysis with our "RWire" R script.

15. The authors made an effort to validate the interactions derived from co-accessibility in scATAC-seq with data from the 4D nucleome database. Unfortunately, no 4C-seq experiments (as suggested) were included for validation. Also, ChIP-seq experiments for CTCF and EBF1 were performed in CLL patient and healthy controls. Taken together, the additional experiments are sufficient to validate some of the results, but were not used for a genome-wide verification. Instead, ChIP-seq analysis is mainly limited to a few examples and the analysis of the newly derived data raises some questions and is not described in the methods section.

Both CTCF and EBF1 data have been used to validate several conclusions made in our initially submitted manuscript: (i) We predicted from publicly available CTCF data that CTCF is enriched at the PMD borders. This conclusion was now confirmed for our CLL samples by the CTCF ChIP-seq data as shown in the bottom panel of Figure 2D. (ii) The differential EBF1 and CTCF occupancy as inferred from ATAC-seq was validated by the EBF1 and CTCF ChIP-seq. For EBF1 this is shown in Figs. 6E, S8A, S8F. Called CTCF ChIP-seq peaks overlapped with 55% and 47% of both gained and lost CTCF binding sites derived from ATAC-seq, respectively. This is now stated in the text. (iii) The CTCF ChIP-seq data reveal clear differences between CLL cells and NBCs (Fig. EV5A). Using DiffBind to extract differentially occupied regions from our CTCF ChIP-seq data we found that CTCF binding is lost in CLL cells at 5964 sites and gained at 441 sites (Fig. EV5B), with 93 % of the lost sites overlapping with peaks from the ENCODE data set for the B-lymphocyte cell line GM12875 (GEO GSM749670). (iv) Following the reviewer's comments, we have now further validated that rewired enhancers are stably occupied by CTCF in both B-cells and malignant cells. Thus, according to our data, differential CTCF binding does not control enhancer-promoter rewiring but rather has functions in direct regulation of transcription as shown in Figure 5E. This

conclusion is in line with other reports showing that CTCF impacts transcriptional activity by stabilizing loops that insulate contact domains (Merkenschlager & Nora, 2016; Nora et al, 2017), while specific enhancer-promoter interactions might involve other factors like YY1 (Weintraub et al, 2017) (see also point #20 and #23). (v) We have included additional genes for the validation of prediction from the EBF1 dependent part of our network (Fig. 6D, E, Fig. S8E). We find that for 3 out of 4 enhancers of EBF1 target genes displayed a CLL specific loss of EBF1 binding as predicted from our network (Results last paragraph, p.13).

16. The manuscript will substantially benefit from a genome-wide comparison of some of the model/epigenetic data derived hypothesis with the 4D interactions/CTCF and EBF1 ChIP-seq. This could be addressed in a few figure panels. See specific comments below.

See comments to points #15.

17. page1) 1,378 differential regulators (TF's and chromatin modifiers) out of 2,804 were detected between CLL and healthy controls. This number is extremely high and likely includes many false positives. Can the authors compare these results with some of their collected data to support this claim? E.g. how many of these regulators are (highly) expressed or differentially expressed in/between CLL and healthy controls. How many of the TF motifs are enriched in the differential ATAC-seq peaks compared to all ATAC-seq peaks?

This issue has already been discussed in point #63 of our rebuttal letter accompanying the first revised version of our manuscript. For the construction of the B cell network we initially used a precompiled list of 5927 regulatory proteins (TFs, associated co-factors, signaling enzymes etc.) (Alvarez et al, 2016). From this list, 3862 proteins were present in the B cell network. Using our RNA-seq data and the VIPER software we computed activities for 2804 of them and found 1378 regulators differentially active with an adjusted p -value/FDR < 0.05 (corresponding to 140 regulators). At a 0.01 threshold this number would only be slightly reduced to 1239 regulators being differentially active. The relevant data are in the xls file "Dataset_EV13-B-cell_network" (and for differential gene expression as calculated by DESeq 2 the in Dataset_EV11-RNAseq-dif-gene-expr). Please note that the activity values are calculated with VIPER from the gene expression of multiple target genes in the B cell network, with an average and median 56 and 45, respectively, of targets per regulator. This makes the resulting activity values more robust than the differential expression signal of an individual gene and also captures activity changes that do not occur at the gene expression level of a given regulator but for example due to posttranslational modifications. The methodology and the statistical methods are described in our manuscript and the cited references. Finally, we would like to emphasize that our study yields a core set of TFs listed in Appendix Table S2 that have been derived from the ATAC based TF motif analysis, are associated with aberrant chromatin features in CLL and have the *additional* requirement that they display differential activity/gene expression.

18. Page 8) Superenhancers (SE) are called using the ROSE program by stitching neighboring TF binding sites that have overall enriched H3K27ac. This means that broad H3K27ac

domains need to have multiple TF binding sites in order to be called a "super-enhancer". In the original manuscript (Whyte et al 2013), the authors used Med1 and H3K27ac. Here, the authors could use the ATAC-seq peaks and H3K27ac. By calling every H3K27ac region larger than 10kb (which is not uncommon for some promoters) a SE, many false positives might be introduced. Furthermore, the authors may consider removing SEs that overlap a TSS, to make sure these are genuine enhancers. Lastly, are the 6.3% and 5.5% of the SE unique to the study (compared to what then?) or unique to the CLL/healthy control? The latter makes more sense, and it would be of interest to perform a differential motif analysis in the ATAC-seq peaks of the differential SE's, using all (union CLL and healthy) ATAC-seq peaks within a SE as control.

We followed the reviewer's recommendations and now included a TF motif analysis within ATAC-seq peaks within SEs as Appendix Fig. S5D.

19. Page 10) CTCF binding was lost at 5,964 sites and gained at 441 sites. Such a dramatic change in CTCF occupancy within the same cell type is a spectacular result, but may indicate a flaw in the analysis. Are these genuine CTCF binding sites (e.g. called with MACS2 and $p < 1e-8$) that have significant changes (e.g. Diffbind or DESeq2) in occupancy? How many CTCF binding sites were detected in total? Figure EV5B indicates sites with very low CTCF coverage, which are likely background. Please add a description of the analysis (similarly for EBF1) to the methods section.

We agree that it is an exciting finding to detect differential CTCF binding at about 6,000 loci. We stand by our results and are convinced that this is not the result of a technical flaw in our analysis. The CTCF binding sites were called with MACS2 and differential occupancy were calculated by DiffBind using an FDR threshold of 0.01 with standard parameters. A description of the workflow is now included in the method section. In total about 35,000 CTCF binding sites were detected for both CLL and NBCs. Each dot in Figure EV5B represents the average value of all samples with a normalized log2 value. Thus, only regions with an at least 10-fold higher coverage than the background were considered in the analysis. The majority of loci showed an enrichment between 16 and 128-fold. Furthermore, our data displayed an 85% overlap with peaks from the ENCODE data set for the B-lymphocyte cell line GM12875 (GEO GSM749670), which gives us confidence that we have identified genuine CTCF sites. Furthermore, 93% of the sites losing CTCF in CLL are also found in the GEO data set (now stated in the text, see also response to point #15). We would like to note that ~40,000 CTCF binding sites per cell type have been identified in previous studies. It has also been reported that ~20-50% of these sites show some cell-type-specific binding and are therefore be considered "variable" (see our previous work (Teif et al, 2014) and references therein). Thus, considering the large number of different chromatin features between CLL and NBCs detected in our study (nucleosome positioning and DNA methylation are particularly relevant here), a total of ~6,000 or 17% sites with differential binding is well within the number of previously reported variable CTCF binding sites.

20. Page 11) 90% of the enhancers had CTCF stably bound in both cell types. While CTCF certainly overlaps with promoters and enhancers, 90% is a lot. Are the authors sure that these are genuine CTCF binding sites (see comment above)?

The 90% number refers to the percentage of rewired enhancers that were constitutively bound by CTCF (see also point #15 and #23). These enhancers were in active chromatin regions (or in other words accessible) in both cell types, which would be in line with the role of CTCF mentioned in point #15.

21. Page 11) We found that 68% of these pairs were also listed as spatial contacts in the 4D nucleome database (Teng et al, 2015), suggesting that many promoter-enhancer pairs involve physical contacts. In total 3955 promoter-enhancer pairs were identified, with most promoters being connected to one enhancer at mean and median distances of 32 kb and 20 kb (CLL) and 23 kb and 10 kb (non-malignant B cells), respectively".

The average promoter-enhancer distances are very short and it will be hard to validate these with 4C-seq or Hi-C given the high background from proximity ligation at this distance from the promoter viewpoint. Therefore, it is surprising that 68% of these interactions could be validated. Furthermore, an upper limit of 100 kb might be too short. The authors should consider the significant interactions derived from a (B-cell specific) 4D nucleome data sets and set their lower- and upper bound distances to e.g. 10% and 90% of the significant interactions in the 4D nucleome data set. Finally, the authors should describe which datasets were taken from the 4D nucleome, whether these are B-cell specific (or closely related) and how the analysis and comparisons were performed (methods section)

We have expanded this part as requested. When all interactions present in the 4D Genome data base (<https://4dgenome.research.chop.edu/Download.html>) for Homo sapiens were considered, 72 % (CLL) and 74 % (non-malignant) of the interactions derived from our scATAC-seq analysis were retrieved in 4D Genome. These numbers changed to 58 % (CLL) and 56 % (NBCs) when only 4D genome pairs for contacts within the 10%-90% distance range were considered, which corresponds to 7.23 kb to 429 kb. These additional data are now mentioned in the legend to Fig. S6G. Most of the distances in the 4D Genome database are below 100 kb with the distribution shown and similar to that derived for the scATAC-seq data (Fig. R3). Thus, the 100 kb cut-off seems reasonable to us.

Figure R3. Histogram of the distance dependence of interactions in the 4D Genome database. It can be seen that the frequency sharply decays above 20 kb and that a 100 kb threshold captures the majority of the interactions.

23. The authors suggest that the correlation in scATAC-seq is predictive of (CTCF) looping, but provide only one example. Given that a relatively high number of differential CTCF sites were found, one would expect a coordinated down-regulation in the scATAC-seq signal of CTCF sites that interact and are downregulated in CLL patients relative to control. Can the authors e.g. show a scatter plot where the correlation in the ATAC-seq signal decrease of two interacting loci is associated with the loss of CTCF occupancy in these two loci, for every differential CTCF binding site overlapping the scATAC-seq signal? A genome browser view of the ATAC-seq and CTCF signal for an exemplary locus would be a nice illustration for the supplement to really reinforce this point.

Our conclusions are different and we have clarified them in the revised manuscript. We tested whether differential CTCF binding was linked to loss/gain of correlation in the scATAC-seq signal between enhancers and promoters. However, we found that for the 90% of these “rewired” enhancer-promoter pairs the CTCF binding site proximal to the enhancer shows stable binding of CTCF in both CLL and NBCs (Fig. EV5G, see also response to points #15 and #20). Thus, we conclude that changes in enhancer-promoter interactions are not dependent on variable CTCF binding.

24. Similarly, the EBV1 [typo, should be EBF1] ChIP-seq was not/barely used to validate some of the results. How many peaks were found in CLL and healthy controls? How many were differential? Were these associated with gain/loss in ATAC-seq and H3K27ac at promoters or enhancers? Were these enhancers connected to the target genes that were predicted from the model and to what extent?

Results from the EBF1 ChIP-seq analysis are presented in Fig. S8. The number of differential sites and overlap with the ATAC-seq analysis are described in Fig. S8A. About 2/3 of lost EBF1 sites in CLL at ChromHMM enhancer states with EBF1 motif also showed loss of the ATAC signal in CLL. The average number of EBF1 binding sites for CLL was 5358 and for non-malignant 6298, which is now stated in the legend to Fig. S8B. The number of differential sites is now stated in the main text (paragraph: “Enhancer-promoter network changes were identified from single cell analysis”) and in the legend to Fig. S8 (“Analysis of EBF1 binding by ChIP-seq”).

25. Fig 6G (referred to in the rebuttal) does not exist

We apologize for this omission. This figure panel is now included as Figure EV5F.

26. CTCF and EBV1 and 4D nucleome analysis are missing in the methods section.

The methods have been added to the analysis section of the manuscript on pages 17 and Appendix Table S6.

27. Discussion. "Our integrative analysis .. can explain more than 80% of the transactional variance of CLL cells". This statement indicates that e.g. a multivariate regression using all the epigenetic signals as predictors can explain the RNA-seq change between CLL and healthy subjects with an adjusted R^2 of 0.80. If this is the case, the authors should show that. However, such an accurate prediction of gene expression changes is typically not feasible,

even with the high-quality integrative dataset the authors collected. In that case, the statement should be adjusted.

This statement refers to results shown in Fig. 6A. As suggested by the reviewer we have rephrased the sentence in the discussion. It now reads "Our integrative analysis of a comprehensive set of readouts revealed altered chromatin features at promoter/enhancer elements for more than 80% of the differentially regulated genes in CLL cells" (discussion page 14).

28. Transcription factor motif analysis. "In addition, only TFs were included that showed a significant differential protein activity (or gene expression for network target genes) as computed from our B cell specific transcriptome network (Fig 1D)". How do the authors compute protein activity from RNA-seq data?

The protein activity referred to here were calculated with VIPER using the B cell network constructed with ARACNE using the gene expression data from RNA-seq. This is described in the context of Fig. 1E (Results, page 5 results) and in Methods in the section that is now entitled "Gene regulatory network construction and activity calculation" (page 23) to make clear that also the activity calculation is described in this part.

29. Text/typos

- while the loss at enhancers might indicate a loss OF enhancer activity (page 1)
- "PMD's were enriched .. conformation capture". Please edit the sentence (page 6).
- Infexion point → Inflection point (legend Fig. EV1)

Thank you very much for picking this up. These errors have been corrected.

Reviewer #3

30. The revised manuscript is improved, but not yet to the point of being acceptable. Some of the deficiencies relate to terminology and can be corrected by reclassifying chromatin features under a different name. However, there are many substantial fundamental defects in both the approach and the analysis in this manuscript which relate primarily to:

We have changed the terminology as described in General Comment 1 and addressed the criticism to our analysis as described in the response to the specific points below.

31. Using an inadequate method to define enhancers: The ChromHMM pipeline is used extensively by the Roadmap Epigenetics Consortium, the authors of ChromHMM, but in my view this is a highly unreliable method to use to predict enhancers as it is not centred on any analysis of open chromatin regions. Based on the results of this manuscript it is highly likely that a large fraction of the predicted enhancers or changes in enhancer state relate more to changes in the activities of transcription units than enhancer activity. It may be relevant that ChromHMM was a default track on the hg18 UCSC genome browser, but not on the more recent hg19 and hg38 browsers, maybe because it has gone out of favour as a useful tool.

These statements are not correct. For example, the ENCODE ChromHMM data sets for the UCSC genome browser hg19 build are listed under the "Regulation – ENCODE Histone Modification" tracks as "Broad ChromHMM" (https://genome.ucsc.edu/cgi-bin/hgTrackUi?hgsid=716060827_FwR3ZiQP0yusQe8kGQ7OS4ConR8z&c=chr21&g=wgEncodeHistoneSuper). The hg38 build of the UCSC genome browser currently has limited track availability and excludes, e.g., Vista Enhancers, ENCODE DNA methylation, ENCODE transcription factor binding ChIP-seq. However, this is more likely due a slow migration of data rather than these tracks having fallen out of favor. Furthermore, the current Ensembl genome browser for hg38 (release 95 from January 2019) includes "regulatory segmentation" that makes use of ChromHMM segmentation from other projects (https://www.ensembl.org/info/genome/funcgen/regulatory_segmentation.html). Finally, the loci in our lists of active enhancers in CLL and NBCs used in our analysis (General Comment 2) are centered around open chromatin regions detected by ATAC-seq (see also response to point #32).

32. The main flaw in this approach is that active genes are essentially covered in varying degrees of H3K4 me1, me2 and me3 and H3 K27 ac, as follows: H3 K4me3 concentrates +/- ~1 kb, K4me2 extends a few more kb and K4me1 extends over most of the gene. While H4 K27ac is highest on either side of the promoter, it extends over the whole gene. H3K36me3 is concentrated on the 3' half of genes. This means that the three different ChromHMM enhancer states, especially genic enhancer and active enhancer 2, will often represent transcribed chromatin and not enhancers. Without alignment with well curated ATAC data, which excludes minor ATAC peaks associated with active nucleosomes, the ChromHMM data has no predictive value. This is abundantly clear in fig 1D, which does at least filter for overlap with ATAC data, but shows that the modifications predicted for active enhancers (enhancer 1 + H3K27ac) are in the minority. The biggest enhancer groups are defined on the basis of either "active enhancer 2" which is just H3 K4me1, found at the 5' half of genes, or "genic enhancer" which is H3 K36 me3, K4me1 and K27ac, found at the 3' half of genes. My original perusal of the table of 238,000 enhancers suggested that these often comprised whole coding regions. The enhancer state 1, marked by K4 me1 and K27ac, which should be found flanking enhancers but not at the enhancer itself, is the most useful group included. However, this is a

tiny fraction of the population of peaks in fig 1D. For confirmation of these views, I suggest that the editor and authors to look at Fig S4 of Wang et al Nature Genetics 40, 897 (2008) and Fig 4 of Barski et al Cell 129, 823, (2007) which were global studies of many histone modifications. The average gene profiles show K27ac, K4me1 and K36me36 on typical transcribed regions. This cannot be used to define enhancers. This is also why many "super-enhancers are defined on regions that include transcribed genes.

See General Comment 1 and 2 and response to reviewer 1, points #7 and #8. We would like to emphasize that we provide different sets of enhancers including lists of active enhancer loci in CLL and NBCs (ATAC-seq signal overlapping with or flanked by ChromHMM states 1 and 9). The later comprises 10145 loci in CLL and 7312 in NBC of which 4771 are shared between the two groups (Dataset EV09). With respect to the distribution of histone modifications at active genes we disagree with the conclusions made by the reviewer. Wang et al show in Fig. S4 that there is some H3K27ac in the gene body but also that H3K27ac levels are about 10x higher at the promoter. Barski et al show that the H3K4me1 signal continuously decays towards the end of the gene. Furthermore, it is noted that the levels of H3K27ac and H3K4me1 are reduced at exons. These features allow it to separate ChromHMM state 1 (predictive of intronic enhancers) from regions of low H3K4me1 and H3K27ac levels, since it is based on the cooccurrence of peaks called for H3K4me1, H3K27ac and H3K36me3. These points are apparent when comparing the end of the TCF4 gene containing a putative intronic enhancer with the transcription start site and exons in Fig. 1A.

33. Including ATAC data in only some of the analyses performed: Most studies should start by mapping open chromatin, looking at adjacent chromatin marks, and identifying motifs centred on open chromatin. The only time such an analysis is presented is in fig 6E, the final figure in the paper, and that shows one motif linked to one ATAC peak. For most of the paper, the criteria described on p22 for inclusion as an enhancer are just 2 of 6 patterns that do not have to include open chromatin, which should be obligatory. This leads to swamping of the data pool with active chromatin marks commonly found on active genes. Worse than that, most of the 238,000 active chromatin regions do not appear to contain ATAC peaks, meaning that any motif analysis is being performed on nucleosomes, not enhancers, and is meaningless. Furthermore, if 2/6 chromatin modifications are needed to define enhancers, this excludes enhancers at open chromatin regions where TFs are bound where there are no histones. This means that true enhancers and enhancer state 1 are mutually exclusive. This is also evident in the table of 238,000 enhancers, where some actual potential regulatory elements are excluded.

This description of our analysis is not correct. As stated in General Comment 3 the TF motif analysis was conducted *only* within ATAC peaks within regions that displayed a differential chromatin feature between CLL and NBCs. We clarified this in the methods section "Transcription factor motif analysis" (p. 23). Our enhancer definition and the rational for using it is given in the General Comments 1 and 2. It is indeed correct that our annotation of predicted enhancers starts from chromatin states 1, 8, 9 and 11 and is histone modification based. Regions lacking any histone modifications studied here would be annotated as state 12 (quiescent). These regions were not included in our enhancer analysis. A histone modification independent enhancer annotation would be via p300 ChIP-seq and bidirectional transcription as used by the FANTOM consortium (Andersson et al, 2014), for which we lack the p300 ChIP-seq. However, it is noted that the FANTOM enhancer annotation is implicitly linked to histone

marks associated with active transcription and histone acetylation. Accordingly, the FANTOM enhancers overlap with H3K27ac and H3K4me1 modifications and are *not* found in regions that are void of histone modifications (Andersson et al, 2014). We also disagree with the statement that open chromatin regions alone would represent “true enhancers” where “TFs are bound”. They might as well simply be biologically inactive regions with low nucleosome occupancy that have no TFs bound (see General Comment 1, last paragraph).

34. It is significant that a recent study published by Ott et al on the same subject this month in Nature Genetics used just ATAC and H3 K27ac, and came to some different conclusions. It is at least apparent that the target genes identified by Ott et al seem to be different to those identified here.

We assume that the reviewer refers to a recent study in Cancer Cell and not in Nature Genetics (Ott et al, 2018). Their TF analysis is very different from the one in our study. We conduct a differential TF motif analysis of CD19+ B cells in CLL vs. NBCs within ATAC peaks for different enhancer annotations (including SEs and yielding a large overlap) and identify TF motifs enriched or depleted at enhancers (Fig 5A, C, Fig S5). The set of 15 central TF motifs in CLL (including promoters) identified in our study is linked to different aberrant chromatin features and is independent of calling SEs (Appendix Table S2). In contrast, Ott et al only analyze SEs. While their analysis includes also a differential SE calling between CLL and NPCs based solely on H3K27ac, they derive master TFs from an “enhancer-based modeling of regulatory circuits and assessments of transcription factor dependencies” in conjunction with cell line survival studies. Based on this work they conclude “that the essential super enhancer factor PAX5 dominates CLL regulatory nodes and is essential for CLL cell survival.” We do not find PAX5 as a top (super-)enhancer hit in our TF motif analysis but other TFs ranked high in the Ott et al study namely IRFs, FOXP1 and RUNX2/3. Since two very different approaches are used this is not surprising. It is thus unclear to us why the results of Ott et al should be a cause of concern with respect to the quality of our work. Rather, we note that our findings in terms NFAT, TCF4 and LEF1 becoming activated in CLL factors like EBF1 and AP-1 getting silenced are in full agreement with two other studies (Beekman et al, 2018; Oakes et al, 2016).

35. Flawed motif analysis: From the outset, the construction of a gene regulation network relies on motifs, but the motif table in table S2 is full of errors. If these motifs were used in the motif analyses, then all these analysed are fatally flawed, as follows: (i) The NFAT motif listed is AAGAAGAAA. This will never bind NFAT which binds (A/T)GGAAA, (ii) the authors are confused by the TCF nomenclature whereby for example human TCF3 and TCF4 belong to the HLH E2A CACCTG E-Box class whereas mouse Tcf3 and TCF1/LEF1 are HMG proteins that bind CAAAG. The authors imply that they are including both motifs in their E2A searches, (iii) the wrong RUNX motif is used. The right motif for RUNX proteins is TGTGGT(T/C)(T/A)without upstream CCCC which will greatly skew the analyses, possible making it 200x less likely to hit the basic motif, (iv) The FOX motif is not right, and at the least should start with G/A and not GA, and (v) the NF-kB motif is better represented by The NF-kB motif is best represented by GGGGAAATCCCC (which has an extra A).

See General Comment 3. We have always used the full Homer motifs retrieved from the analysis and apologize for the misleading sequence representation in Appendix Table S2. The table has been revised to address the issues raised by the reviewer.

36. Another fatal flaw, explained above, is that most motifs defined in this study are defined based on groups of modified nucleosomes, where there are no TFs, and the analyses may actively exclude enhancers bound by TFs where there are no modified histones.

See General Comment 1 and 2 and response to point #32 and #33.

Specific comments:

37. Regarding terminology, many objections are easily resolved by replacing every single reference to "enhancer regions" on every single occasion, without exception, with a reference to "active chromatin Regions", and also in e.g. Fig 1B, 1D etc etc.

This is a good suggestion and we have changed the ChromHMM state names as described in the General Comments 1 to make clear that ChromHMM does not provide an enhancer definition but an initial step to annotate enhancer candidate regions.

38. The main persistent problem that remains is the underlying assumption throughout the manuscript that enhancers can be defined as broad regions of modified chromatin. They must by necessity be defined as small regions of 200-500 bp of open chromatin bound by TFs. This authors methodology sidesteps the most important fundamental concept of gene regulation which is that the genome is reprogrammed by TFs which establish open chromatin regions. The modifications came later. In the case of changes in DNA methylation, sometimes much later.

See General Comment 1 and 2 and response to point #32 and #33. We disagree with the statement that "TFs ... establish open chromatin regions. The modifications came later." Given the much higher stability of a nucleosome (complex lifetimes in the range of hours to days) as compared to TFs (chromatin bound state typically on the minute time scale and below), most TFs are unable to simply displace a nucleosome by competitive binding. Rather, several lines of evidence support the view that histone modifications (or other signals) guide chromatin remodeling to subsequently open up chromatin and to allow TF binding (Erdel et al, 2011). This mechanism might be particularly relevant at enhancers with H3K4me1 targeting the activity of the BAF chromatin remodeling complexes (Local et al, 2018) to create an open chromatin region. Active chromatin remodeling might also contribute to differential CTCF binding (Teif et al, 2014) at the large fraction of sites that show cell type specific CTCF binding (see point #19).

39. Following on from above, any analyses of motifs should focus on +/- 200 bp at most of an ATAC peak. In their response to this suggestion in the original review, the authors question where my previous definition of enhancers as ~300 bp regions comes from (point 58). The answer is very simple and based on our fundamental understanding of mechanisms of gene regulation. For essentially every enhancer defined in the genome, enhancers are occupied by nucleosomes in the inactive or poised state, and are nucleosome-free once TFs have evicted 1 or 2 nucleosomes. 1 nucleosome + 2 linkers occupies $40 + 160 + 40 = 240$ bp. This is why DHSs are typically 250- 300 bp of open chromatin bound by TFs. The TFs do not as a whole bind outside this zone where chromatin is inaccessible. This is why motif analyses must be of ATAC peaks and not H3 peaks.

In their rebuttal, the authors say they look at motifs +/- 1 kb from an ATAC peak (point 58). This will create a massive background as it adds 1700 bp of nucleosomal DNA to each 300 bp enhancer regions, where most motifs are +/- 100 bp, the window the authors use in Fig 5D. Why not use the same window to define motifs and thereby get better enrichments.

See General Comment 2 and 3 and response to point #10. All of our TF motif analysis was conducted only within the ATAC-seq peak regions (median 327 bp). The extension to ± 1 kb from the ATAC peak center was used to define active enhancer loci according to the extension of their characteristic marks (Chen et al, 2018) but not for the TF motif analysis.

40. All analyses included in this study should be restricted to those samples that included ATAC, which is most, and always use ATAC as one of the two criteria used to define regulatory elements. This still allows for 3 normal samples where both ATAC and ChromHMM are available, which should be sufficient given the differences between these and the 18 CLLs that can be included.

See General Comment 3. For the differential analysis of a given readout we have used the full number of samples available that met the quality criteria. From this sample set a consensus feature list was constructed, which was then used with DiffBind to compare CLL and non-malignant samples. In case of ATAC-seq we have acquired data for 7 NBC pools (4 in replicates) and 19 CLL B cell patient samples (all in replicates).

41. ATAC should be at the top of the flow diagram in fig S1B, and not on the fourth tier where it sits now as an afterthought.

As explained in the General Comments 1 and 2 we used a workflow that starts with the ChromHMM annotation. We then intersected these regions with the ATAC peaks and conducted the TF motif analysis or prediction of active enhancer loci.

42. A major technical deficiency in this manuscript is the inadequate analysis of the TF motifs underlying the chromatin patterns which are presented in great detail. Analyses of motif occurrence is restricted to meta data showing probabilities etc. but not the primary data. To have confidence in the enhancer definitions it is necessary to plot motif positions on the plots of ATAC peaks in the style of the analysis shown in fig 5B. To verify enhancers adequately, the same figure should include the profile of ATAC peaks centred on a window +/- 1 kb, ranked by ATAC enrichment, with flanking histone modifications, and confirming the presence of motifs at the centre of the peaks more enriched in the enriched ATAC peaks. None of this has been done. Fig 5B is aligned on motifs, not ATAC peaks, and even then uses a flawed NFAT motif.

As described in General Comment 3, the motif representation is now consistently giving the Homer motifs identified in the analysis. It would be interesting to also further analyze the motif position within the ATAC peaks, e.g. by plotting the motif location with respect to the peak center for all TFs of interest. However, given that we have already expanded the TF motif analysis to address other reviewer requests in the text and added an additional figure (Appendix Fig. S5) we refrain from adding further figures. Our manuscript is already at the size limit. We would like to note, however, that we provide all data to conduct such and other analyses.

43. The authors should show the matrix logos and provide the PWMs used in this study. If these differ to the same motifs annotated on HOMER, these need to be shown alongside.

See General Comment 3. The Homer motif logos and exact Homer motif names are now provided in Appendix Table S2. From these the PWM can be retrieved.

44. It would be preferable to simplify the number of motifs depicted in figures such as 3E by restricting the motifs depicted to a single representative example in cases such as ETV/ELK/ETS and KLF where they all bind the same motifs. This figure probably needs ETS, KLF, E-box, SP1, NFY, NRF and no more. Fig 5C only needs one AP-1 and one RUNX motif etc.

See General Comment 3. The motifs names used in the TF motif enrichment plots are those used by HOMER for a consistent assignment to address the criticism of the reviewer above. The reduction to selected TFs based on motif recognition and differential activity is done in the revised Appendix Table S2.

45. As stated above, there is an over reliance on the ChromHMM gene prediction tool, released in 2010, which has not really gained wide acceptance in the gene regulation field as a front line approach for identifying enhancers. Published evidence from groups such as the Hardison and Cohen group suggests that enhancers should be defined on the basis of transcription factors or on function. Most studies measuring actual enhancer activity find ~10% of active chromatin regions have enhancer activity. While it is helpful to know about chromatin states in the context of gene regulation, over-use of this tool is likely to lead to misleading results. ChromHMM seems to tell us more about whether a region is transcribed or not.

See General Comment 1 and 2 and response to points #32 and #33. As stated above in response to point #37, we have changed the terminology to make clear that we do not consider ChromHMM an enhancer identification tool and also have intersected the ChromHMM states with ATAC peaks in our further analysis. The reviewer states that “Published evidence from groups such as the Hardison and Cohen group suggests that enhancers should be defined on the basis of transcription factors or on function.” It is, however, unclear to us how their work would be related to better approaches for the annotation of enhancers in our study. Zhang and Hardison have published a chromatin annotation approach termed IDEAS and compared it to ChromHMM (Zhang & Hardison, 2017). The software appears to perform somewhat better in terms of predicting “validated” enhancers from the FANTOM set. However, on an overall scale many chromatin states including “enhancers” and “bivalent enhancers” were “commonly identified with similar proportions in the genome.” The Cohen group has conducted work on the effect of the location of enhancer sequences in the genome. They conclude that the regional chromatin context strongly affects the activity of *cis*-regulatory sequences (Maricque et al, 2019), which is the reason why we use the ChromHMM annotation.

46. The ChromHMM browser tracks often show enhancers and promoters as 10 kb tracts of chromatin spanning open chromatin regions. As an example, ENCODE shows the first 50 kb of the B cell gene PAX5 at hg18 chr9:36,977,389-37,029,852 is covered by either H3 K4 me3 or me1 in lymphoblastoid cells and is shown as either enhancer or promoter region on the Broad Institute ChroHMM track in the same cells. This is highly misleading. Just because the current authors get the same patterns as the ChromHMM tracks loaded on the UCSC genome browser

does not mean that this tool is suitable to use as the front line approach to defining enhancers. It would in fact be a disservice to the gene regulation field to give de facto acceptance of this methodology by prominently presenting it as a state of the art approach to be used to predict enhancers. It is not, it is an adjunct to define the neighbourhood of enhancers.

As stated in the General Comment 1 and in response to point #37 we have renamed the ChromHMM states and rephrased the text to clarify that the ChromHMM states are no “enhancer definition”. Nevertheless, we consider them useful to predict different sets of potential enhancer loci when combined with ATAC peaks as discussed above.

47. To illustrate another feature of histone profiles, some ChromHMM data nicely show a gap between the K4me1 and me3 marks, just where K4me2 would be predicted. This is what is expected in a transcription unit, but ChromHMM would show it as a gap between chains of enhancers.

For our enhancer analysis we have always removed a region of ± 1 kb around the TSS (s. General Comment 2). Thus, any dependence of the ChromHMM segmentation on the H3K4me1, H3K4me2 and H3K4me3 distribution in this region is irrelevant for the enhancer assignment in our analysis.

48. Page 8 line 15 refers to transitions from the quiescent state to the K4 me1 E2 state, and from the transcribed K36me3 state to the K36me3/K4me1 genic enhancer state. Both represent gain of K4me1, which may just reflect differences in sensitivity of K4me1 detection, and in the second case depends on an ill-defined boundary between states seen at the 5' and 3' halves of genes. There is plenty of room for over-interpretation of the data here.

We have rephrased this section of the text with the revised ChromHMM terminology to make the distinction between ChromHMM states and enhancers. With respect to the genic enhancer state, which has been now renamed to “Active 3” (state 1), we would like to point out that it contains H3K4me1, H3K36me3 and H3K27ac. Thus, for a transition from only H3K4me1 (state 8) or H3K36me3 (state 2), it is additionally required to gain H3K27ac (see also response to point #8).

49. The opening figure, Fig 1A, is poorly described and has inconsistencies. The sentence “the histone modifications identified a downstream enhancer, which was activated in CLL cells as judged from the enrichment of H3K4me1 and H3K27ac (Fig 1A)” should be turned the other way around to say that “a potential enhancer region was identified far downstream of the TCF4 gene as a cluster of ATAC peaks embedded within an ~ 60 kb region of chromatin enriched in H3K4me1 and H3K27ac in CLL cells”. This approach needs to be taken throughout the manuscript whereby any description of a potential regulatory region must begin with a description of the ATAC data.

As stated in response to point #41 we disagree that we “must begin with a description of the ATAC data”.

50. The 2 RNA-Seq scales in Fig 1A should have some degree of equalisation across the genome, as it is not acceptable to show normal on a scale of 100 and upregulated on a scale of 8000, as it suggests the opposite of what it is. For example, a uniformly active region of the genome could be used to either normalise the FPKM data. Alternatively the authors could

adjust the scales to make them equivalent, and then maintain the same ratio in the scales. Furthermore, (i) some of the ATAC peaks at TCF4 look bigger in normal cells, and (ii) the whole TCF4 locus is plastered with H3K4me1 and H3K27Ac in CLL. Does that mean the whole TCF4 locus is one big super-enhancer? Because the whole gene is involved, this data could be actually be used to undermine the very concept of the "super-enhancer", a process that is already happening in other studies that dispute the relevance of this definition (eg at the alpha globin locus, and in Barakat et al recently in Cell Stem Cell).

RNA-seq data were normalized to RPKM values. In response to the previous reviewer request (see previous revision letter, response to point #6) we have introduced different scales for CLL vs NBCs in Fig. 1A as stated in the figure legend. In this manner it can be seen that TCF4 is already lowly expressed in NBCs as also evident from the presence of the H3K36me3 mark but that the expression level in CLL is considerably higher. This is stated in the figure legend. The large enhancer region linked to TCF4 is indeed also called as a differential SE in the analysis (Fig. 4D). This is now stated in the text.

51. The significance of the PMDs is unclear. They were defined at 10 kb resolution which is too high to be meaningful. Either 500 or 1000 bp by would be better, as a 10 kb region can contain one 10% and one 90% methylated region. Fig 2A shows a highly variegated pattern of methylation, so maybe this is indeed the case. It would be better to show verification with single clones from bisulphite sequencing of a few hundred bp where meCG is traditionally shown as black dots in a series of clones to reveal the trend. The 50% could also be a population average. Basic mechanisms of activation and repression tend to drive the unstable 50% meCG state one way or the other, so 50% is unexpected at one discrete region. On p8 line 11, when the authors say PMD loss and meCG loss correlate with loss of enhancers, I suggest these changes simply correlate with gain in transcription.

PMDs are defined in the literature as large regions (>100kb) containing highly-variable, disordered DNA methylation patterns. To identify PMDs in CLL, we used the approach described by Berman *et al.* (Berman et al, 2011), which is detailed in the methods section. PMDs were defined at single CpG resolution by computing the average methylation value at each CpG within a sliding window of 10 kb size. Thus, our resolution is better than 10 kb. Due to the large region size of >1 Mb needed to display PMDs as shown in Fig 2A, it is impossible to indicate individual CpG methylation sites. The impact of PMDs on gene expression is shown in Figure 2C. We also report separately on meC loss at individual differentially methylated regions (DMRs) in Figure EV2E, F and Appendix Figure S2G. These regions are provided in Dataset EV3. The meC loss at DMRs was correlated with enhancer activation as shown for NFAT binding sites in Fig. 5B.

52. the data on nucleosome changes at promoters and could be left out. This section does not provide any clear insights. The data is not convincing.

We believe that the nucleosome analysis provides an additional layer of information and points to B cell receptor signaling as a central player in the pathomechanism of CLL. It is unclear to us what the reviewer means with the generalizing comments "does not provide any clear insights" and "The data is not convincing" without any specific suggestions for improvement. Rather, based on the experience from our previous work on nucleosome positioning (Teif et al, 2014; Teif et al, 2012), we would argue that the nucleosome positioning data provided in

our current study are of exceptionally high quality and unique as they are obtained with primary tumor samples. They allow it to detect the gain or loss of individual nucleosomes as illustrated in Fig. EV3C. Thus, we are confident that this part of our work will be appreciated by other scientists and will prove to be a valuable resource.

53. Some major conclusions in this study are based on the flawed motifs TCF4 (at the centre of the network) and NFAT (e.g. 1E, 5A, EV2F).

See General Comment 3 and response to point #35. All motifs used in our analysis were taken from HOMER and are included now in Appendix Table S2. We apologize for the confusion.

54. The concept of "superenhancers" remains controversial and has not been widely accepted in the basic gene regulation community, even though many groups are now trying to define these regions. Most problematic with this concept is that many of the superenhancers defined include whole transcription units and promoters flanked by enhancers and other regulatory elements. This configuration was originally defined as the "active chromatin domain", and the clusters of enhancers were defined as LCRs. In my view the redefinition of these as superenhancers is a retrograde step. In the original paper, Whyte et al defined 231 superenhancers as a small group of highly specialised LCRs. Even their prime examples depicted in the figures include whole transcription units. This is also true of two of the three prime examples shown in the present study in fig 4I where the CDKN1a and KLF13 genes make up a big part of the superenhancers. The current authors may also be a bit over-zealous as they defined 829 superenhancers, 4x the expected number.

In their rebuttal (point 49 and 68) the authors use recent descriptions of "liquid phase compartments" from the Sharp lab as evidence that superenhancers are meaningful. I would suggest that this new definition is another way of restating the fact that active genes reside within active chromatin domains, which are more accessible, dynamic and interactive. This concept has been around for decades, and like LCRs and superenhancers, has been reinvented. The study Barakat et al also show most regions within a "superenhancers" do not have enhancer activity. The authors can claim that active chromatin domains enhance TF recruitment at localised enhancers, but not that TFs bind cooperatively all across domains > 10 kb. This is a fallacy.

In contrast to the Ott et al study mentioned above by the reviewer, our work has neither a focus on SEs nor do we claim that SEs are particularly important for deregulated epigenetic signaling in CLL. We simply consider them as another subset of putative enhancers that are assigned according to a specific genome annotation used in the literature (General Comment 2). As pointed out by the reviewer the concept is controversially discussed. Thus, we consider it informative to provide also a SE analysis in our study as we expect it to be of interest to the readers of our study. Finally, as discussed in our previous rebuttal letter, it seems premature to conclude that the "functional" part of enhancers must be restricted to the nucleosome depleted region and that large extensions of chromatin features like H3K27ac are irrelevant for enhancer function.

55. Fig 6A confirms that most upregulated "enhancers" are marked by H3 K4me1 and not K27ac. This should be seen as a warning sign that these are poorly defined.

The figure shows differential chromatin signals linked to gene expression rather than the appearance of the signals per se. Thus, it cannot be concluded from the figure that the up-regulated H3K4me1 sites are enhancers not marked by H3K27ac.

56. In fig EV4a the authors need to refer to Mb of active chromatin, not enhancers.

We agree with the reviewer and have revised the figure accordingly.

57. As mentioned in the original review, the patterns for TCF4 and EBF1 in Fig 5D do not resemble footprints but reflect sequence specificity within a ATAC peak centred on a sequence. Only CTCF shows a true footprint. This figure also highlights the error in the TCF4 motif. This should be the TCF3/TCF4 motif CACCTG, but the authors show the motif for TCF1/LEF1 which is CAAAG. This confirms that the wrong motifs are being assigned.

While the footprints indeed include a significant contribution from the Tn5 sequence preference this is independent of the strong gain/loss of protection towards Tn5 integration in CLL versus NBCs. The error with the TCF4/LEF1 labeling has been corrected (see General Comment 3 and response to point #35).

58. Too many figures are far too small and could be double or trebled in size, especially in the supplement and EV where space is not an issue. As a general criticism, almost all of the genome browser views are at much too low a resolution. Gene regulation and active chromatin are best studied at the level of 100 kb or less.

The readability of figures was improved wherever possible. In order to increase size of the figure labels we have introduced the abbreviation for NBC for “non-malignant B cell”).

59. Although the authors have addressed some concerns, many of the problems highlighted in the first review remain. These have been addressed above so I will not address the rebuttal letter in detail. Overall, the authors repeatedly refer to studies from the creators of ChromHMM to validate the use of ChromHMM, which is hardly an independent validation. If other groups are already using better targeted approaches to define enhancers in CLL, and getting different answers, it does not justify published a less well-designed study that may contradict it.

The statement that “other groups are already using better targeted approaches to define enhancers in CLL” appears to be without merit (see response to point #34 and #45). Furthermore, our findings that NFAT, TCF4 and LEF1 were activated in CLL while other factors like EBF1 and AP-1 were silenced is in full agreement with two other studies (Beekman et al, 2018; Oakes et al, 2016). Based on our comprehensive analysis of aberrant chromatin features in CLL and their assignment to 15 central TF motifs (Fig 6B, Appendix Table S2) we have developed an integrated gene regulatory enhancer containing network, for which we have validated predictions for EBF1. Our network reveals a number of new links to the CLL pathophenotype to be tested in further studies. In addition, we provide a rich multi-readout resource of CLL chromatin features which, we are sure, will help to understand the molecular pathology of CLL with integrative approaches.

References

- Alvarez MJ, Shen Y, Giorgi FM, Lachmann A, Ding BB, Ye BH, Califano A (2016) Functional characterization of somatic mutations in cancer using network-based inference of protein activity. *Nat Genet* **48**: 838-847
- Andersson R, Gebhard C, Miguel-Escalada I, Hoof I, Bornholdt J, Boyd M, Chen Y, Zhao X, Schmidl C, Suzuki T, Ntini E, Arner E, Valen E, Li K, Schwarzfischer L, Glatz D, Raithel J, Lilje B, Rapin N, Bagger FO et al (2014) An atlas of active enhancers across human cell types and tissues. *Nature* **507**: 455-461
- Basso K, Saito M, Sumazin P, Margolin AA, Wang K, Lim WK, Kitagawa Y, Schneider C, Alvarez MJ, Califano A, Dalla-Favera R (2010) Integrated biochemical and computational approach identifies BCL6 direct target genes controlling multiple pathways in normal germinal center B cells. *Blood* **115**: 975-984
- Beekman R, Chapaprieta V, Russinol N, Vilarrasa-Blasi R, Verdaguer-Dot N, Martens JHA, Duran-Ferrer M, Kulis M, Serra F, Javierre BM, Wingett SW, Clot G, Queiros AC, Castellano G, Blanc J, Gut M, Merkel A, Heath S, Vlasova A, Ullrich S et al (2018) The reference epigenome and regulatory chromatin landscape of chronic lymphocytic leukemia. *Nat Med* **24**: 868-880
- Berman BP, Weisenberger DJ, Aman JF, Hinoue T, Ramjan Z, Liu Y, Noushmehr H, Lange CP, van Dijk CM, Tollenaar RA, Van Den Berg D, Laird PW (2011) Regions of focal DNA hypermethylation and long-range hypomethylation in colorectal cancer coincide with nuclear lamina-associated domains. *Nat Genet* **44**: 40-46
- Calo E, Wysocka J (2013) Modification of enhancer chromatin: what, how, and why? *Mol Cell* **49**: 825-837
- Chen H, Li C, Peng X, Zhou Z, Weinstein JN, Cancer Genome Atlas Research N, Liang H (2018) A Pan-Cancer Analysis of Enhancer Expression in Nearly 9000 Patient Samples. *Cell* **173**: 386-399 e312
- Erdel F, Krug J, Langst G, Rippe K (2011) Targeting chromatin remodelers: signals and search mechanisms. *Biochim Biophys Acta* **1809**: 497-508
- Local A, Huang H, Albuquerque CP, Singh N, Lee AY, Wang W, Wang C, Hsia JE, Shiau AK, Ge K, Corbett KD, Wang D, Zhou H, Ren B (2018) Identification of H3K4me1-associated proteins at mammalian enhancers. *Nat Genet* **50**: 73-82
- Loven J, Hoke HA, Lin CY, Lau A, Orlando DA, Vakoc CR, Bradner JE, Lee TI, Young RA (2013) Selective inhibition of tumor oncogenes by disruption of super-enhancers. *Cell* **153**: 320-334
- Maricque BB, Chaudhari HG, Cohen BA (2019) A massively parallel reporter assay dissects the influence of chromatin structure on cis-regulatory activity. *Nat Biotechnol* **37**: 90-95
- Merkenschlager M, Nora EP (2016) CTCF and Cohesin in Genome Folding and Transcriptional Gene Regulation. *Annu Rev Genomics Hum Genet* **17**: 17-43
- Nora EP, Goloborodko A, Valton AL, Gibcus JH, Uebersohn A, Abdennur N, Dekker J, Mirny LA, Bruneau BG (2017) Targeted Degradation of CTCF Decouples Local Insulation of Chromosome Domains from Genomic Compartmentalization. *Cell* **169**: 930-944 e922
- Oakes CC, Seifert M, Assenov Y, Gu L, Przekopowicz M, Ruppert AS, Wang Q, Imbusch CD, Serva A, Koser SD, Brocks D, Lipka DB, Bogatyrova O, Weichenhan D, Brors B, Rassenti L, Kipps TJ, Mertens D, Zapatka M, Lichter P et al (2016) DNA methylation dynamics during B

cell maturation underlie a continuum of disease phenotypes in chronic lymphocytic leukemia. *Nat Genet* **48**: 253-264

Ott CJ, Federation AJ, Schwartz LS, Kasar S, Klitgaard JL, Lenci R, Li Q, Lawlor M, Fernandes SM, Souza A, Polaski D, Gadi D, Freedman ML, Brown JR, Bradner JE (2018) Enhancer Architecture and Essential Core Regulatory Circuitry of Chronic Lymphocytic Leukemia. *Cancer Cell* **34**: 982-995 e987

Pradeepa MM, Grimes GR, Kumar Y, Olley G, Taylor GC, Schneider R, Bickmore WA (2016) Histone H3 globular domain acetylation identifies a new class of enhancers. *Nat Genet* **48**: 681-686

Soufi A, Garcia MF, Jaroszewicz A, Osman N, Pellegrini M, Zaret KS (2015) Pioneer transcription factors target partial DNA motifs on nucleosomes to initiate reprogramming. *Cell* **161**: 555-568

Teif VB, Beshnova DA, Vainshtein Y, Marth C, Mallm JP, Hofer T, Rippe K (2014) Nucleosome repositioning links DNA (de)methylation and differential CTCF binding during stem cell development. *Genome Res* **24**: 1285-1295

Teif VB, Vainshtein Y, Caudron-Herger M, Mallm JP, Marth C, Hofer T, Rippe K (2012) Genome-wide nucleosome positioning during embryonic stem cell development. *Nat Struct Mol Biol* **19**: 1185-1192

Weintraub AS, Li CH, Zamudio AV, Sigova AA, Hannett NM, Day DS, Abraham BJ, Cohen MA, Nabet B, Buckley DL, Guo YE, Hnisz D, Jaenisch R, Bradner JE, Gray NS, Young RA (2017) YY1 Is a Structural Regulator of Enhancer-Promoter Loops. *Cell* **171**: 1573-1588 e1528

Whyte WA, Orlando DA, Hnisz D, Abraham BJ, Lin CY, Kagey MH, Rahl PB, Lee TI, Young RA (2013) Master transcription factors and mediator establish super-enhancers at key cell identity genes. *Cell* **153**: 307-319

Zhang Y, Hardison RC (2017) Accurate and reproducible functional maps in 127 human cell types via 2D genome segmentation. *Nucleic Acids Res* **45**: 9823-9836

Thank you again for sending us your revised manuscript. We are now satisfied with the modifications made and I am pleased to inform you that your paper has been accepted for publication.

Corresponding Author Name: Karsten Rippe, Daniel Mertens

Manuscript Number: MSB-18-8339